# RoboDepth: Robust Out-of-Distribution Depth Estimation under Corruptions

Lingdong Kong[1,2]    Shaoyuan Xie[3]    Hanjiang Hu[4]    Lai Xing Ng[5,6]
Benoit R. Cottereau[2,6,7]    Wei Tsang Ooi[1,6]
[1]National University of Singapore    [2]CNRS@CREATE
[3]Huazhong University of Science and Technology
[4]Carnegie Mellon University    [5]Institute for Infocomm Research, A*STAR
[6]IPAL, CNRS IRL 2955, Singapore    [7]CerCo, CNRS UMR 5549, Université Toulouse III

https://github.com/ldkong1205/RoboDepth

## Abstract

Depth estimation from monocular images is pivotal for real-world visual perception systems. While current learning-based depth estimation models train and test on meticulously curated data, they often overlook out-of-distribution (OoD) situations. Yet, in practical settings – especially safety-critical ones like autonomous driving – common corruptions can arise. Addressing this oversight, we introduce a comprehensive robustness test suite, *RoboDepth*, encompassing **18** corruptions spanning three categories: *i)* weather and lighting conditions; *ii)* sensor failures and movement; and *iii)* data processing anomalies. We subsequently benchmark **42** depth estimation models across indoor and outdoor scenes to assess their resilience to these corruptions. Our findings underscore that, in the absence of a dedicated robustness evaluation framework, many leading depth estimation models may be susceptible to typical corruptions. We delve into design considerations for crafting more robust depth estimation models, touching upon pre-training, augmentation, modality, model capacity, and learning paradigms. We anticipate our benchmark will establish a foundational platform for advancing robust OoD depth estimation.

## 1  Introduction

Monocular depth estimation (MDE) involves predicting a scene's depth information from monocular images, without relying on data acquired from more sophisticated sensors [11, 28, 25, 37]. These images are predominantly captured using RGB cameras mounted on diverse platforms like drones, mobile robots, and vehicles [65, 41, 9, 30]. As an instrumental facet of visual perception, precise MDE paves the way for a broad array of applications. Bolstered by the rise of learning-based paradigms, numerous MDE algorithms have emerged, demonstrating remarkable depth estimation performances on standard benchmark datasets [14, 52, 49, 8, 47].

However, the resilience of existing MDE models to out-of-distribution (OoD) challenges is yet to be thoroughly explored, especially under the lens of real-world corruptions such as adverse weather [19, 46] and sensor malfunctions [29]. The prevailing learning-based visual perception models often display heightened sensitivity to nuances in lighting, noise, texture variations, among other factors, which are compromising the accuracy of depth predictions [18, 23]. The ability to generalize across new scenes, objects, and backgrounds, especially when they have not been part of the training data, is another pivotal challenge [40].

Despite the strides achieved on relatively pristine datasets [14, 49, 8], a lacuna exists: a robustness benchmark tailored to foster the evolution of resilient and scalable MDE systems. In light of these

challenges, models dedicated to MDE often inadvertently embed systematic errors, stemming from real-world image imperfections like altered lighting, motion blur, shadows, and data compression, which the current MDE solutions rarely address effectively [4, 34].

Seeking to bridge this gap, our contribution charts the inaugural path towards robust and reliable MDE, unveiling the *KITTI-C*, *NYUDepth2-C*, and *KITTI-S* benchmarks. Contrasting prior works that merged datasets for cross-domain MDE [44, 33, 62] or devised adversarial patches to subvert MDE models [7, 4], our benchmarks meticulously simulate commonplace corruptions that are intrinsic to real-world settings. As delineated in Fig. 2, we structure eighteen corruption varieties across *three* cardinal categories: *i)* weather and lighting conditions, *ii)* sensor malfunctions and movements, and *iii)* data processing complications. Further stratified by diverse severity, these corruptions encapsulate a gamut of scenarios fostering image distortions, texture shifts, or degraded visuals [18, 40].

Given that MDE models intrinsically depend on lucid visual cues for depth inference, the aforementioned corruptions naturally impose significant hurdles. Our preliminary analysis, visualized in Fig.1, showcases a spectrum of responses from distinct MDE model architectures when faced with diverse corruptions. Penetrating into these dynamics is quintessential to understanding the underlying causes of performance faltering, enabling us to architect MDE models that are both robust and reliable. Pursuing this vision, we undertake a meticulous benchmarking of extant MDE models on these new datasets, embarking on an exhaustive study of their robustness vis-a-vis the spectrum of corruptions. We probe queries about the resilience of MDE models to real-world corruptions, the influence of training input modalities, and learning paradigms – addressed in Sec.4.2. Concurrently, we assess the fidelity of our simulated corruptions and delve into the impact of texture-shift corruptions (style alterations) on gauging MDE model resilience.

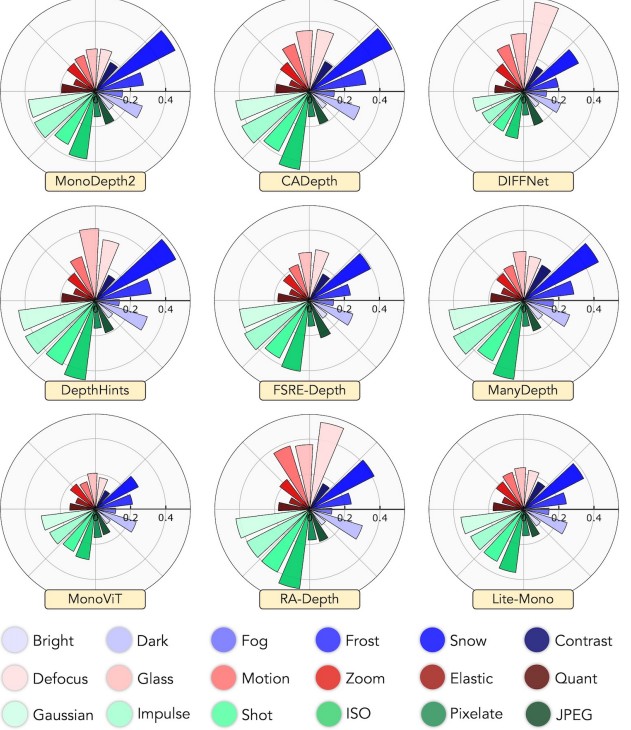

Figure 1: The depth estimation robustness (in terms of depth estimation error (DEE) defined in Sec. 3.3) under 18 corruptions in radar charts. Different MDE models exhibit diverse strengths and weaknesses against different corruptions that occur in the real world.

From our benchmark findings, we distill several intriguing insights and proffer recommendations to amplify robustness – focusing on strategies like model pre-training, input resolution tuning, model sizing, complexity modulation, and corrupt-image fine-tuning (Sec. 4.3).

To encapsulate, our work offers the following seminal contributions:

⋄ We introduce **RoboDepth**, the first systematically designed robustness evaluation suite for MDE under data corruptions, sensor failure, and style shifts. See our repository at this link for more details.

⋄ We benchmark 42 state-of-the-art MDE models from indoor and outdoor scenes, on their robustness against corruptions, via three newly established datasets: *KITTI-C*, *NYUDepth2-C*, and *KITTI-S*. The corruption simulation toolkit has been open-sourced to facilitate future development.

⋄ Based on our observations, we draw in-depth discussion and analysis on the design considerations of building more robust MDE models for reliable, scalable, and practical applications.

⋄ Furthermore, we initiated the *RoboDepth Challenge* [27], which garnered participation from over one hundred teams, underscoring the community's interest and the challenge's relevance. Comprehensive details about the competition can be accessed on our competition website at this link.

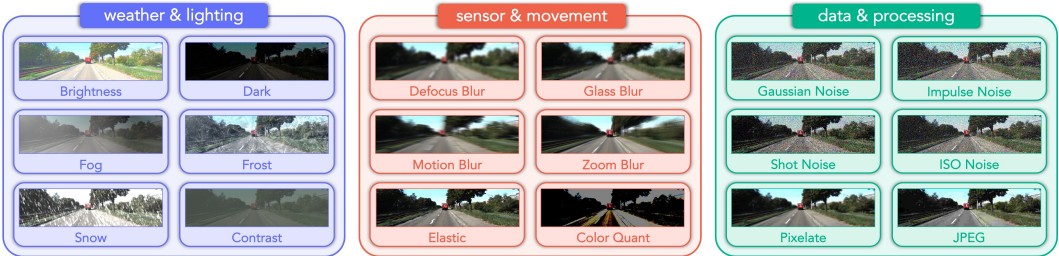

Figure 2: **Corruption taxonomy**. We break down common corruptions in depth estimation scenarios into *three* categories: *i)* Weather and lighting conditions, such as sunny, low-light, fog, frost, snow, and contrast conditions. *ii)* Sensor failure and movement, such as potential blurs (defocus, glass, motion, zoom) caused by motion. *iii)* Data processing issues, such as noises (Gaussian, impulse, ISO) happen due to hardware malfunctions. Examples shown are from the proposed *KITTI-C* benchmark.

## 2 Related Work

**Monocular Depth Estimation (MDE)**. Since the pioneering works [10, 12, 68, 16] first adopted deep neural networks to perform monocular depth estimation, significant progress has been made in many aspects. Notable innovations include network architectures [31, 43, 66, 63, 21], optimization functions [17, 64, 6], internal constraints [60, 67], multi-task learning [55, 22], geometry constraint [56, 51], and various sources of supervisions [44, 50, 33]. Based on the learning paradigm, most MDE methods can be split into supervised or self-supervised models. The former mainly focuses on indoor scenes and uses ground truth from RGB-D cameras or LiDAR sensors to train a regression model [1, 35]; while the latter formulates MDE as a novel view synthesis task to minimize the photometric loss between stereo pairs or from monocular video frames [68]. Although promising results have been achieved, the robustness of MDE models under adverse scenarios is still unknown. Due to the lack of relevant datasets, existing models are at risk of being vulnerable to corruptions. In this work, we fill in this gap by establishing comprehensive evaluation benchmarks and testing 42 MDE models from both indoor and outdoor environments to analyze their OoD robustness.

**Robust MDE**. To the best of our knowledge, only a few works targeted robust learning of MDE and they focused on different aspects. Ranftl *et al.* [44] proposed a unified objective for merging multiple datasets with different depth scales and ranges for training robust models. Similar works [58, 33, 53, 5, 62] resort to web stereo data or 3D movies to train MDE models and adapt them to unseen datasets. Kopf *et al.* [29] estimate stable camera trajectories for hand-held cellphone videos. SC-DepthV3 [50] generates pseudo-depth to refine depth details for scenes with dynamic objects. Li *et al.* [32] proposed an attention module to choose scene-specific

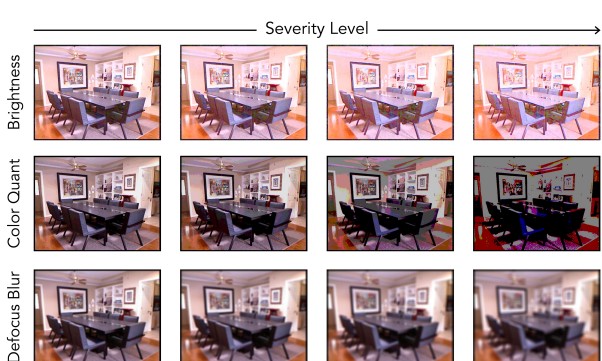

Figure 3: **Corruption severity level**. We create versatile corruption sets with different levels of severity. Examples shown are from the proposed *NYUDepth2-C* benchmark.

features for MDE on both indoor and outdoor scenes. SeasonDepth [19] contributed a dataset with depth maps under sunny, cloudy, and foliage weather. Most recently, there are works [7, 4] design adversarial patches to attack MDE models. Conversely, we aim to test the MDE robustness to corruptions that occur in real-world environments. We establish the first benchmark of this kind and incorporate an ample number of MDE models for in-depth analysis.

**Corruption Robustness**. ImageNet-C [18] is the pioneering work in this line of research which benchmarks classical image classification models to common corruptions and perturbations. Follow-up studies extend on the aspect to other visual perception tasks, *e.g.*, object detection [40], segmentation [23, 26, 38], navigation [3], video classification [61], and pose estimation [54]. The essentiality of evaluating model robustness has been repeatedly validated. Since we are targeting a different task, *i.e.*, MDE, most of the well-studied corruption types become less realistic or suitable for such a data

format. This motivates us to explore a new taxonomy for defining more proper corruption types for MDE. In this work, we contribute new datasets and benchmarks for probing the MDE robustness.

## 3 The RoboDepth Benchmark

In this section, we first introduce the taxonomy of corruptions included in our benchmarks (Sec. 3.1). We then elaborate on more details of the proposed datasets (Sec. 3.2) and corresponding robustness evaluation metrics (Sec. 3.3). Examples from our datasets are shown in Fig. 2, Fig. 3, and this page.

### 3.1 Corruption Definition

**Weather & Lighting Condition**. The cameras on drones or vehicles operating under different weather and times of day capture distribution-shifted images which are rare or lacking in current MDE datasets [14, 49]. To probe the robustness of MDE models under adverse weather and lighting conditions, we simulate six corruptions, *i.e.*, 'brightness', 'dark', 'fog', 'frost', 'snow', and 'contrast', which commonly occur in the real-world environment. Compared to clean images, these corruptions tend to affect the intensity and color of the light source, leading to hazy, blurry, and noise-contaminated images, which increase the difficulties for the MDE model to make accurate depth predictions.

**Sensor Failure & Movement**. An MDE system must behave robustly against motion perturbation and sensor failure to maintain safety requirements for practical applications. To achieve this pursuit, we mimic four motion-related corruptions, *i.e.*, 'defocus', 'glass', 'motion', and 'zoom' blurs; we also generate images under 'elastic transformation' and 'color quantization', which happen during sensor malfunction. These corruption types are often associated with issues including edge distortions, contrast loss, and pattern shifts.

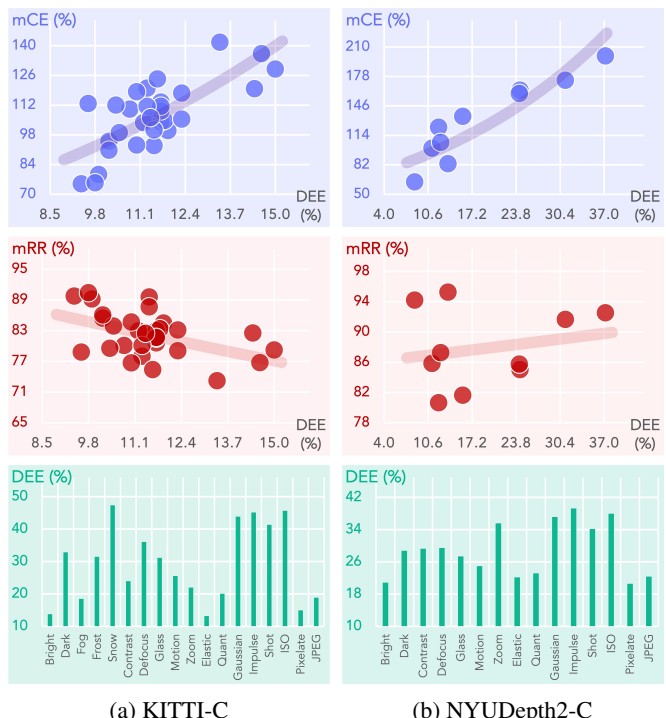

(a) KITTI-C      (b) NYUDepth2-C

Figure 4: Benchmarking results of **42** MDE models on *KITTI-C* and *NYUDepth2-C*. Figures from top to bottom: the depth estimation error (DEE) *vs.* **[1st row]** mean corruption error (mCE), **[2nd row]** mean resilience rate (mRR), and **[3rd row]** sensitivity analysis among different corruption types.

**Data & Processing Issue**. Data collection and transmission are inevitably associated with various sources of noise and potential loss of information. We include four such random variations, *i.e.*, 'Gaussian', 'impulse', 'shot', and 'ISO' noises. In addition, we investigate the degradation caused by 'pixelate' and 'JPEG compression' which are common corruptions in handling image data. Compared to clean images, the noise-contaminated data introduce errors in the intensity values of pixels, leading to a grainy or speckled appearance. The pixelation and lossy compression tend to lead to a loss of detail and clarity in the image and can result in visible artifacts, such as blockiness or blurring.

### 3.2 Benchmark Establishment

**KITTI-C**. Based on the KITTI Vision Suite [14], we establish a robustness benchmark for outdoor MDE. We simulate the defined 18 corruptions using data from the KITTI *val* set under Eigen's split. Similar to [18], we design five severity levels for each corruption to further consolidate the evaluation

of robustness changes. As a result, this robustness probing dataset has a total number of $62,730$ RGB images with a resolution of $192 \times 640$. We also include the high-resolution version ($320 \times 1024$) for evaluating the robustness of MDE models which take larger images as the input.

**NYUDepth2-C**. We construct a benchmark for robust indoor MDE based on NYU Depth V2 [49]. 15 of the defined corruptions are used, excluding 'fog', 'frost', and 'snow' which rarely occur in the indoor scenes. Since the indoor environments are less variant than outdoor ones, we only include four severity levels for each corruption. To sum up, this dataset contains $39,240$ images of size $480 \times 640$, which cover 23 different types of indoor scenes, such as basement, bathroom, bedroom, study, *etc*.

**KITTI-S**. Style changes, consisting mostly of texture shifts, have proven helpful for analyzing model robustness [18]. To further investigate the root cause of MDE robustness degradation, we form another collection based on KITTI [14] with stylized images via the style transfer model AdaIn [20]. This dataset has $8,364$ images from 12 styles, including 'cartoon', 'digital art', 'ink painting', 'kids' drawing', 'murals', 'oil painting', 'penciling', 'shadow play', 'sketch', 'stained glass', 'relief', and 'water color'. Due to space limitations, please refer to Appendix or this page for additional examples.

**Simulation Toolkit**. To facilitate a similar study on other MDE datasets, we have open-sourced the code at this link for simulating corruptions and style shifts given arbitrary "clean" images.

### 3.3 Evaluation Metrics

**Depth Estimation Error (DEE)**. We combine Abs Rel (error rate) and $\delta_1$ (accuracy), the two main measures defined in [10, 36], into a unified metric as $\text{DEE} = \frac{\text{Abs Rel} - \delta_1 + 1}{2}$, which is constantly used as the indicator of depth estimation error in our benchmark. See Appendix for more formal definitions.

**Corruption Error (CE)**. We follow [18] and use the mean CE (mCE) as the primary metric in comparing models' robustness. To normalize the severity effects, we choose MonoDepth2 [17] and AdaBins [1] as the baseline models for the *KITTI-C* and *NYUDepth2-C* benchmarks, respectively. The CE across $L$ levels of severity and mCE across $N$ corruption types can be calculated as follows:

$$\text{CE}_i = \frac{\sum_{l=1}^{L}(\text{DEE}_{i,l})}{\sum_{l=1}^{L}(\text{DEE}_{i,l}^{\text{baseline}})}, \quad \text{mCE} = \frac{1}{N}\sum_{i=1}^{N}\text{CE}_i . \tag{1}$$

**Resilience Rate (RR)**. We define mean RR (mRR) as the relative robustness indicator for measuring how much accuracy can an MDE model retain when evaluated under the corruption scenarios, *i.e.*,

$$\text{RR}_i = \frac{\sum_{l=1}^{L}(1 - \text{DEE}_{i,l})}{L \times (1 - \text{DEE}_{\text{clean}})}, \quad \text{mRR} = \frac{1}{N}\sum_{i=1}^{N}\text{RR}_i , \tag{2}$$

where $\text{DEE}_{\text{clean}}$ denotes the task-specific accuracy (or error rate) score on the "clean" evaluation set.

## 4 Experiments

### 4.1 Benchmark Configuration

**Depth Estimation Models**. We benchmark 42 depth estimation models and model variants, which cover most of the open-source MDE models so far. 32 of them are for outdoor MDE and the remaining 10 are for indoor MDE. More detailed descriptions of these models are attached in the Appendix.

**Datasets**. All benchmarked depth estimation models have been trained on the official *training* splits of the *KITTI* [14] (for outdoor MDE) or *NYU-Depth V2* [49] (for indoor MDE) datasets, and are tested accordingly on the official *val* splits and also our proposed *KITTI-C*, *NYUDepth2-C*, and *KITTI-S* datasets. Additionally, we resort to the real-world *ACDC* [45], *nuScenes* [2], *Cityscapes* [8], and *Foggy-Cityscapes* [46] datasets for validating the fidelity of our simulated corruptions. Our datasets and model evaluation toolkit can be downloaded from this page.

**Evaluation Protocols**. To avoid any unfairness in the MDE robustness comparison, we unify the common configurations among different candidate models, such as *backbones*, *data augmentations*, and *post-processing*. We use public checkpoints whenever possible and reproduce the reported results based on official settings. More details on this aspect are included in the Appendix.

Table 1: **Self-Supervised MDE Robustness Benchmark** consisting of 32 models on *KITTI-C*. The mCE and mRR scores are given in percentage (%). Blocks from top to bottom: **[1st]:** The baseline MonoDepth2 R18 [17]; **[2nd]:** Methods *w/* monocular inputs; **[3rd]:** Methods *w/* stereo inputs; **[4th]:** Methods *w/* monocular + stereo inputs. **Bold**: Best in column. Underline: Second best in column. The best score of each metric across three categories is shaded in color.

| Method | Overall Robustness | | | Weather & Lightning | | | Sensor & Movement | | | Data & Processing | | |
|---|---|---|---|---|---|---|---|---|---|---|---|---|
| | mCE | mRR | mDEE | mCE | mRR | mDEE | mCE | mRR | mDEE | mCE | mRR | mDEE |
| MonoDepth2 R18 [17] | 100.00 | 84.46 | 0.256 | 100.00 | 84.37 | 0.257 | 100.00 | 90.33 | 0.204 | 100.00 | 78.66 | 0.307 |
| MonoDepth2 nopt [17] | 119.75 | 82.50 | 0.294 | 146.17 | 78.58 | 0.327 | 103.28 | 92.45 | 0.209 | 109.80 | 76.48 | 0.345 |
| MonoDepth2 HR [17] | 106.06 | 82.44 | 0.270 | 109.95 | 80.38 | 0.288 | 115.99 | 85.59 | 0.242 | 92.25 | 81.36 | 0.279 |
| MonoDepth2 R50 [17] | 113.43 | 80.59 | 0.288 | 104.53 | 83.28 | 0.265 | 128.68 | 82.45 | 0.272 | 107.07 | 76.05 | 0.329 |
| MaskOcc R18 [48] | 104.05 | 82.97 | 0.267 | 100.98 | 84.13 | 0.257 | 108.85 | 87.67 | 0.226 | 102.30 | 77.12 | 0.319 |
| DNet R18 [59] | 104.71 | 83.34 | 0.265 | 103.04 | 83.56 | 0.263 | 115.56 | 86.07 | 0.241 | 95.53 | 80.39 | 0.291 |
| CADepth [60] | 110.11 | 80.07 | 0.286 | 102.59 | 82.04 | 0.268 | 119.67 | 83.91 | 0.252 | 108.06 | 74.25 | 0.338 |
| HR-Depth [39] | 103.73 | 82.93 | 0.264 | 100.41 | 83.82 | 0.256 | 116.01 | 85.32 | 0.242 | 94.76 | 79.64 | 0.293 |
| DIFFNet [67] | 94.96 | 85.41 | 0.233 | 79.33 | 89.53 | 0.196 | 124.69 | 81.68 | 0.267 | 80.86 | 85.00 | 0.237 |
| ManyDepth R18 [56] | 105.41 | 83.11 | 0.271 | 104.97 | 84.15 | 0.262 | 102.57 | 90.06 | 0.210 | 108.70 | 75.10 | 0.341 |
| FSRE-Depth [22] | 99.05 | 83.86 | 0.253 | 89.12 | 87.39 | 0.221 | 101.36 | 88.65 | 0.210 | 106.66 | 75.53 | 0.327 |
| MonoViT [66] | 79.33 | 89.15 | 0.197 | 72.92 | 91.16 | 0.179 | 81.62 | 92.67 | 0.165 | 83.47 | 83.61 | 0.247 |
| MonoViT HR [66] | 74.95 | 89.72 | 0.187 | 72.24 | 90.78 | 0.177 | 74.60 | 93.86 | 0.150 | 78.02 | 84.51 | 0.234 |
| DynaDepth R18 [64] | 110.38 | 81.50 | 0.280 | 102.92 | 83.45 | 0.263 | 131.29 | 81.67 | 0.279 | 96.95 | 79.39 | 0.299 |
| DynaDepth R50 [64] | 119.99 | 77.98 | 0.308 | 105.81 | 81.70 | 0.275 | 143.38 | 78.09 | 0.307 | 110.79 | 74.15 | 0.342 |
| RA-Depth [42] | 112.73 | 78.79 | 0.288 | 89.39 | 85.34 | 0.229 | 137.47 | 78.19 | 0.293 | 111.32 | 72.82 | 0.342 |
| TriDepth R18 [6] | 109.26 | 81.56 | 0.280 | 115.07 | 80.79 | 0.287 | 104.61 | 88.79 | 0.216 | 108.10 | 75.10 | 0.337 |
| Lite-Mono Tiny [63] | 92.92 | 86.69 | 0.233 | 90.57 | 88.31 | 0.219 | 95.47 | 90.87 | 0.196 | 92.71 | 80.90 | 0.284 |
| Lite-Mono Small [63] | 100.34 | 84.67 | 0.251 | 98.98 | 85.52 | 0.243 | 109.87 | 87.12 | 0.229 | 92.16 | 81.37 | 0.280 |
| Lite-Mono Base [63] | 93.16 | 85.99 | 0.235 | 89.20 | 87.49 | 0.221 | 96.71 | 90.22 | 0.197 | 93.59 | 80.26 | 0.286 |
| Lite-Mono Large [63] | 90.75 | 85.54 | 0.232 | 83.91 | 87.23 | 0.217 | 91.97 | 90.42 | 0.188 | 96.38 | 78.95 | 0.291 |
| MonoDepth2 R18 [17] | 117.69 | 79.05 | 0.307 | 111.08 | 81.79 | 0.283 | 121.92 | 84.95 | 0.255 | 120.08 | 70.41 | 0.383 |
| MonoDepth2 nopt [17] | 128.98 | 79.20 | 0.327 | 145.57 | 77.96 | 0.337 | 110.41 | 91.45 | 0.223 | 130.97 | 68.20 | 0.420 |
| MonoDepth2 HR [17] | 111.46 | 81.65 | 0.279 | 111.73 | 81.03 | 0.284 | 127.51 | 82.96 | 0.268 | 95.16 | 80.96 | 0.285 |
| DepthHints [57] | 111.41 | 80.08 | 0.290 | 99.81 | 83.22 | 0.262 | 122.07 | 83.78 | 0.257 | 112.36 | 73.22 | 0.351 |
| DepthHints nopt [57] | 141.61 | 73.18 | 0.366 | 155.51 | 73.52 | 0.363 | 121.71 | 86.47 | 0.251 | 147.60 | 59.57 | 0.484 |
| DepthHints HR [57] | 112.02 | 79.53 | 0.287 | 98.36 | 83.30 | 0.254 | 133.82 | 79.89 | 0.284 | 103.88 | 75.41 | 0.324 |
| MonoDepth2 R18 [17] | 124.31 | 75.36 | 0.334 | 109.43 | 80.86 | 0.285 | 127.18 | 82.75 | 0.269 | 136.34 | 62.46 | 0.448 |
| MonoDepth2 nopt [17] | 136.25 | 76.72 | 0.345 | 165.90 | 72.79 | 0.378 | 106.08 | 92.17 | 0.213 | 136.77 | 65.20 | 0.443 |
| MonoDepth2 HR [17] | 106.06 | 82.44 | 0.270 | 109.95 | 80.38 | 0.288 | 115.99 | 85.59 | 0.242 | 92.25 | 81.36 | 0.279 |
| CADepth [60] | 118.29 | 76.68 | 0.318 | 103.66 | 81.40 | 0.276 | 119.87 | 83.99 | 0.253 | 131.33 | 64.64 | 0.425 |
| MonoViT [66] | 75.39 | 90.49 | 0.184 | 72.54 | 91.83 | 0.172 | 78.42 | 93.29 | 0.159 | 75.21 | 86.35 | 0.221 |

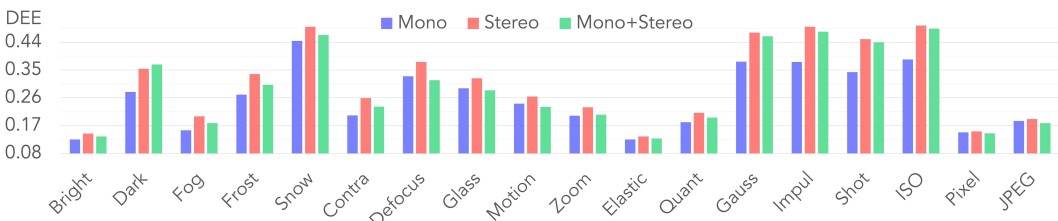

Figure 5: The depth estimation robustness comparisons among different input modalities. **Mono:** monocular images; **Stereo:** stereo pairs; and **Mono+Stereo:** both monocular and stereo images.

## 4.2 MDE Robustness Probing

In this section, based on our benchmark, we aim to understand the corruption robustness among different MDE models by answering the following representative questions.

**Q-1:** *"Are existing MDE models robust under real-world corruptions?"* **A:** No, to a certain extent. Our benchmark results in Fig. 4 reveal that state-of-the-art MDE models are at risk of being vulnerable to corruptions. Existing MDE models, either from indoor or outdoor scenes, show a flattened or even inverse relationship between the DEE scores and robustness metrics. Due to the lack of a suitable robustness evaluation suite, current MDE models are over-fitted on "clean" sets while ignoring the OoD scenarios that are likely to occur in the real world. Nevertheless, we observe varying behaviors of different MDE models under different corruption types (see Fig. 1), caused by different design choices on model architecture. The results from Tab. 1 and Tab. 2 further show that the Transformers-based MDE models exhibit better robustness compared to conventional CNNs. Diving deeper, we can observe from the per-severity error rates in Fig. 9 that the above conclusion holds true for most corruption types under different severity levels. The qualitative results shown in Fig. 10 also validate that models with long-range receptive fields, such as MonoViT [66] (74.95% mCE) and Lite-Mono

Table 2: **Supervised MDE Robustness Benchmark** consisting of 10 models on the *NYUDepth2-C* dataset. The mCE and mRR scores are given in percentage (%). **Bold**: Best in column. Underline: Second best in column. The best score of each metric across three categories is shaded in color.

| Method | Overall Robustness | | | Weather & Lightning | | | Sensor & Movement | | | Data & Processing | | |
|---|---|---|---|---|---|---|---|---|---|---|---|---|
| | mCE | mRR | mDEE | mCE | mRR | mDEE | mCE | mRR | mDEE | mCE | mRR | mDEE |
| AdaBins [EB5] [1] | 100.00 | 85.83 | 0.238 | 100.00 | 92.42 | 0.179 | 100.00 | 87.97 | 0.219 | 100.00 | 80.41 | 0.286 |
| BTS [R50] [31] | 122.78 | 80.63 | 0.292 | 125.51 | 87.97 | 0.228 | 121.86 | 84.21 | 0.261 | 122.34 | 73.39 | 0.356 |
| AdaBins [R50] [1] | 134.69 | 81.62 | 0.313 | 140.99 | 88.64 | 0.254 | 131.88 | 85.63 | 0.279 | 134.37 | 74.11 | 0.376 |
| DPT [ViT-B] [43] | 83.22 | **95.25** | 0.177 | 93.66 | **96.57** | 0.166 | 81.82 | **96.24** | 0.168 | 79.41 | 93.60 | 0.191 |
| SimIPU [nopt] [34] | 200.17 | 92.52 | 0.419 | 241.79 | 92.20 | 0.421 | 201.63 | 94.96 | 0.404 | 177.90 | 90.23 | 0.433 |
| SimIPU [ImageNet] [34] | 163.06 | 85.01 | 0.357 | 190.62 | 87.52 | 0.338 | 166.90 | 86.97 | 0.343 | 145.45 | 81.79 | 0.382 |
| SimIPU [KITTI] [34] | 173.78 | 91.64 | 0.370 | 210.25 | 91.81 | 0.368 | 170.75 | 95.37 | 0.344 | 158.56 | 87.81 | 0.396 |
| SimIPU [WaymoOpen] [34] | 159.46 | 85.73 | 0.351 | 190.30 | 87.41 | 0.338 | 158.90 | 88.77 | 0.328 | 144.59 | 81.86 | 0.380 |
| DepthFormer [SwinT-1k] [35] | 106.34 | 87.25 | 0.237 | 122.01 | 89.10 | 0.220 | 107.22 | 88.80 | 0.223 | 97.64 | 84.78 | 0.258 |
| DepthFormer [SwinT-22k] [35] | **63.47** | 94.19 | **0.139** | **70.11** | 95.84 | **0.124** | **69.32** | 93.25 | **0.148** | **54.29** | **94.29** | **0.138** |

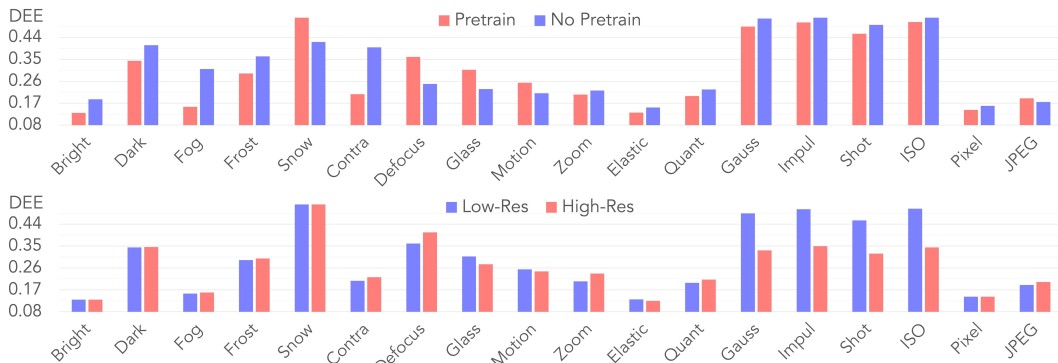

Figure 6: Depth estimation robustness of MonoDepth2 [17] under different training configurations. **[top]:** Different pre-training techniques. **[bottom]:** Different resolutions of the input images.

[63] (90.75% mCE), can better maintain accurate depth predictions under corruptions coped with texture shift, edge distortion, and noise contamination.

**Q-2:** *"Which MDE modality is the most robust against corruptions?"* **A:** MDE models trained with purely monocular images rather than stereo pairs as input have better robustness, as shown in Fig. 5. Here we analyze the robustness of MonoDepth2 [17] variants trained with different input modalities and observe that using stereo or a mix of monocular and stereo images leads to robustness degradation. MDE models trained with stereo pairs may rely more on the scene structural consistency between left-right pairs, which could be destroyed when dealing with corrupted data. Such constraints, however, could be relaxed if the depth estimation model is trained only on monocular sequences.

**Q-3:** *"Which learning paradigm is better in terms of corruption robustness?"* **A:** We find different sensitivities between the supervised and self-supervised MDE paradigms. From the third row of Fig. 4 we can observe that the self-supervised MDE models are less sensitive to lighting changes (texture shift) and motion blurs (local distortion), compared to the supervised models. On the other hand, both models suffer from noise perturbation and behave robustly against lossy image compression.

**Q-4:** *"Are simulated corruptions realistic enough to mimic real-world scenarios?"* **A:** Yes, to a certain extent. We verify this statement by augmenting MDE models via fine-tuning with our corruptions. Fig. 7 (right) shows that the generated corruptions are helpful for closing the distribution gap between the clean training set and real-world datasets [2, 8, 46], especially for the weather contaminated data in the *Foggy-Cityscapes* dataset [46]. This supports that our datasets are of high fidelity and scalability. Additionally, we show in the Appendix that the simulated corruptions, especially for the types that are related to weather and lighting changes, are of good quality compared to some real-world corrupted data [2, 45, 8].

**Q-5:** *"Are MDE models robust against style changes?"* **A:** We resort to the *KITTI-S* dataset (see Tab. 3) for probing model characteristics and find that most MDE models perform worse on this stylized dataset than on *KITTI-C*. We conjecture that the loss of texture cues for objects (*e.g.* car and building) for *KITTI-S* data causes the main impediment. We also observe that MDE models with

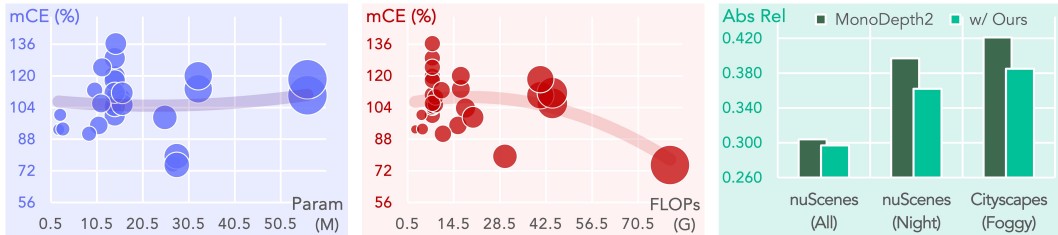

Figure 7: MDE robustness comparisons based on **[left]:** Model size (# of parameters) and **[middle]:** Model complexity (FLOPs); and **[right]:** Corruption fidelity verification by fine-tuning MonoDepth2 [17] with our corruptions on *KITTI* and testing it on the official *val* splits of *nuScenes* [2] (including its validation set and the nighttime split of the validation set) and *Foggy-Cityscapes* [46].

Table 3: The **Depth Estimation Error (DEE)** of 18 models on *KITTI-S*. **Bold**: Best in column. Underline: 2nd best in column. Blue : Best in row. Red : 2nd best in row. Green : Worst in row.

| Method | mDEE | Cart | Digit | Ink | Kids | Mural | Oil | Pencil | Shadow | Sketch | Glass | Relief | Water |
|---|---|---|---|---|---|---|---|---|---|---|---|---|---|
| MonoDepth2 R18 [17] | .365 | .324 | .434 | .351 | .259 | .326 | .328 | .418 | .388 | .416 | .566 | .255 | .317 |
| MonoDepth2 nopt [17] | .378 | .279 | .289 | .589 | .235 | .412 | .249 | .290 | .589 | .288 | .596 | .259 | .464 |
| MaskOcc R18 [48] | .368 | .358 | .356 | .260 | .265 | .358 | .375 | .333 | .314 | .576 | .554 | .288 | .384 |
| DNet R18 [59] | .400 | .444 | .381 | .293 | .280 | .389 | .336 | .422 | .290 | .608 | .553 | .290 | .516 |
| CADepth [60] | .406 | .446 | .380 | .379 | .250 | .421 | .470 | .317 | | .585 | .543 | .242 | .401 |
| HR-Depth [39] | .324 | .318 | .332 | .238 | .228 | .299 | .306 | .337 | .316 | .388 | .531 | .265 | .331 |
| DIFFNet [67] | .310 | .227 | .351 | .206 | .206 | .386 | .244 | .276 | .289 | .385 | .502 | .294 | .360 |
| ManyDepth [56] | .323 | .251 | .373 | .212 | .205 | .344 | .300 | .360 | .343 | .331 | .553 | .310 | .300 |
| FSREDepth [22] | .293 | .275 | .277 | .294 | .221 | .310 | .220 | .270 | .301 | .318 | .417 | .261 | .352 |
| MonoViT [66] | **.238** | **.179** | **.229** | .252 | **.196** | .240 | **.203** | **.237** | **.208** | .325 | **.356** | **.205** | **.221** |
| DynaDepth R18 [64] | .371 | .349 | .358 | .291 | .264 | .288 | .293 | .338 | .363 | .492 | .534 | .298 | .585 |
| DynaDepth R50 [64] | .447 | .502 | .408 | .259 | .287 | .398 | .437 | .492 | .372 | .595 | .565 | .460 | .589 |
| RA-Depth [42] | .365 | .304 | .335 | .354 | .262 | .340 | .310 | .342 | .425 | .372 | .476 | .379 | .475 |
| TriDepth R18 [6] | .379 | .304 | .403 | .256 | .217 | .440 | .352 | .480 | .318 | .517 | .573 | .249 | .436 |
| Lite-Mono Tiny [63] | .280 | .236 | .310 | .251 | .221 | .254 | .283 | .285 | .232 | .356 | .446 | .237 | .252 |
| Lite-Mono Small [63] | .303 | .210 | .346 | .218 | .210 | **.224** | .238 | .374 | .307 | .445 | .435 | .295 | .336 |
| Lite-Mono Base [63] | .288 | .234 | .313 | .262 | .218 | .231 | .241 | .291 | .240 | .459 | .476 | .229 | .264 |
| Lite-Mono Large [63] | .323 | .208 | .435 | .249 | .306 | .348 | .258 | .363 | .210 | .458 | .507 | .263 | .266 |

global awareness, *e.g.* Lite-Mono [63] and MonoViT [66], suffer less degradation in this case. Due to the page limit, kindly refer to the Appendix for more detailed findings on this style-shifted dataset.

### 4.3 MDE Robustness Enhancement

**Pertaining strategy tends to help improve MDE robustness**. We observe that transferring knowledge from other tasks, such as classification and segmentation, brings both strengths and weaknesses to MDE robustness. Fig. 6 (top) highlights that MDE models pre-trained on object-centric datasets, *e.g.* ImageNet, are more robust against weather and lighting changes (except for 'snow') and data processing noises, which are mostly texture-shifted corruptions. Motion and sensor corruptions, however, contain more edge and object distortions and could be eased by models without ImageNet pre-training. More concrete results in Tab. 1 imply that the CNN-based MDE models like MonoDepth2 [17] could become more shape-biased when pre-trained on object-centric datasets. More evidence from the *KITTI-S* benchmark (see Tab. 3) further verifies this finding, where the stylized data (texture-biased) are causing severe degradation for models with ImageNet pre-training.

**Training on high-resolution images yields more robust MDE models**. From Fig. 6 (bottom) we observe that MDE models trained with higher resolution inputs will likely yield more robust feature learning (relative $30\%$ better) on noise-contaminated corruptions (Gaussian, impulse, shot, ISO noises). Since these noise contamination types mainly affect the global pixel distribution instead of the local ones, the CNN-based MDE models trained with high-resolution images are likely to capture more fine-grained information to suppress the degradation caused by noises. For more details, kindly refer to the per-corruption scores of benchmarked models in the Appendix.

**Larger model size might not lead to better MDE robustness**. It is often intuitive that larger models are likely to learn more general representations and lead to better performance on unseen data. However, we observe from Fig. 7 (left) that MDE models with more trainable parameters are getting less robust, mainly because they are deliberately tuned towards clean distribution and thus

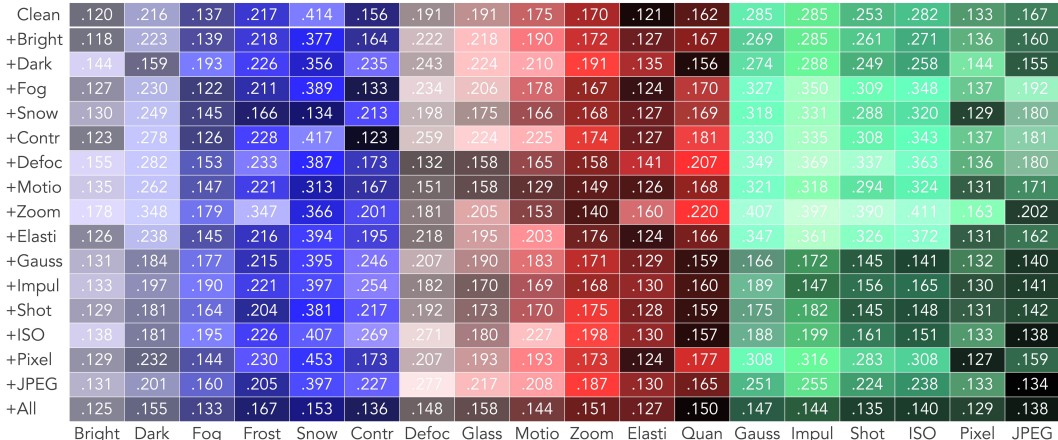

Figure 8: The **Per-Corruption Error** scores of MonoDepth2 [17] fine-tuned with typical corruption as data augmentations and tested on each of the corruption types. For each **column** (corruption) in this figure: The darker the color, the lower the Abs Rel score (better robustness), and vice versa.

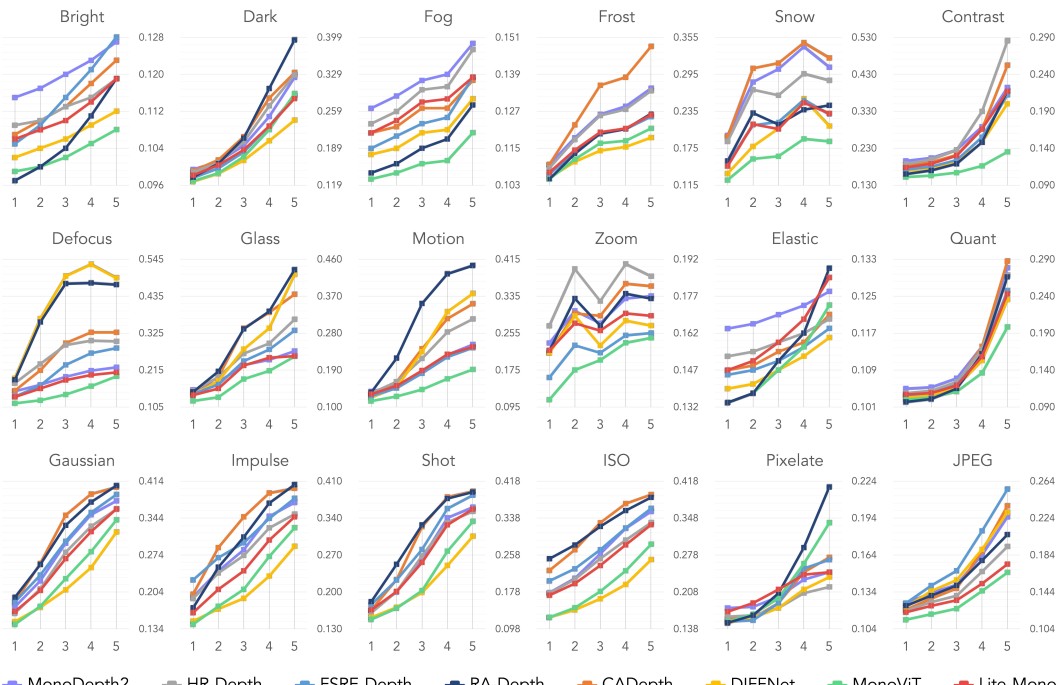

Figure 9: The **Per-Severity Error Rate** of eight typical MDE models [17, 39, 22, 42, 60, 67, 66, 63] trained on the clean *KITTI* dataset [14] and tested on each of the corruption set in *KITTI-C*. For each **subfigure** (corruption) in this figure: The horizontal axis denotes severity levels from one to five. The vertical axis denotes the depth estimation error (Abs Rel). Best viewed in color.

losing generalizability. Our results suggest that a moderate-sized model with suitable corruption suppression modules is likely to yield the best possible trade-off between robustness and efficiency.

**Model complexity shows a direct correlation with MDE robustness**. Based on Fig. 7 (middle) we reveal that higher training complexity tends to lead to better robustness. This is mainly because existing MDE models with high FLOPs per parameter are either trained with high-resolution monocular inputs or contain the computationally intensive self-attention mechanism and yield more stable depth predictions (as analyzed in previous discussions) but cause heavier training overhead.

**Training with corrupted images does not always enhance model robustness**. The robust fine-tuning study in Fig. 8 shows that overfitting the evaluation distribution (*e.g.* train and test on the same

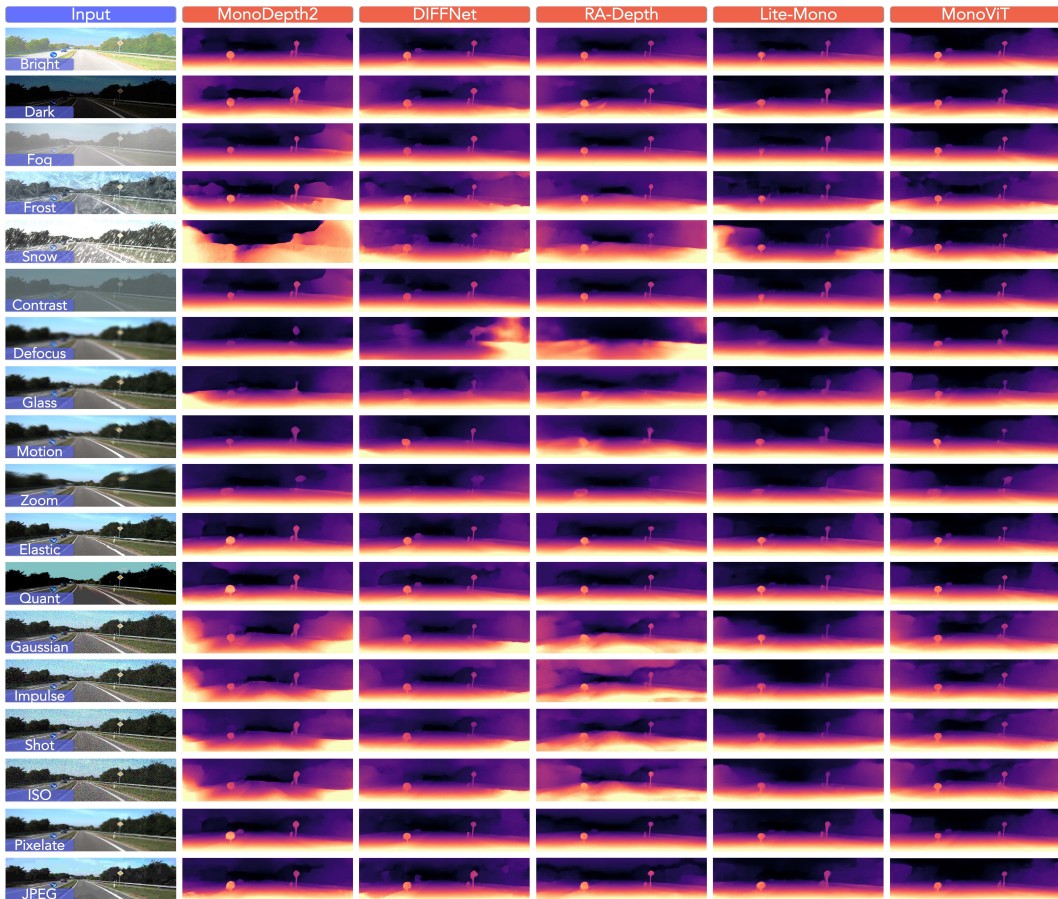

Figure 10: Qualitative results of representative MDE models [17, 67, 42, 63, 66] under defined corruptions in the *KITTI-C* benchmark. Best viewed in color and zoom-ed in for details.

corruption) may help improve MDE robustness for this specific type of corruption, as shown from the darker-colored cell in the diagonal. However, those corruptions with severe perturbations (weather and movement) will likely lead to sub-par performance on other tested corruptions and even hurt the overall robustness. This indicates that sophisticated considerations, *e.g.* case study, must be made before applying corruptions as data augmentations to the training stage.

# 5  Discussion & Conclusion

Throughout this research, we introduced the ***RoboDepth*** benchmark, a dedicated tool tailored for probing the OoD robustness of MDE models under a range of corruptions. By crafting three distinct datasets (*KITTI-C*, *NYUDepth2-C*, and *KITTI-S*) coupled with two metrics (mCE and mRR), we offered a comprehensive landscape for gauging both indoor and outdoor MDE robustness. Our findings underscore the imperative nature of robustness evaluation in the realm of depth estimation. Moreover, by dissecting various influential factors – encompassing architecture, modality, pre-training approaches, resolution, and model capacity – we provided insights to fortify model resilience against corruptions. In essence, our endeavors serve as a compass directing the evolution of robust MDE methodologies. As we navigate forward, our ambition is to augment the breadth and granularity of our benchmark, aiming to encompass a wider spectrum of MDE models and corruption variants.

**Potential Limitations**. Despite our extensive evaluation of diverse MDE models across expansive corruption sets, and validating the authenticity of our simulated corruptions, certain avenues remain uncharted. Notably, our current framework does not accommodate scenarios wherein multiple corruptions manifest simultaneously. Additionally, evolving from rigidly defined five severity levels to a more nuanced, continuous scale might offer deeper insights into MDE robustness. These unexplored terrains present intriguing prospects for subsequent research endeavors.

## Acknowledgements

This research is part of the programme DesCartes and is supported by the National Research Foundation, Prime Minister's Office, Singapore under its Campus for Research Excellence and Technological Enterprise (CREATE) programme.

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

# Appendix

In this appendix, we supplement the following materials to support the findings and conclusions drawn in the main body of this paper:

- Sec. A documents necessary information about the proposed datasets and benchmarks.
- Sec. B elaborates on more details in terms of benchmark definitions and implementations, as well as the details of data collection, organization, licensing, and access.
- Sec. C introduces The RoboDepth Challenge, an academic competition held based on the datasets and benchmarks constructed in this work.
- Sec. D provides the complete quantitative results of different monocular depth estimation models in the established *KITTI-C*, *NYUDepth2-C*, and *KITTI-S* benchmarks.
- Sec. E contains additional qualitative results of different monocular depth estimation models under out-of-distribution corruption scenarios.
- Sec. F acknowledges the public resources used during the course of this work.

## A   Datasheets

In this section, we follow Gebru *et al.* [13] to document necessary information about the proposed datasets and benchmarks.

### A.1   Motivation

The questions in this section are primarily intended to encourage dataset creators to clearly articulate their reasons for creating the dataset and to promote transparency about funding interests. The latter may be particularly relevant for datasets created for research purposes.

1. *"For what purpose was the dataset created?"* **A:** The dataset was created to facilitate relevant research in the area of depth estimation robustness under out-of-distribution corruptions.

2. *"Who created the dataset (e.g., which team, research group) and on behalf of which entity (e.g., company, institution, organization)?"* **A:** The dataset was created by Lingdong Kong (National University of Singapore), Shaoyuan Xie (Huazhong University of Science and Technology), Hanjiang Hu (Carnegie Mellon University), Lai Xing Ng (Institute for Infocomm Research, A*STAR), Benoit Cottereau (CNRS), and Wei Tsang Ooi (National University of Singapore).

3. *"Who funded the creation of the dataset?"* **A:** The creation of the dataset is funded by related affiliations of the authors in this work, as listed in the above item.

4. *"Any other comments?"* **A:** N/A.

### A.2   Composition

Most of the questions in this section are intended to provide dataset consumers with the information they need to make informed decisions about using the dataset for their chosen tasks. Some of the questions are designed to elicit information about compliance with the EU's General Data Protection Regulation (GDPR) or comparable regulations in other jurisdictions. Questions that apply only to datasets that relate to people are grouped together at the end of the section. We recommend taking a broad interpretation of whether a dataset relates to people. For example, any dataset containing text that was written by people relates to people.

1. *"What do the instances that comprise our datasets represent (e.g., documents, photos, people, countries)?"* **A:** The instances that comprise the dataset are mainly images captured by camera sensors, providing visual representations of indoor and outdoor scenes observed.

2. *"How many instances are there in total (of each type, if appropriate)?"* **A:** The *KITTI-C* dataset contains a total number of $62,730$ RGB images from $18$ corruption types, with a resolution of $192 \times 640$. The *NYUDepth2-C* dataset contains a total number of $39,240$

RGB images from 15 corruption types, with a resolution of $480 \times 640$. The *KITTI-S* dataset contains a total number of $8,364$ RGB images from $12$ style type, with a resolution of $192 \times 640$. Each image in these three datasets is associated with a single-channel depth map of the same size as the image.

3. *"Does the dataset contain all possible instances or is it a sample (not necessarily random) of instances from a larger set?"* **A:** Yes, our datasets contain all possible instances that have been collected so far.

4. *"Is there a label or target associated with each instance?"* **A:** Yes, each instance in our datasets is associated with a single-channel depth map of the same size as the image.

5. *"Is any information missing from individual instances?"* **A:** No.

6. *"Are relationships between individual instances made explicit (e.g., users' movie ratings, social network links)?"* **A:** Yes, the relationship between individual instances is explicit.

7. *"Are there recommended data splits (e.g., training, development/validation, testing)?"* **A:** Yes, we provide detailed data splits for our datasets.

8. *"Is the dataset self-contained, or does it link to or otherwise rely on external resources (e.g., websites, tweets, other datasets)?"* **A:** Yes, our datasets are self-contained.

9. *"Does the dataset contain data that might be considered confidential (e.g., data that is protected by legal privilege or by doctor–patient confidentiality, data that includes the content of individuals' non-public communications)?"* **A:** No, all data are clearly licensed.

10. *"Does the dataset contain data that, if viewed directly, might be offensive, insulting, threatening, or might otherwise cause anxiety?"* **A:** No.

11. *"Any other comments?"* **A:** N/A.

## A.3 Collection Process

In addition to the goals outlined in the previous section, the questions in this section are designed to elicit information that may help researchers and practitioners create alternative datasets with similar characteristics. Again, questions that apply only to datasets that relate to people are grouped together at the end of the section.

1. *"How was the data associated with each instance acquired?"* **A:** Please refer to the details listed in Section B.

2. *"What mechanisms or procedures were used to collect the data (e.g., hardware apparatuses or sensors, manual human curation, software programs, software APIs)?"* **A:** Please refer to the details listed in Section B.

3. *"If the dataset is a sample from a larger set, what was the sampling strategy (e.g., deterministic, probabilistic with specific sampling probabilities)?"* **A:** Please refer to the details listed in Section B.

## A.4 Preprocessing, Cleaning, and Labeling

The questions in this section are intended to provide dataset consumers with the information they need to determine whether the "raw" data has been processed in ways that are compatible with their chosen tasks. For example, text that has been converted into a "bag-of-words" is not suitable for tasks involving word order.

1. *"Was any preprocessing/cleaning/labeling of the data done (e.g., discretization or bucketing, tokenization, part-of-speech tagging, SIFT feature extraction, removal of instances, processing of missing values)?"* **A:** Yes, we preprocessed and cleaned data in our datasets.

2. *"Was the 'raw' data saved in addition to the preprocessed/cleaned/labeled data (e.g., to support unanticipated future uses)?"* **A:** Yes, raw data is accessible.

3. *"Is the software that was used to preprocess/clean/label the data available?"* **A:** Yes, the necessary software used to preprocess and clean the data is publicly available.

4. *"Any other comments?"* **A:** N/A.

### A.5 Uses

The questions in this section are intended to encourage dataset creators to reflect on tasks for which the dataset should and should not be used. By explicitly highlighting these tasks, dataset creators can help dataset consumers make informed decisions, thereby avoiding potential risks or harms.

1. *"Has the dataset been used for any tasks already?"* **A:** No.

2. *"Is there a repository that links to any or all papers or systems that use the dataset?"* **A:** Yes, we provide such links in our GitHub repository.

3. *"What (other) tasks could the dataset be used for?"* **A:** The dataset could be used for relevant perception, tracking, and planning tasks based on camera sensors.

4. *"Is there anything about the composition of the dataset or the way it was collected and preprocessed/cleaned/labeled that might impact future uses?"* **A:** N/A.

5. *"Are there tasks for which the dataset should not be used?"* **A:** N/A.

6. *"Any other comments?"* **A:** N/A.

### A.6 Distribution

Dataset creators should provide answers to these questions prior to distributing the dataset either internally within the entity on behalf of which the dataset was created or externally to third parties.

1. *"Will the dataset be distributed to third parties outside of the entity (e.g., company, institution, organization) on behalf of which the dataset was created?"* **A:** No.

2. *"How will the dataset be distributed (e.g., tarball on website, API, GitHub)?"* **A:** Very likely to be distributed by website, API, and GitHub repository.

3. *"When will the dataset be distributed?"* **A:** The datasets are publicly accessible.

4. *"Will the dataset be distributed under a copyright or other intellectual property (IP) license, and/or under applicable terms of use (ToU)?"* **A:** Yes, the dataset is under the Creative Commons Attribution-NonCommercial-ShareAlike 4.0 International License.

5. *"Have any third parties imposed IP-based or other restrictions on the data associated with the instances?"* **A:** No.

6. *"Do any export controls or other regulatory restrictions apply to the dataset or to individual instances?"* **A:** No.

7. *"Any other comments?"* **A:** N/A.

### A.7 Maintenance

As with the questions in the previous section, dataset creators should provide answers to these questions prior to distributing the dataset. The questions in this section are intended to encourage dataset creators to plan for dataset maintenance and communicate this plan to dataset consumers.

1. *"Who will be supporting/hosting/maintaining the dataset?"* **A:** The authors of this work serve for supporting, hosting, and maintaining the datasets.

2. *"How can the owner/curator/manager of the dataset be contacted (e.g., email address)?"* **A:** The curators can be contacted via emails as follows:
    - Lingdong Kong (`lingdong@comp.nus.edu.sg`);
    - Shaoyuan Xie (`shaoyuanxie@hust.edu.cn`);
    - Hanjiang Hu (`hanjianghu@cmu.edu`);
    - Lai Xing Ng (`ng_lai_xing@i2r.a-star.edu.sg`);
    - Benoit Cottereau (`benoit.cottereau@cnrs.fr`);
    - Wei Tsang Ooi (`ooiwt@comp.nus.edu.sg`).

3. *"Is there an erratum?"* **A:** There is no explicit erratum; updates and known errors will be specified in future versions.

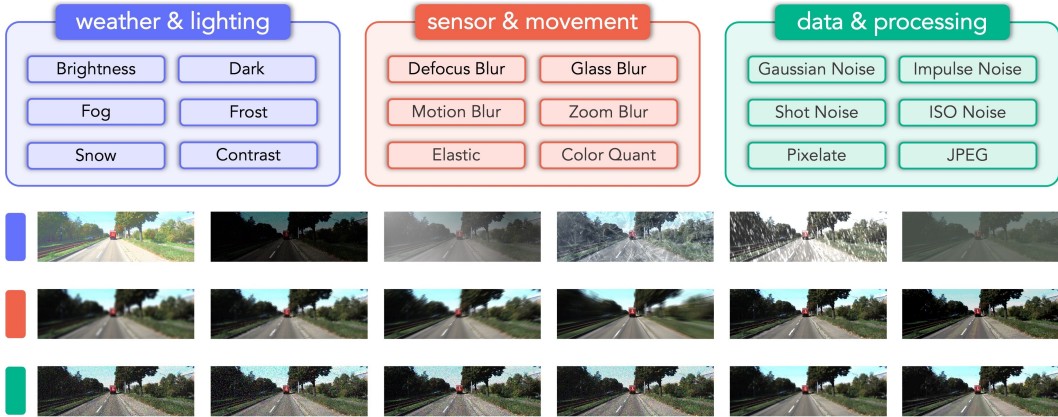

Figure A: The 18 corruption types from three main categories defined in the RoboDepth benchmark. Examples shown are from the proposed *KITTI-C* benchmark.

4. *"Will the dataset be updated (e.g., to correct labeling errors, add new instances, delete instances)?"* **A:** No, for the current version. Future updates (if any) will be posted on the dataset website.

5. *"Will older versions of the dataset continue to be supported/hosted/maintained?"* **A:** Yes. This is the first version of the release; future updates will be posted and older versions will be replaced.

6. *"If others want to extend/augment/build on/contribute to the dataset, is there a mechanism for them to do so?"* **A:** Yes, we provide detailed instructions for future extensions.

7. *"Any other comments?"* **A:** N/A.

## B    Datasets and Benchmarks

In this section, we first provide the detailed definitions of different corruption types in the RoboDepth benchmark (Sec. B.1), we then elaborate on the data collection process (Sec. B.2), the license of our datasets (Sec. B.3), and the procedures for downloading these datasets (Sec. B.4). Lastly, we attach the summary of technical contributions and implementation details for the benchmarked monocular depth estimation methods (Sec. B.5). An overview of the 18 corruption types is shown in Fig. A. The histogram of pixel values under each corruption is shown in Fig. B.

### B.1    Corruption Definition

- **Brightness** refers to the level of lightness or darkness in an image. It is affected by the intensity and color of the light source, as well as the reflectivity of the objects in the scene. Images captured under different lighting conditions may have varying levels of brightness, which can affect the visibility and depth estimation quality.

- **Dark** images can result from low-lighting conditions or underexposure. These images can have low contrast and low visibility, making it difficult to distinguish the objects and backgrounds in the scene. Dark images can also suffer from increased electronic noise and color distortion.

- **Fog** is a type of atmospheric scattering that can reduce the contrast and visibility of objects and backgrounds in a scene. It occurs when water droplets or ice crystals in the air scatter and absorb light, leading to a hazy or blurry appearance.

- **Frost** forms when lenses or windows are coated with ice crystals, leading to image distortion and inaccuracies in depth estimation. Frost can be modeled as a convolution of the image with a non-uniform kernel that depends on the shape and size of the ice crystals. It can affect the accuracy of depth estimation by reducing the contrast of the image and distorting the edges of objects.

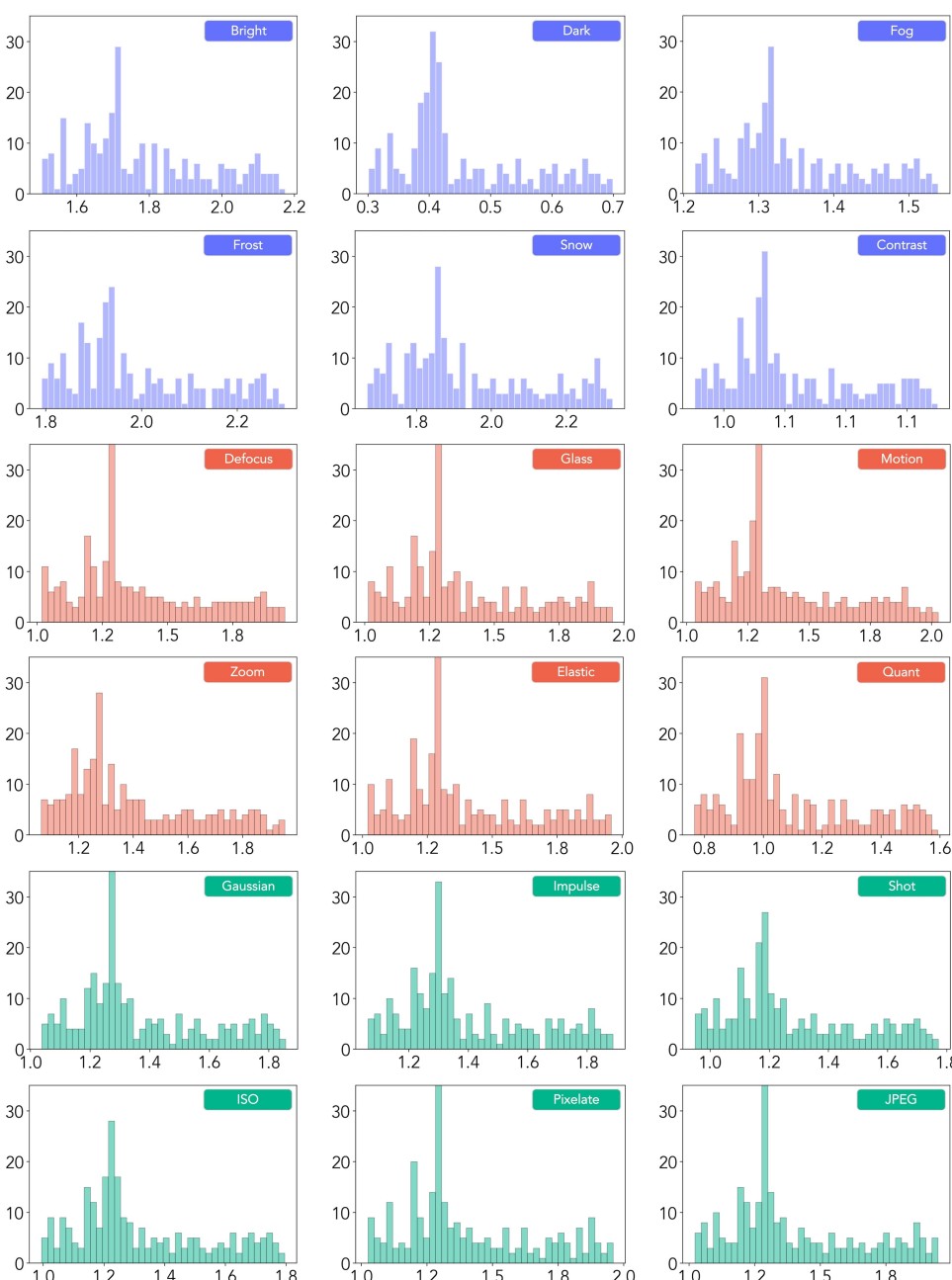

Figure B: The histogram of pixel values under each of the 18 corruption types in *KITTI-C*. Note that the **horizontal axis** of each subfigure is scaled, shifted, and centered to fit the window. The numerical range is at the scale of $1e6$. Zoomed-in for more details.

- **Snow** is a visually obstructive form of precipitation that can obscure objects and backgrounds in the image and make it difficult to estimate accurate depth. It can be modeled as a random distribution of white pixels with a high probability of occurrence. Snow can affect the accuracy of depth estimation by introducing errors in the intensity values of pixels and obscuring the edges of objects.

- **Contrast** is the difference in luminance or color between different parts of an image. High-contrast images have large differences between light and dark areas, while low-contrast images have more similar levels of brightness. Contrast can be affected by lighting conditions and the color and texture of the objects and backgrounds in the scene.

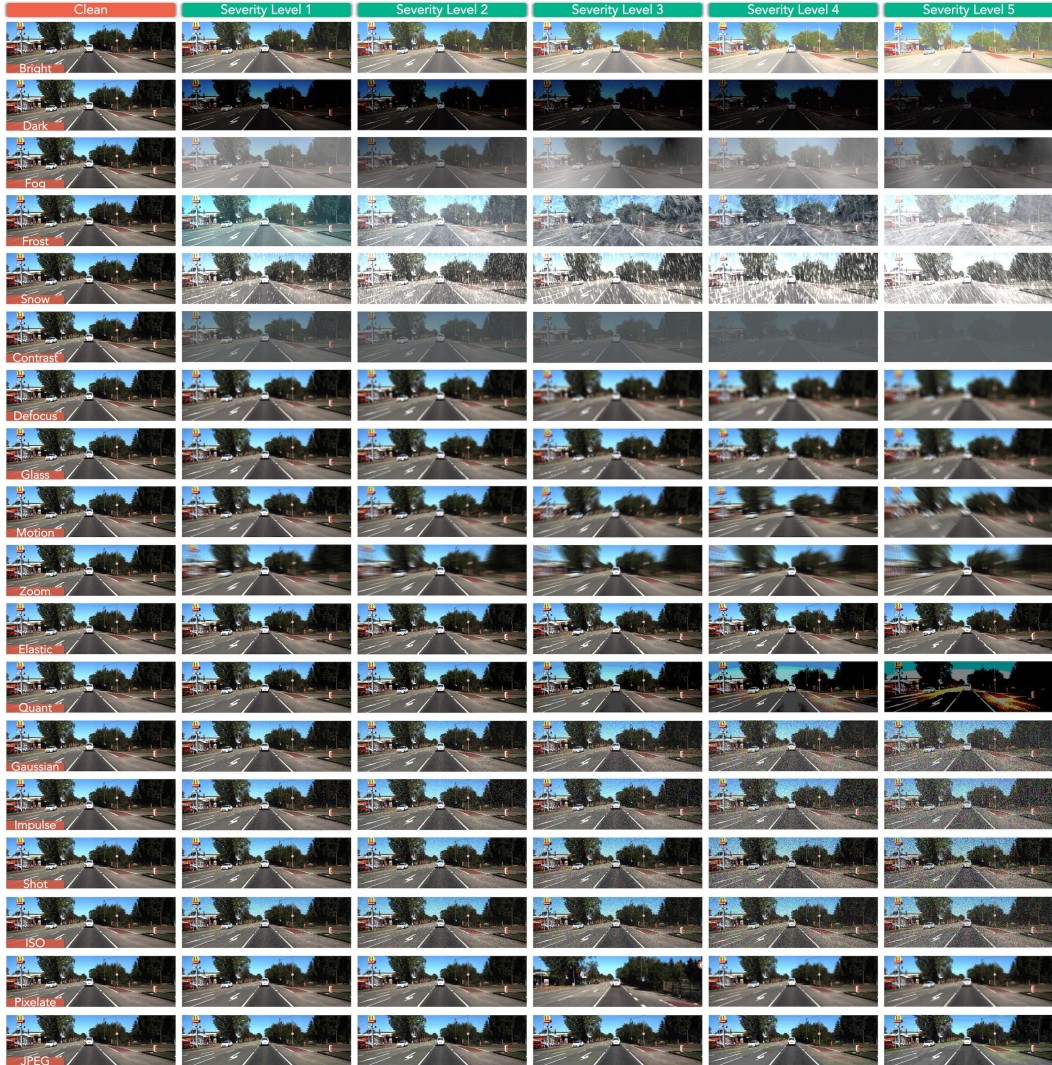

Figure C: Illustrative examples of each corruption type across the five severity levels (level 1 is the lightest and level 5 is the hardest) in the *KITTI-C* benchmark. Best viewed in color. Zoomed-in for more details.

- **Defocus blur** occurs when an image is out of focus, causing objects and backgrounds in the image to appear blurred. It can be modeled as a convolution of the image with a Gaussian kernel, where the blur size depends on the distance between the camera and the object. Defocus blur can affect the accuracy of depth estimation by reducing the contrast of the image and distorting the edges of objects.

- **Glass blur** is a type of image distortion that appears with glass windows or panels. It can lead to inaccuracies in depth estimation due to the opaque and irregular nature of the glass. Glass blur can be modeled as a convolution of the image with a non-uniform kernel that depends on the shape and thickness of the glass.

- **Motion blur** appears when a camera is moving quickly, causing objects and backgrounds in the image to appear blurred. It can be modeled as a convolution of the image with a motion kernel that depends on the direction and speed of the camera movement.

- **Zoom blur** occurs when a camera moves toward an object rapidly, causing the image to appear blurred. It can be modeled as a convolution of the image with a non-uniform kernel that depends on the zoom factor and the distance between the camera and the object. Zoom

blur can affect the accuracy of depth estimation by distorting the edges of objects and reducing the contrast of the image.

- **Elastic transformations** are spatial transformations that deform small regions of an image while preserving the overall structure. They are commonly used in data augmentation techniques for deep learning models to increase the robustness of the model to deformations and variations in the input images.

- **Color quantization** is the process of reducing the number of colors in an image while preserving its overall visual appearance. This technique is commonly used to reduce the storage and processing requirements for digital images. However, this process can result in a loss of detail and color accuracy in the image, particularly in areas with complex color gradients or patterns.

- **Gaussian noise** is a type of noise that appears in low-light conditions and can cause random fluctuations in image intensity. It is modeled as additive white Gaussian noise and has a normal distribution with zero mean and a standard deviation that represents the noise level. Gaussian noise can affect the accuracy of depth estimation by reducing the contrast of the image and introducing errors in intensity values.

- **Impulse noise** is a type of noise that can be caused by bit errors and appears as isolated pixels with incorrect intensity values. It can be modeled as a random distribution of black and white pixels with a low probability of occurrence. It can affect the accuracy of depth estimation by introducing errors in the intensity values of pixels and distorting the image.

- **Shot noise**, also called Poisson noise, is electronic noise caused by the discrete nature of light itself. It occurs when photons hit a sensor and is modeled as a Poisson distribution. Shot noise can lead to irregularities in image intensity and affect the accuracy of depth estimation, especially in low-light conditions.

- **ISO noise** is a type of noise that can appear in digital images captured with high ISO settings. ISO refers to the sensitivity of the camera's sensor to light, with higher ISO values resulting in brighter images. However, increasing the ISO setting can also introduce additional electronic noise into the image, leading to a grainy or speckled appearance. This type of noise can be particularly challenging to remove without losing important image details.

- **Pixelation** occurs when an image is displayed or printed at a low resolution, resulting in individual pixels becoming visible. This can lead to a loss of detail and clarity in the image, bringing extra challenges for depth estimation models.

- **JPEG** is a lossy image compression format commonly used for storing and sharing digital images. JPEG compression reduces the file size of an image by removing some of the image data that is deemed less important or less noticeable to the human eye. Such compression can result in visible artifacts, such as blockiness or blurring, in the compressed image.

### B.2 Corruption Simulation

The corruption simulation tools are from two resources. We use the `imagecorruptions` tool[1] from Michaelis *et al.* [40] to simulate 'brightness', 'fog', 'frost', 'snow', 'contrast', 'defocus blur', 'glass blur', 'motion blur', 'zoom blur', 'elastic transform', 'Gaussian noise', 'impulse noise', 'shot noise', 'pixelate', and 'JPEG compression'. Additionally, We use the `3DCC` tool[2] by Kar *et al.* [24] to simulate 'dark', 'color quantization', and 'ISO noise'.

We follow Hendrycks and Dietterich [18] to split each corruption simulation into five severity levels for *KITTI-C* and four severity levels for *NYUDepth2-C*. The split strategy is the same as the ImageNet-C paper [18]. Illustrative examples of different severity levels in *KITTI-C* are shown in Fig. C. Illustrative examples of different severity levels in *NYUDepth2-C* are shown in Fig. D, Fig. E, and Fig. F.

We provide a comprehensive corruption simulation toolkit that can be used to generate the defined 18 corruption types on any image dataset. This toolkit is publicly accessible at: `https://github.com/ldkong1205/RoboDepth/blob/main/docs/CREATE.md`. Refer to this page for more details and the code implementations of corruption simulations.

---

[1]More details at: `https://github.com/bethgelab/imagecorruptions`.

[2]More details at: `https://github.com/EPFL-VILAB/3DCommonCorruptions`.

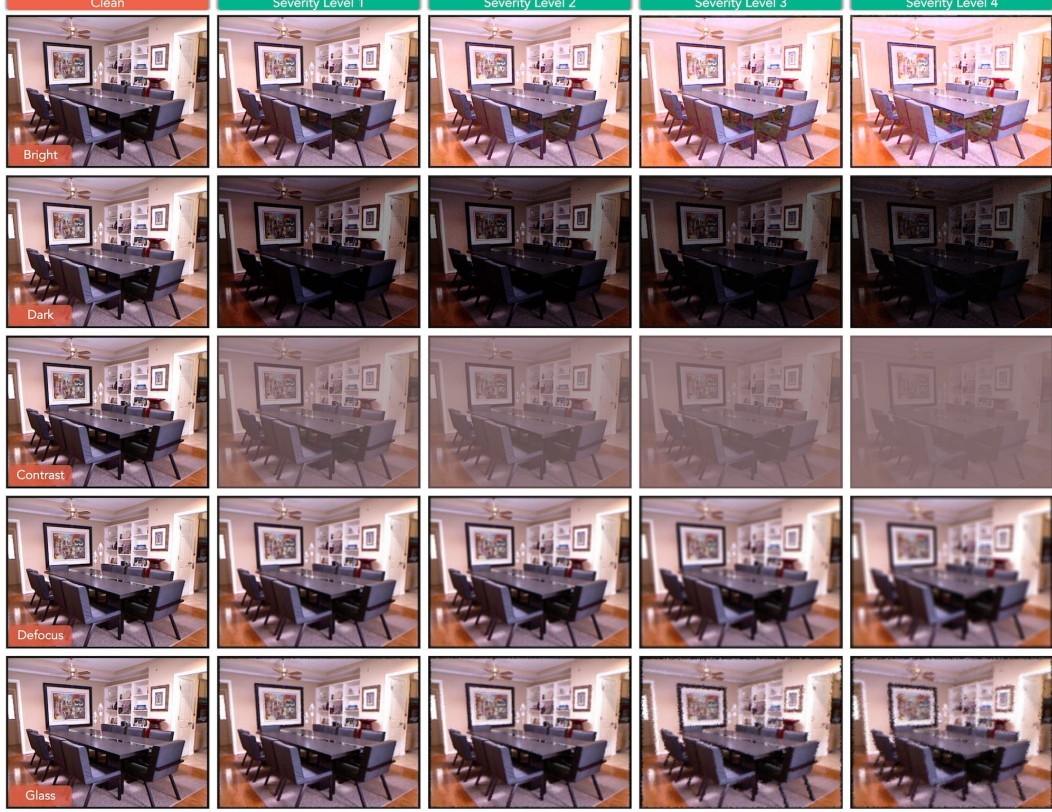

Figure D: Illustrative examples of each of the five corruption types ('brightness', 'dark', 'contrast', 'defocus blur', and 'glass blur') across four severity levels (level 1 is the lightest and level 4 is the hardest) in the *NYUDepth2-C* benchmark. Best viewed in color. Zoomed-in for more details.

To validate the simulated corruptions are of high fidelity, especially for the types that are related to weather and lighting changes, we conduct a study on the pixel distributions. Assuming that a corruption simulation is realistic enough to reflect real-world situations, the distribution of a corrupted "clean" set should be similar to that of the real-world corruption set. We validate this using *ACDC* [45], *nuScenes* [2], *Cityscapes* [8], and *Foggy-Cityscapes* [46], since these datasets contain: *i)* real-world corruption data; and *ii)* clean data collected by the same sensor types from the same physical locations.

We simulate corruptions using "clean" images and compare the distribution patterns with their corresponding real-world corrupted data. We do this to ensure that there is no extra distribution shift from aspects like sensor difference (*e.g.* FOVs and resolutions) and location discrepancy (*e.g.* environmental and semantic changes). The results are shown in this paper: https://github.com/ldkong1205/RoboDepth/blob/main/docs/VALIDITY.md.

What is more, we follow Geirhos *et al.* [15] and use AdaIn [20] to generate stylized images in *KITTI-S*. AdaIn is a classical style transfer model that takes a pair of two images (a reference image and a style image) as input and outputs a stylized image. The checkpoint of this style transfer model can be downloaded at: https://github.com/naoto0804/pytorch-AdaIN. The detailed procedures for generating stylized images are attached at: https://github.com/rgeirhos/Stylized-ImageNet. Illustrative examples of different styles in *KITTI-S* are shown in Fig. G.

### B.3 License

Our datasets and benchmark toolkit are released under the Attribution-NonCommercial-ShareAlike 4.0 International (CC BY-NC-SA 4.0) license[3], under the following terms:

---

[3]More details at: https://creativecommons.org/licenses/by-nc-sa/4.0.

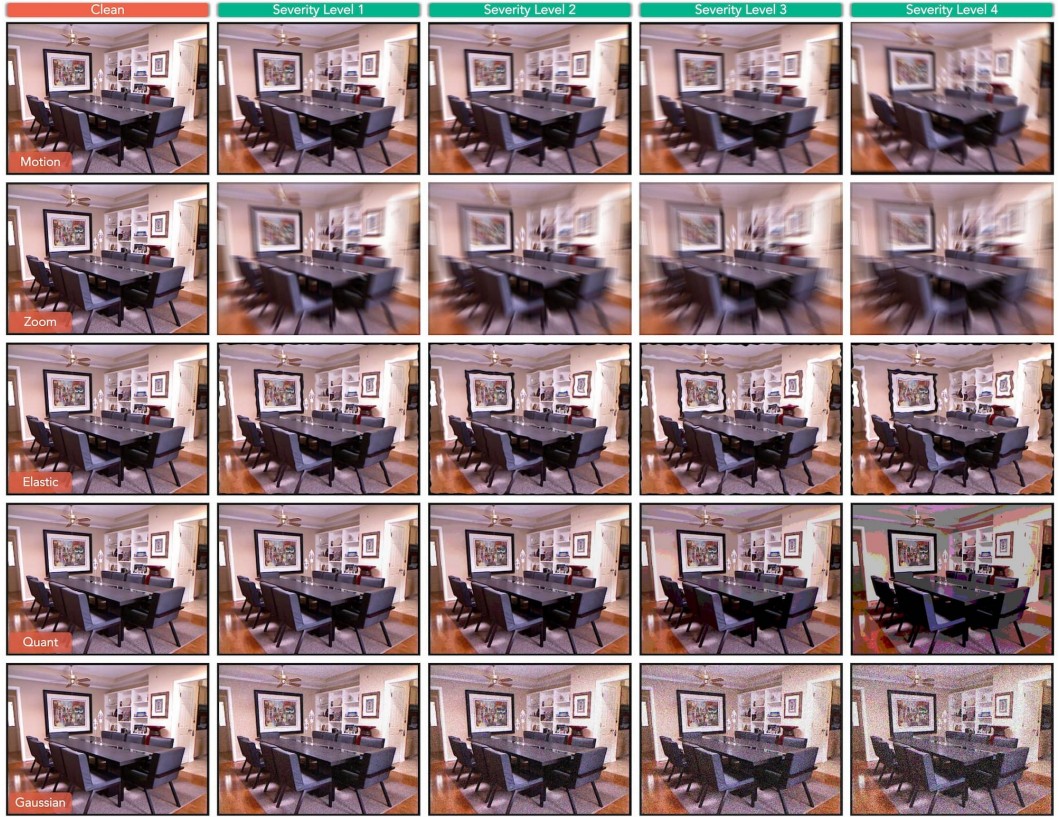

Figure E: Illustrative examples of each of the five corruption types ('motion blur', 'zoom blur', 'elastic transform', 'color quantization', and 'Gaussian noise') across four severity levels (level 1 is the lightest and level 4 is the hardest) in the *NYUDepth2-C* benchmark. Best viewed in color. Zoomed-in for more details.

- **Attribution** — You must give appropriate credit, provide a link to the license, and indicate if changes were made. You may do so in any reasonable manner, but not in any way that suggests the licensor endorses you or your use.

- **NonCommercial** — You may not use the material for commercial purposes.

- **ShareAlike** — If you remix, transform, or build upon the material, you must distribute your contributions under the same license as the original.

## B.4  Download

Our datasets and benchmark toolkit are publicly accessible. We provide detailed procedures for accessing and downloading them. Please refer to the following pages for more details:

- GitHub Repo: `https://github.com/ldkong1205/RoboDepth`.

- Installation & Environment: `https://github.com/ldkong1205/RoboDepth/blob/main/docs/INSTALL.md`.

- Data Preparation: `https://github.com/ldkong1205/RoboDepth/blob/main/docs/DATA_PREPARE.md`.

- Benchmark Details: `https://github.com/ldkong1205/RoboDepth#benchmark`.

- Corruption Simulation Tool: `https://github.com/ldkong1205/RoboDepth/blob/main/docs/CREATE.md`.

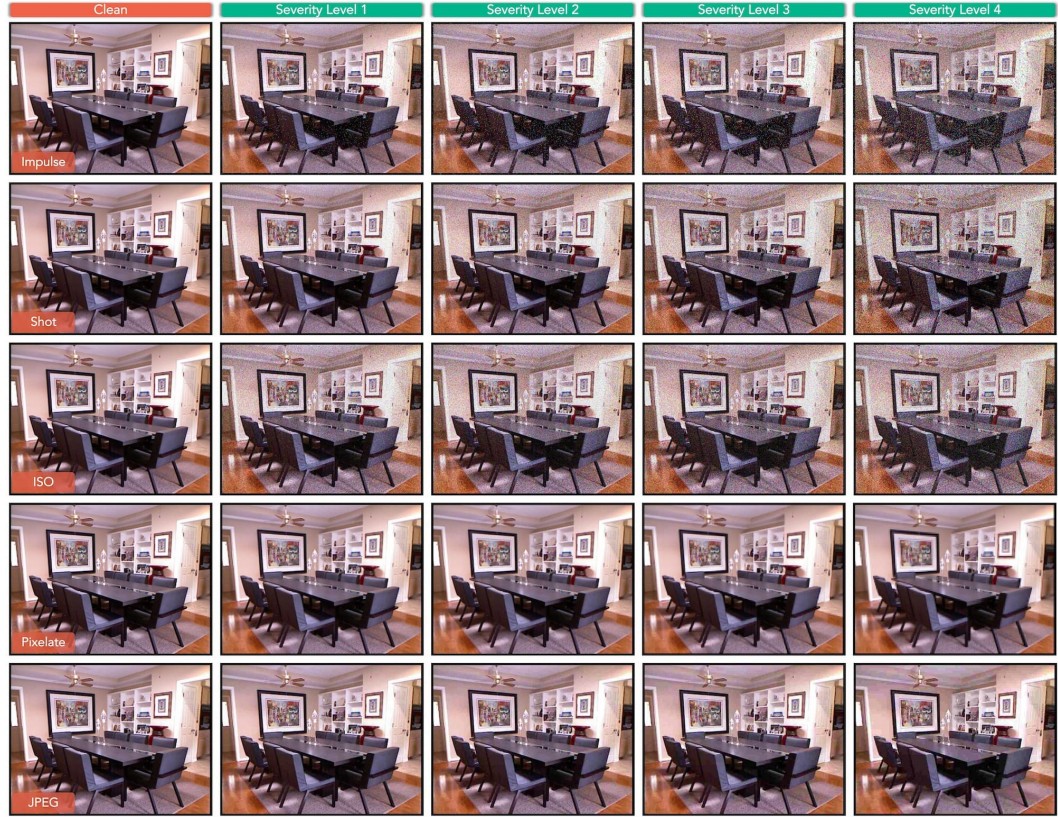

Figure F: Illustrative examples of each of the five corruption types ('impulse noise', 'shot noise', 'ISO noise', 'pixelate', and 'JPEG compression') across four severity levels (level 1 is the lightest and level 4 is the hardest) in the *NYUDepth2-C* benchmark. Best viewed in color. Zoomed-in for more details.

## B.5 Benchmarked Methods

- **MonoDepth2** [17]: A seminar work in the field of MDE which proposed a series of improvements on top of prior works, including a per-pixel minimum re-projection loss, an auto-masking strategy, and a multi-scale estimation framework. We evaluate three versions of this model, under the Mono, Stereo, and Mono+Stereo settings, respectively. We also test this model with different backbones (ResNet-18 and ResNet-50) and pretraining strategies. The code is accessible at: https://github.com/nianticlabs/monodepth2.

- **DepthHints** [57]: A stereo model that makes use of depth hints, which are small but informative cues about the scenes and can be extracted from the image itself. These hints are then used to guide the network during training, allowing the model to learn a better depth estimation representation without requiring any explicit ground truth depth information. The code is accessible at: https://github.com/nianticlabs/depth-hints.

- **MaskOcc** [48]: A monocular model takes into account occlusion in depth estimation. The occlusions occur when objects in the scene block each other from view. The proposed method masks out occluded regions in the input image during training, allowing the model to focus on learning from only the visible parts of the scene. This approach results in more accurate depth estimation, especially in regions with occlusion. The code is accessible at: https://github.com/schelv/monodepth2.

- **DNet** [59]: A monocular model consists of a coarse-to-fine depth estimation pipeline, where a deep neural network is trained to predict absolute depth maps from a single input image. This model is built upon the MonoDepth2 baseline. The code is accessible at: https://github.com/TJ-IPLab/DNet.

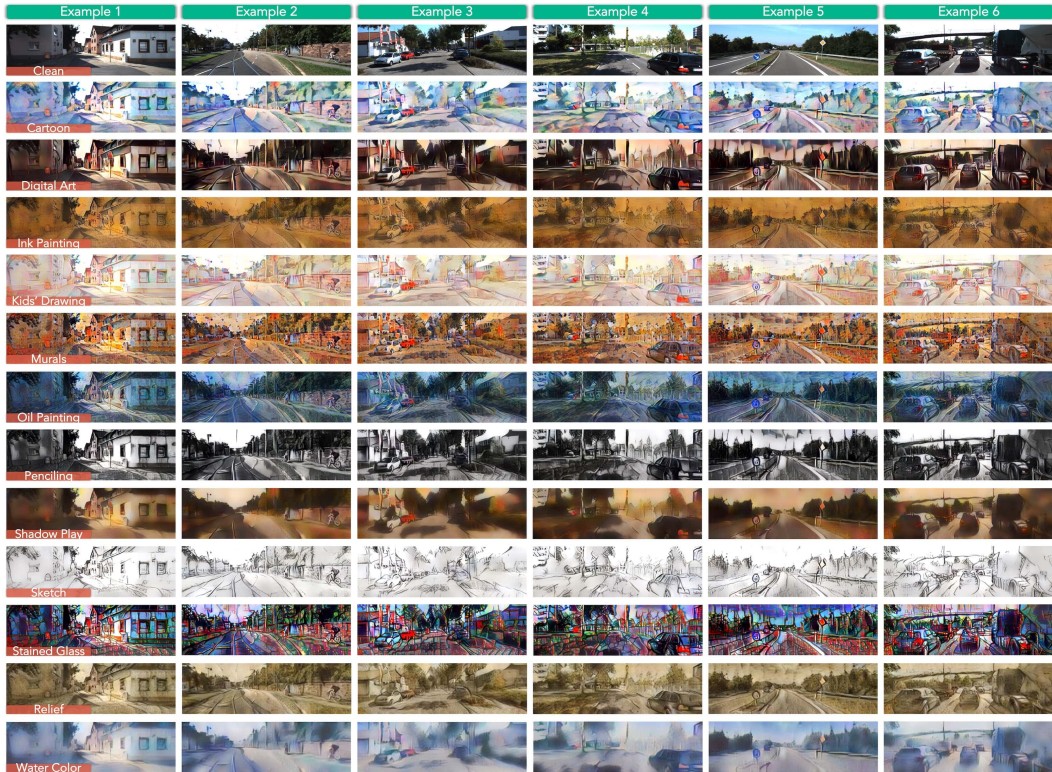

Figure G: Illustrative examples of 12 different styles in the *KITTI-S* benchmark. Best viewed in color. Zoomed-in for more details.

- **CADepth** [60]: A new network architecture that employs attention mechanisms to selectively attend to certain image regions, which helps to improve the accuracy of the depth estimation. The code is accessible at: `https://github.com/kamiLight/CADepth-master`.

- **HR-Depth** [39]: A new depth estimation model that can generate high-resolution depth maps from a single input image. The main contribution of the work is the introduction of a hierarchical residual pyramid network architecture that leverages multi-scale features to produce accurate depth estimates. The code is accessible at: `https://github.com/shawLyu/HR-Depth`.

- **DIFFNet** [67]: A monocular model based on a deep neural network architecture that uses an internal feature fusion mechanism to combine information from different layers of the network. Such a mechanism used in the network helps to capture global and local features of the input images, which improves the accuracy of the depth estimates. The code is accessible at: `https://github.com/brandleyzhou/DIFFNet`.

- **ManyDepth** [56]: A multi-monocular model that leverages temporal consistency between adjacent frames in a video sequence to estimate accurate depth maps. The proposed framework consists of a network that takes as input multiple frames of a video sequence and outputs a corresponding depth map for each frame. The code is accessible at: `https://github.com/nianticlabs/manydepth`.

- **FSRE-Depth** [22]: A monocular model consists of a two-stage network architecture that enhances the representation of the input image by incorporating fine-grained semantic information. The first stage of the network uses a semantic segmentation module to extract semantic information from the input image. The second stage of the network utilizes a depth refinement module to refine the initial depth prediction and generate the final depth estimation. The code is accessible at: `https://github.com/hyBlue/FSRE-Depth`.

- **MonoViT** [66]: A monocular model that consists of a novel training scheme that leverages the spatial and semantic representations learned by the Vision Transformers to predict depth

from a single RGB image. The code is accessible at: `https://github.com/zxcqlf/MonoViT`.

- **DynaDepth** [64]: A monocular model that incorporates the motion dynamics data from an inertial measurement unit (IMU) to improve the accuracy and robustness of the depth estimation in challenging conditions. The proposed method is able to estimate the scale of the scene, which is a major challenge in monocular depth estimation, and is also more robust to illumination changes and dynamic scenes. The code is accessible at: `https://github.com/SenZHANG-GitHub/ekf-imu-depth`.

- **RA-Depth** [42]: A monocular model that can adaptively adjust the output resolution based on the input image resolution. This is achieved through a two-stage architecture that first generates a coarse depth map and then refines it to the desired output resolution using a depth super-resolution network. The code is accessible at: `https://github.com/hmhemu/RA-Depth`.

- **TriDepth** [6]: A monocular model with a new depth estimation loss function that incorporates a scale-invariant gradient term. This allows the model to learn to predict sharper edges and finer details in the depth map, which is critical for accurate depth estimation. The code is accessible at: `https://github.com/xingyuuchen/tri-depth`.

- **Lite-Mono** [63]: A monocular model with a lightweight convolution backbone and transformer architecture for self-supervised monocular depth estimation. The authors propose a hybrid approach that combines the strengths of both CNNs and transformers to achieve state-of-the-art performance on several benchmarks while using significantly fewer parameters compared to existing methods. The code is accessible at: `https://github.com/noahzn/Lite-Mono`.

- **BTS** [31]: A monocular model involves detecting and utilizing local planar structures in the image at multiple scales to improve depth estimation accuracy. Specifically, the authors propose a novel deep neural network architecture that incorporates a local planar guidance module at each scale of a feature pyramid. This module predicts a set of local planar patches in the image and uses them to provide guidance for the depth estimation module. The code is accessible at: `https://github.com/cleinc/bts`.

- **AdaBins** [1]: A monocular model with an adaptive binning scheme that dynamically adjusts the bin sizes of the network to better capture the distribution of depth values in the scene. In traditional binning schemes, fixed bin sizes are used, which can lead to under or over-representation of certain depth values. AdaBins overcomes this limitation by adapting the bin sizes based on the distribution of depth values in the input image. The code is accessible at: `https://github.com/shariqfarooq123/AdaBins`.

- **DPT** [43]: A monocular model that leverages Vision Transformers in place of CNNs as the backbone for dense prediction tasks including MDE. Several new techniques are proposed to facilitate dense predictions using self-attention. The code is accessible at: `https://github.com/isl-org/DPT`.

- **SimIPU** [34]: A multi-modal contrastive learning framework consists of a simple pertaining strategy that leverages the spatial perception module to learn a spatial-aware representation from images and point clouds. The code is accessible at: `https://github.com/zhyever/SimIPU`.

- **DepthFormer** [35]: A monocular model that is tailored to facilitate the long-range correlation for accurate MDE. The backbone is adopted from Vision Transformers, where a hierarchical aggregation and heterogeneous interaction module is proposed to enhance the features via element-wise interaction. The code is accessible at: `https://github.com/zhyever/Monocular-Depth-Estimation-Toolbox/tree/main/configs/depthformer`.

## B.6 Depth Estimation Metrics

In our benchmark, we adopt the conventional reporting of `Abs Rel` (error rate) and $\delta_1$ (accuracy) for measuring the depth estimation performance.

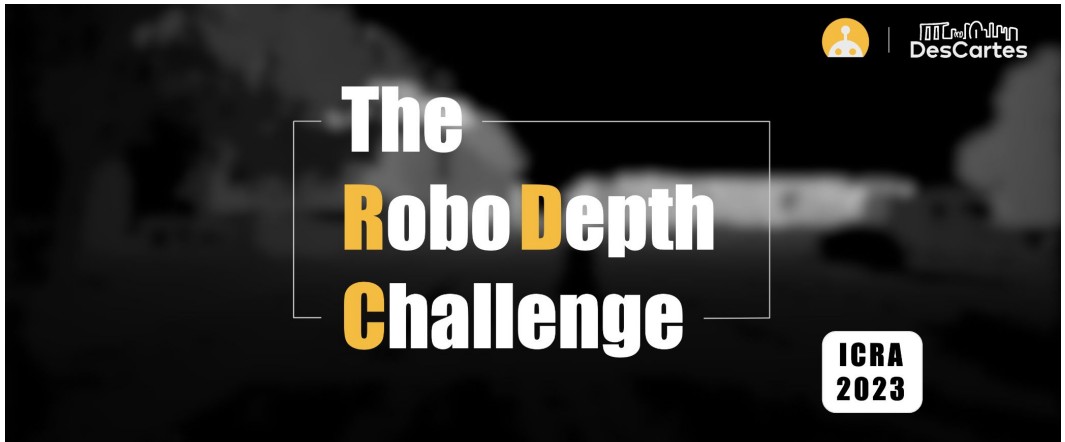

Figure H: We successfully hosted The RoboDepth Challenge [27] at ICRA 2023, which is an academic competition established based on the datasets and benchmarking toolkit proposed in this work. More details of this competition can be found at: https://robodepth.github.io.

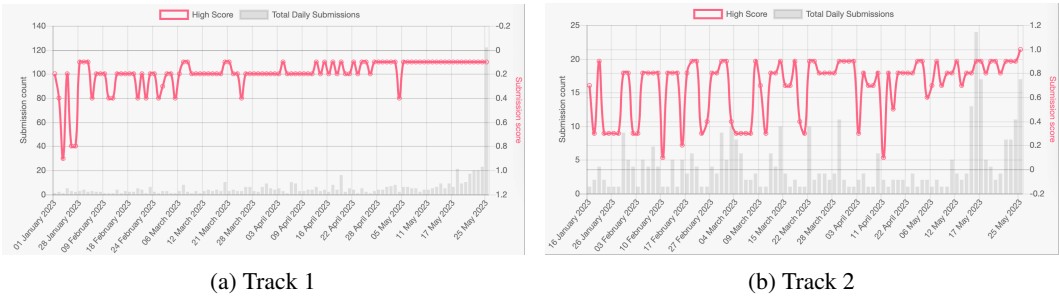

(a) Track 1          (b) Track 2

Figure I: The submission and scoring statistics for the two tracks in the RoboDepth competition.

Abs Rel measures the absolute relative difference between the ground-truth (gt) and the prediction (pred), as calculated via the following equation:

$$\text{Abs Rel} = \frac{1}{|D|} \sum_{pred \in D} \frac{|gt - pred|}{gt} \ . \tag{3}$$

The $\delta$ metric is the depth estimation accuracy given the threshold:

$$\delta_t = \frac{1}{|D|} |\{ \, pred \in D \, | \max \left( \frac{gt}{pred}, \frac{pred}{gt} \right) < 1.25^t \}| \times 100\% \ , \tag{4}$$

where $\delta_1 = \delta < 1.25, \delta_2 = \delta < 1.25^2, \delta_3 = \delta < 1.25^3$ are the three conventionally used accuracy scores in prior works.

We combine Abs Rel and $\delta_1$, the two main measures, into a unified metric as DEE = $\frac{\text{Abs Rel} - \delta_1 + 1}{2}$, which is constantly used as the indicator of depth estimation error in our benchmark.

## C   The RoboDepth Challenge

In this section, we introduce The RoboDepth Challenge – an academic competition that is established based on the datasets and benchmarks proposed in this work.

### C.1   Competition Overview

The RoboDepth Challenge [27] aims to facilitate relevant research in the area of robust monocular depth estimation. It is hosted by the 40th IEEE Conference on Robotics and Automation (ICRA 2023). More materials on this competition are provided in the following links:

Table A: The top-ten performing teams on the leaderboard of Track 1 in the RoboDepth competition.

| Team Name | Abs Rel ↓ | Sq Rel ↓ | RMSE ↓ | log RMSE ↓ | $\delta < 1.25$ ↑ | $\delta < 1.25^2$ ↑ | $\delta < 1.25^3$ ↑ |
|---|---|---|---|---|---|---|---|
| OpenSpaceAI | 0.121 | 0.919 | 4.981 | 0.200 | 0.861 | 0.953 | 0.980 |
| USTC-IAT-United | 0.123 | 0.932 | 4.873 | 0.202 | 0.861 | 0.954 | 0.979 |
| YYQ | 0.123 | 0.885 | 4.983 | 0.201 | 0.848 | 0.950 | 0.979 |
| zs_dlut | 0.124 | 0.899 | 4.938 | 0.203 | 0.852 | 0.950 | 0.979 |
| UMCV | 0.124 | 0.845 | 4.883 | 0.202 | 0.847 | 0.950 | 0.980 |
| THU_ZS | 0.124 | 0.892 | 4.928 | 0.203 | 0.851 | 0.951 | 0.980 |
| THU_Chen | 0.125 | 0.865 | 4.924 | 0.203 | 0.846 | 0.950 | 0.980 |
| seesee | 0.126 | 0.990 | 4.979 | 0.206 | 0.857 | 0.952 | 0.978 |
| namename | 0.126 | 0.994 | 4.950 | 0.204 | 0.860 | 0.953 | 0.979 |
| USTCxNetEaseFuxi | 0.129 | 0.973 | 5.100 | 0.208 | 0.846 | 0.948 | 0.978 |
| MonoDepth2 [17] | 0.221 | 1.988 | 7.117 | 0.312 | 0.654 | 0.859 | 0.938 |

- Competition Page: `https://robodepth.github.io`.
- Competition Report: `https://arxiv.org/abs/2307.15061`.
- Toolkit: `https://github.com/ldkong1205/RoboDepth/tree/main/competition`.
- Workshop Recordings: `https://www.youtube.com/watch?v=mYhdTGiIGCY&list=PLxxrIfcH-qBGZ6x_e1AT2_YnAxiHIKtkB`.
- Server for Track 1: `https://codalab.lisn.upsaclay.fr/competitions/9418`.
- Server for Track 2: `https://codalab.lisn.upsaclay.fr/competitions/9821`.

## C.2 Statistics

The RoboDepth Challenge started on January 1, 2023, and ended on May 25, 2023. There are two tracks in this competition: a self-supervised learning track for robust outdoor depth estimation and a supervised learning track for robust indoor depth estimation.

This competition attracted a lot of participants from around the world. Specifically, **226** teams registered at our evaluation servers. Among them, **66** teams made a total number of **1137** valid submissions. The detailed statistics in terms of submission and scoring are shown in Fig. I. The top-three performing teams of Track 1 achieved the Abs Rel scores of 0.121, 0.123, and 0.123, respectively. The top-three performing teams of Track 2 achieved the a1 scores of 0.940, 0.928, and 0.898. More results are attached to our evaluation servers and competition homepage.

For more details, please refer to our competition page at: `https://github.com/ldkong1205/RoboDepth/tree/main/competition`.

## C.3 Workshop

We hosted The RoboDepth Workshop at ICRA on June 02, 2023. The video recordings of this workshop are publicly available at: `https://www.youtube.com/watch?v=mYhdTGiIGCY&list=PLxxrIfcH-qBGZ6x_e1AT2_YnAxiHIKtkB`.

The slides of this workshop can be downloaded from `https://ldkong.com/talks/icra23_robodepth.pdf`.

## C.4 Leaderboards

The results of the top-ten teams from each of the two tracks are shown in Table A and Table B, respectively. For more details, please refer to our competition page at: `https://github.com/ldkong1205/RoboDepth/tree/main/competition`.

# D  Additional Quantitative Result

In this section, we provide the complete results of different monocular depth estimation models in the established *KITTI-C*, *NYUDepth2-C*, and *KITTI-S* benchmarks.

Table B: The top-ten performing teams on the leaderboard of Track 2 in the RoboDepth competition.

| Team Name | Abs Rel ↓ | Sq Rel ↓ | RMSE ↓ | log RMSE ↓ | $\delta < 1.25$ ↑ | $\delta < 1.25^2$ ↑ | $\delta < 1.25^3$ ↑ |
|---|---|---|---|---|---|---|---|
| USTCxNetEaseFuxi | 0.088 | 0.046 | 0.347 | 0.115 | 0.940 | 0.985 | 0.996 |
| OpenSpaceAI | 0.095 | 0.045 | 0.341 | 0.117 | 0.928 | 0.990 | 0.998 |
| GANCV | 0.104 | 0.060 | 0.391 | 0.131 | 0.898 | 0.982 | 0.995 |
| shinonomei | 0.123 | 0.080 | 0.450 | 0.153 | 0.861 | 0.975 | 0.993 |
| YYQ | 0.125 | 0.085 | 0.470 | 0.159 | 0.851 | 0.970 | 0.989 |
| Hyq | 0.124 | 0.089 | 0.474 | 0.158 | 0.851 | 0.967 | 0.990 |
| DepthSquad | 0.137 | 0.085 | 0.462 | 0.158 | 0.845 | 0.976 | 0.996 |
| kinda | 0.146 | 0.095 | 0.480 | 0.165 | 0.831 | 0.973 | 0.993 |
| dx3 | 0.131 | 0.095 | 0.507 | 0.170 | 0.825 | 0.963 | 0.989 |
| uuht | 0.150 | 0.100 | 0.492 | 0.168 | 0.822 | 0.973 | 0.993 |
| DepthFormer [35] | 0.190 | 0.179 | 0.717 | 0.248 | 0.655 | 0.898 | 0.970 |

## D.1  KITTI-C

The clean performances of 32 MDE models under the standard KITTI Eigen split are shown in Table C.

The per-corruption DEE scores, CE scores, and RR scores of 32 MDE models in the *KITTI-C* benchmark are shown in Table D, Table E, and Table F, respectively.

## D.2  NYUDepth2-C

The per-corruption DEE scores, CE scores, and RR scores of 10 MDE models in the *NYUDepth2-C* benchmark are shown in Table G, Table H, and Table I, respectively.

## D.3  KITTI-S

The per-corruption DEE scores of 32 MDE models in the *KITTI-S* benchmark are shown in Table J.

# E  Additional Qualitative Result

In this section, we provide additional qualitative results of different monocular depth estimation models under out-of-distribution corruption scenarios.

Specifically, the qualitative results of MonoDepth2 [17], Lite-Mono [63], and MonoViT [66] under each of the 18 corruption types across five severity levels in our benchmark are shown in Fig. J, Fig. K, and Fig. L, respectively.

The qualitative results of MonoDepth2 [17], DIFFNet [67], RA-Depth [42], Lite-Mono [63], and MonoViT [66], under each of the 18 corruption types in our benchmark are shown in Fig. M and Fig. N.

The qualitative results of MonoDepth2 [17], DIFFNet [67], RA-Depth [42], Lite-Mono [63], and MonoViT [66], under each of the 12 styles in our benchmark are shown in Fig. O.

Table C: The reproduced scores of 32 monocular depth estimation models under the standard KITTI Eigen split. Blocks from top to bottom: **[1st]** the baseline MonoDepth2 $_{R18}$ [17]; **[2nd]** methods w/ monocular inputs; **[3rd]** methods w/ stereo inputs; **[4th]** methods w/ monocular + stereo inputs. M+S denotes models with both monocular and stereo inputs. The checkpoints for reproduction will be publicly available in our GitHub repository.

| Method | Modality | Backbone | Resolution | Pretrain | Abs Rel ↓ | Sq Rel ↓ | RMSE ↓ | log RMSE ↓ | $\delta < 1.25$ ↑ | $\delta < 1.25^2$ ↑ | $\delta < 1.25^3$ ↑ |
|---|---|---|---|---|---|---|---|---|---|---|---|
| MonoDepth2 $_{R18}$ [17] | Mono | ResNet-18 | 640 x 192 | ImageNet | 0.115 | 0.905 | 4.864 | 0.193 | 0.877 | 0.959 | 0.981 |
| MonoDepth2 $_{R18\ nopt}$ [17] | Mono | ResNet-18 | 640 x 192 | None | 0.132 | 1.042 | 5.138 | 0.210 | 0.845 | 0.948 | 0.977 |
| MonoDepth2 $_{HR}$ [17] | Mono | ResNet-18 | 1024 x 320 | ImageNet | 0.106 | 0.807 | 4.631 | 0.193 | 0.877 | 0.959 | 0.980 |
| MaskOcc $_{R18}$ [48] | Mono | ResNet-18 | 640 x 192 | ImageNet | 0.113 | 0.865 | 4.788 | 0.191 | 0.878 | 0.960 | 0.981 |
| DNet $_{R18}$ [59] | Mono | ResNet-18 | 640 x 192 | ImageNet | 0.114 | 0.867 | 4.819 | 0.191 | 0.877 | 0.960 | 0.981 |
| CADepth [60] | Mono | ResNet-50 | 640 x 192 | ImageNet | 0.107 | 0.803 | 4.592 | 0.183 | 0.890 | 0.963 | 0.983 |
| HR-Depth [39] | Mono | ResNet-18 | 640 x 192 | Cityscapes | 0.109 | 0.792 | 4.632 | 0.185 | 0.884 | 0.962 | 0.983 |
| DIFFNet [67] | Mono | HRNet | 640 x 192 | ImageNet | 0.102 | 0.753 | 4.459 | 0.179 | 0.897 | 0.965 | 0.983 |
| ManyDepth $_{R18}$ [56] | Mono | ResNet-18 | 640 x 192 | ImageNet | 0.118 | 0.894 | 4.765 | 0.192 | 0.871 | 0.959 | 0.982 |
| FSRE-Depth [22] | Mono | ResNet-18 | 640 x 192 | ImageNet | 0.105 | 0.708 | 4.546 | 0.182 | 0.886 | 0.964 | 0.983 |
| MonoViT [66] | Mono | MPViT | 640 x 192 | ImageNet | 0.099 | 0.708 | 4.373 | 0.175 | 0.900 | 0.967 | 0.984 |
| MonoViT $_{HR}$ [66] | Mono | MPViT | 1024 x 320 | ImageNet | 0.096 | 0.713 | 4.292 | 0.172 | 0.908 | 0.968 | 0.984 |
| DynaDepth $_{R18}$ [64] | Mono | ResNet-18 | 640 x 192 | ImageNet | 0.112 | 0.839 | 4.848 | 0.192 | 0.877 | 0.959 | 0.981 |
| DynaDepth $_{R50}$ [64] | Mono | ResNet-50 | 640 x 192 | ImageNet | 0.109 | 0.785 | 4.645 | 0.188 | 0.882 | 0.961 | 0.982 |
| RA-Depth [42] | Mono | ResNet-18 | 640 x 192 | ImageNet | 0.096 | 0.632 | 4.216 | 0.171 | 0.903 | 0.968 | 0.985 |
| TriDepth $_{R18}$ [6] | Mono | ResNet-18 | 640 x 192 | ImageNet | 0.112 | 0.775 | 4.574 | 0.186 | 0.878 | 0.962 | 0.983 |
| Lite-Mono $_{Tiny}$ [63] | Mono | ResNet-18 | 640 x 192 | ImageNet | 0.110 | 0.841 | 4.715 | 0.187 | 0.880 | 0.960 | 0.982 |
| Lite-Mono $_{Small}$ [63] | Mono | ResNet-18 | 640 x 192 | ImageNet | 0.110 | 0.809 | 4.677 | 0.186 | 0.879 | 0.961 | 0.982 |
| Lite-Mono $_{Base}$ [63] | Mono | ResNet-18 | 640 x 192 | ImageNet | 0.107 | 0.766 | 4.560 | 0.183 | 0.886 | 0.963 | 0.983 |
| Lite-Mono $_{Large}$ [63] | Mono | ResNet-18 | 640 x 192 | ImageNet | 0.101 | 0.732 | 4.459 | 0.178 | 0.897 | 0.965 | 0.983 |
| MonoDepth2 $_{R18}$ [17] | Stereo | ResNet-18 | 640 x 192 | ImageNet | 0.110 | 0.877 | 4.967 | 0.209 | 0.864 | 0.948 | 0.975 |
| MonoDepth2 $_{R18\ nopt}$ [17] | Stereo | ResNet-18 | 640 x 192 | None | 0.130 | 1.143 | 5.482 | 0.232 | 0.831 | 0.933 | 0.968 |
| MonoDepth2 $_{HR}$ [17] | Stereo | ResNet-18 | 1024 x 320 | ImageNet | 0.107 | 0.851 | 4.776 | 0.202 | 0.873 | 0.953 | 0.977 |
| DepthHints [57] | Stereo | ResNet-18 | 640 x 192 | ImageNet | 0.104 | 0.791 | 4.655 | 0.191 | 0.878 | 0.959 | 0.981 |
| DepthHints $_{nopt+HR}$ [57] | Stereo | ResNet-18 | 1024 x 320 | None | 0.118 | 0.971 | 5.099 | 0.211 | 0.851 | 0.947 | 0.976 |
| DepthHints $_{HR}$ [57] | Stereo | ResNet-18 | 1024 x 320 | ImageNet | 0.097 | 0.734 | 4.454 | 0.187 | 0.889 | 0.961 | 0.981 |
| MonoDepth2 $_{R18}$ [17] | M+S | ResNet-18 | 640 x 192 | ImageNet | 0.106 | 0.821 | 4.752 | 0.196 | 0.874 | 0.957 | 0.979 |
| MonoDepth2 $_{R18\ nopt}$ [17] | M+S | ResNet-18 | 640 x 192 | None | 0.127 | 1.027 | 5.263 | 0.220 | 0.836 | 0.943 | 0.974 |
| MonoDepth2 $_{HR}$ [17] | M+S | ResNet-18 | 1024 x 320 | ImageNet | 0.106 | 0.807 | 4.631 | 0.193 | 0.877 | 0.959 | 0.980 |
| CADepth [60] | M+S | ResNet-18 | 640 x 192 | ImageNet | 0.102 | 0.788 | 4.645 | 0.192 | 0.881 | 0.958 | 0.980 |
| MonoViT [66] | M+S | MPViT | 640 x 192 | ImageNet | 0.098 | 0.683 | 4.333 | 0.174 | 0.904 | 0.967 | 0.984 |

Table D: The **Depth Estimation Error (DEE)** of 32 monocular depth estimation models in the *KITTI-C* benchmark. Blocks from top to bottom: [1st] the baseline MonoDepth2 R18 [17]; [2nd] methods *w/* monocular inputs; [3rd] methods *w/* stereo inputs; [4th] methods *w/* monocular + stereo inputs. **Bold**: Best in col. Underline: Second best in col. Blue: Best in row. Red: Worst in row.

| Method | mDEE | Bright | Dark | Fog | Frost | Snow | Contr | Defoc | Glass | Motio | Zoom | Elast | Quant | Gaus | Impul | Shot | ISO | Pixel | JPEG |
|---|---|---|---|---|---|---|---|---|---|---|---|---|---|---|---|---|---|---|---|
| MonoDepth2 R18 [17] | .119 | .130 | .280 | .155 | .277 | .511 | .187 | .244 | .242 | .216 | .201 | .129 | .193 | .384 | .389 | .340 | .388 | .145 | .196 |
| MonoDepth2 nopt [17] | .144 | .183 | .343 | .311 | .312 | .399 | .416 | .254 | .232 | .199 | .207 | .148 | .212 | .441 | .452 | .402 | .453 | .153 | .171 |
| MonoDepth2 HR [17] | .144 | .129 | .376 | .155 | .271 | .582 | .214 | .393 | .257 | .230 | .232 | .123 | .215 | .326 | .352 | .317 | .344 | .138 | .198 |
| MonoDepth2 R50 [17] | .117 | .127 | .294 | .155 | .287 | .492 | .233 | .427 | .392 | .277 | .208 | .130 | .198 | .409 | .403 | .368 | .425 | .155 | .211 |
| MaskOcc R18 [48] | .117 | .130 | .285 | .154 | .283 | .492 | .200 | .318 | .295 | .228 | .201 | .129 | .184 | .403 | .410 | .364 | .417 | .143 | .177 |
| DNet R18 [59] | .118 | .128 | .264 | .156 | .317 | .504 | .209 | .348 | .320 | .242 | .215 | .131 | .189 | .362 | .366 | .326 | .357 | .145 | .190 |
| CADepth [60] | .108 | .121 | .300 | .142 | .324 | .529 | .193 | .356 | .347 | .285 | .208 | .121 | .192 | .423 | .433 | .383 | .448 | .144 | .195 |
| HR-Depth [39] | .112 | .121 | .289 | .151 | .279 | .481 | .213 | .356 | .300 | .263 | .224 | .124 | .187 | .363 | .373 | .336 | .374 | .135 | .176 |
| DIFFNet [67] | .102 | .111 | .222 | .131 | .199 | .352 | .161 | .513 | .330 | .280 | .197 | .114 | .165 | .292 | .266 | .255 | .270 | .135 | .202 |
| ManyDepth [56] | .123 | .135 | .274 | .169 | .288 | .479 | .227 | .254 | .279 | .211 | .194 | .134 | .189 | .430 | .450 | .387 | .452 | .147 | .182 |
| FSREDepth [22] | .109 | .128 | .261 | .139 | .237 | .393 | .170 | .291 | .273 | .214 | .185 | .119 | .179 | .400 | .414 | .370 | .407 | .147 | .224 |
| MonoViT [66] | .099 | .106 | .243 | .116 | .213 | .275 | .119 | .180 | .204 | .163 | .179 | .118 | .146 | .310 | .293 | .271 | .290 | .162 | .154 |
| MonoViT HR [66] | .094 | .102 | .238 | .114 | .225 | .269 | .117 | .145 | .171 | .145 | .184 | .108 | .145 | .302 | .277 | .259 | .285 | .135 | .148 |
| DynaDepth R18 [64] | .117 | .128 | .289 | .156 | .289 | .509 | .208 | .501 | .347 | .305 | .207 | .127 | .186 | .379 | .379 | .336 | .379 | .141 | .180 |
| DynaDepth R50 [64] | .113 | .128 | .298 | .152 | .324 | .549 | .201 | .532 | .454 | .318 | .218 | .125 | .197 | .418 | .437 | .382 | .448 | .153 | .216 |
| RA-Depth [42] | .096 | .113 | .314 | .127 | .239 | .413 | .165 | .499 | .368 | .378 | .214 | .122 | .178 | .423 | .403 | .402 | .455 | .175 | .192 |
| TriDepth R18 [6] | .117 | .131 | .300 | .188 | .338 | .498 | .265 | .268 | .301 | .212 | .190 | .126 | .199 | .418 | .438 | .402 | .438 | .142 | .205 |
| Lite-Mono Tiny [63] | .115 | .127 | .257 | .157 | .225 | .354 | .191 | .257 | .248 | .198 | .186 | .127 | .159 | .358 | .342 | .336 | .360 | .147 | .161 |
| Lite-Mono Small [63] | .115 | .127 | .251 | .162 | .251 | .430 | .238 | .353 | .282 | .246 | .204 | .128 | .161 | .350 | .336 | .319 | .356 | .154 | .164 |
| Lite-Mono Base [63] | .110 | .119 | .259 | .144 | .245 | .384 | .177 | .224 | .237 | .221 | .196 | .129 | .175 | .361 | .340 | .334 | .363 | .151 | .165 |
| Lite-Mono Large [63] | .102 | .110 | .227 | .126 | .255 | .433 | .149 | .222 | .225 | .220 | .192 | .121 | .148 | .363 | .348 | .329 | .362 | .160 | .184 |
| MonoDepth2 R18 [17] | .123 | .133 | .348 | .161 | .305 | .515 | .234 | .390 | .332 | .264 | .209 | .135 | .200 | .492 | .509 | .463 | .493 | .144 | .194 |
| MonoDepth2 nopt [17] | .150 | .181 | .422 | .292 | .352 | .435 | .342 | .266 | .232 | .217 | .229 | .156 | .236 | .539 | .564 | .521 | .556 | .164 | .178 |
| MonoDepth2 HR [17] | .117 | .132 | .285 | .167 | .356 | .529 | .238 | .432 | .312 | .279 | .246 | .130 | .206 | .343 | .343 | .322 | .344 | .150 | .209 |
| DepthHints [57] | .113 | .124 | .310 | .137 | .321 | .515 | .164 | .350 | .410 | .263 | .196 | .130 | .192 | .440 | .447 | .412 | .455 | .157 | .192 |
| DepthHints nopt [57] | .134 | .173 | .476 | .301 | .374 | .463 | .393 | .357 | .289 | .241 | .231 | .142 | .247 | .613 | .658 | .599 | .692 | .152 | .191 |
| DepthHints HR [57] | .104 | .122 | .282 | .141 | .317 | .480 | .180 | .459 | .363 | .320 | .262 | .118 | .183 | .397 | .421 | .380 | .424 | .141 | .183 |
| MonoDepth2 R18 [17] | .116 | .127 | .404 | .150 | .295 | .536 | .199 | .447 | .346 | .283 | .204 | .128 | .203 | .577 | .605 | .561 | .629 | .136 | .179 |
| MonoDepth2 nopt [17] | .146 | .193 | .460 | .328 | .421 | .428 | .440 | .228 | .221 | .216 | .230 | .153 | .229 | .570 | .596 | .549 | .606 | .161 | .177 |
| MonoDepth2 HR [17] | .114 | .129 | .376 | .155 | .271 | .582 | .214 | .393 | .257 | .230 | .232 | .123 | .215 | .326 | .352 | .317 | .344 | .138 | .198 |
| CADepth [60] | .110 | .123 | .357 | .137 | .311 | .556 | .169 | .338 | .412 | .260 | .193 | .126 | .186 | .546 | .559 | .524 | .582 | .145 | .192 |
| MonoViT [66] | .098 | .104 | .245 | .122 | .213 | .215 | .131 | .179 | .184 | .161 | .168 | .112 | .147 | .277 | .257 | .242 | .260 | .147 | .144 |

Table E: The **Corruption Error (CE)** of 32 monocular depth estimation models in *KITTI-C*. All scores are given in percentage (%). Blocks from top to bottom: [1st] the baseline MonoDepth2 R18 [17]; [2nd] methods w/ monocular inputs; [3rd] methods w/ stereo inputs; [4th] methods w/ monocular + stereo inputs. **Bold**: Best in col. Underline: Second best in col. Blue : Best in row. Red : Worst in row.

| Method | mCE | Bright | Dark | Fog | Frost | Snow | Contr | Defoc | Glass | Motio | Zoom | Elast | Quant | Gaus | Impul | Shot | ISO | Pixel | JPEG |
|---|---|---|---|---|---|---|---|---|---|---|---|---|---|---|---|---|---|---|---|
| MonoDepth2 R18 [17] | 100.0 | 100.0 | 100.0 | 100.0 | 100.0 | 100.0 | 100.0 | 100.0 | 100.0 | 100.0 | 100.0 | 100.0 | 100.0 | 100.0 | 100.0 | 100.0 | 100.0 | 100.0 | 100.0 |
| MonoDepth2 nopt [17] | 119.8 | 140.8 | 122.5 | 200.7 | 112.6 | 78.1 | 222.5 | 104.1 | 95.9 | 92.1 | 103.0 | 114.7 | 109.8 | 114.8 | 116.2 | 118.2 | 116.8 | 105.5 | 87.2 |
| MonoDepth2 HR [17] | 106.1 | 99.2 | 134.3 | 100.0 | 97.8 | 113.9 | 114.4 | 161.1 | 106.2 | 106.5 | 115.4 | 95.4 | 111.4 | 84.9 | 90.5 | 93.2 | 88.7 | 95.2 | 101.0 |
| MonoDepth2 R50 [17] | 113.4 | 97.7 | 105.0 | 100.0 | 103.6 | 96.3 | 124.6 | 175.0 | 162.0 | 128.2 | 103.5 | 100.8 | 102.6 | 106.5 | 103.6 | 108.2 | 109.5 | 106.9 | 107.7 |
| MaskOcc R18 [48] | 104.1 | 100.0 | 101.8 | 98.7 | 102.2 | 96.3 | 107.0 | 130.3 | 121.9 | 105.6 | 100.0 | 100.0 | 95.3 | 105.0 | 105.4 | 107.1 | 107.5 | 98.6 | 90.3 |
| DNet R18 [59] | 104.7 | 98.5 | 94.3 | 100.7 | 114.4 | 98.6 | 111.8 | 142.6 | 132.2 | 112.0 | 107.0 | 101.6 | 97.9 | 94.3 | 94.1 | 95.9 | 92.0 | 100.0 | 96.9 |
| CADepth [60] | 110.1 | 93.1 | 107.1 | 91.6 | 117.0 | 103.5 | 103.2 | 145.9 | 143.4 | 131.9 | 103.5 | 93.8 | 99.5 | 110.2 | 111.3 | 112.7 | 115.5 | 99.3 | 99.5 |
| HR-Depth [39] | 103.7 | 93.1 | 103.2 | 97.4 | 100.7 | 94.1 | 113.9 | 145.9 | 124.0 | 121.8 | 111.4 | 96.1 | 96.9 | 94.5 | 95.9 | 98.8 | 96.4 | 93.1 | 89.8 |
| DIFFNet [67] | 95.0 | 85.4 | 79.3 | 84.5 | 71.8 | 68.9 | 86.1 | 210.3 | 136.4 | 129.6 | 98.0 | 88.4 | 85.5 | 103.9 | 68.4 | 75.0 | 69.6 | 93.1 | 103.1 |
| ManyDepth R18 [56] | 105.4 | 103.9 | 97.9 | 109.0 | 104.0 | 93.7 | 121.4 | 104.1 | 115.3 | 97.7 | 96.5 | 103.9 | 97.9 | 112.0 | 115.7 | 113.8 | 116.5 | 101.4 | 92.9 |
| FSRE-Depth [22] | 99.1 | 98.5 | 93.2 | 89.7 | 85.6 | 76.9 | 90.9 | 119.3 | 112.8 | 99.1 | 92.0 | 92.3 | 92.8 | 104.2 | 106.4 | 108.8 | 104.9 | 101.4 | 114.3 |
| MonoViT R18 [66] | 79.3 | 81.5 | 86.8 | 74.8 | 76.9 | 53.8 | 63.6 | 73.8 | 84.3 | 75.5 | 89.1 | 91.5 | 75.7 | 80.7 | 75.3 | 79.7 | 74.7 | 111.7 | 78.6 |
| MonoViT HR [66] | **75.0** | 78.5 | 85.0 | 73.6 | 81.2 | 52.6 | 62.6 | 59.4 | 70.7 | 67.1 | 91.5 | 83.7 | 75.1 | 78.7 | 71.2 | 76.2 | 73.5 | 93.1 | 75.5 |
| DynaDepth R18 [64] | 110.4 | 98.5 | 103.2 | 100.7 | 104.3 | 99.6 | 111.2 | 205.3 | 143.4 | 141.2 | 103.0 | 98.5 | 96.4 | 98.7 | 97.4 | 98.8 | 97.7 | 97.2 | 91.8 |
| DynaDepth R50 [64] | 120.0 | 98.5 | 106.4 | 98.1 | 117.0 | 107.4 | 107.5 | 218.0 | 187.6 | 147.2 | 108.5 | 96.9 | 102.1 | 108.9 | 112.3 | 112.4 | 115.5 | 105.5 | 110.2 |
| RA-Depth [42] | 112.7 | 86.9 | 112.1 | 81.9 | 86.3 | 80.8 | 88.2 | 204.5 | 152.1 | 175.0 | 106.5 | 94.6 | 92.2 | 110.2 | 103.6 | 118.2 | 117.3 | 120.7 | 98.0 |
| TriDepth R18 [6] | 109.3 | 100.8 | 107.1 | 121.3 | 122.0 | 97.5 | 141.7 | 109.8 | 124.4 | 98.2 | 94.5 | 97.7 | 103.1 | 108.9 | 112.6 | 111.8 | 112.9 | 97.9 | 104.6 |
| Lite-Mono Tiny [63] | 92.9 | 97.7 | 91.8 | 101.3 | 81.2 | 69.3 | 102.1 | 105.3 | 102.5 | 91.7 | 92.5 | 98.5 | 82.4 | 93.2 | 87.9 | 98.8 | 92.8 | 101.4 | 82.1 |
| Lite-Mono Small [63] | 100.3 | 97.7 | 89.6 | 104.5 | 90.6 | 84.2 | 127.3 | 144.7 | 116.5 | 113.9 | 101.5 | 99.2 | 83.4 | 91.2 | 86.4 | 93.8 | 91.8 | 106.2 | 83.7 |
| Lite-Mono Base [63] | 93.2 | 91.5 | 92.5 | 92.9 | 88.5 | 75.2 | 94.7 | 91.8 | 97.9 | 102.3 | 97.5 | 100.0 | 90.7 | 94.0 | 87.4 | 98.2 | 93.6 | 104.1 | 84.2 |
| Lite-Mono Large [63] | 90.8 | 84.6 | 81.1 | 81.3 | 92.1 | 84.7 | 79.7 | 91.0 | 93.0 | 101.9 | 95.5 | 93.8 | 76.7 | 94.5 | 89.5 | 96.8 | 93.3 | 110.3 | 93.9 |
| MonoDepth2 R18 [17] | 117.7 | 102.3 | 124.3 | 103.9 | 110.1 | 100.8 | 125.1 | 159.8 | 137.2 | 122.2 | 104.0 | 104.7 | 103.6 | 128.1 | 130.9 | 136.2 | 127.1 | 99.3 | 99.0 |
| MonoDepth2 nopt [17] | 129.0 | 139.2 | 150.7 | 188.4 | 127.1 | 85.1 | 182.9 | 109.0 | 95.9 | 100.5 | 113.9 | 120.9 | 122.3 | 140.4 | 145.0 | 153.2 | 143.3 | 113.1 | 90.8 |
| MonoDepth2 HR [17] | 111.5 | 101.5 | 101.8 | 107.7 | 128.5 | 103.5 | 127.3 | 177.1 | 128.9 | 129.2 | 122.4 | 100.8 | 106.7 | 89.3 | 88.2 | 94.7 | 88.7 | 103.5 | 106.6 |
| DepthHints [57] | 111.4 | 95.4 | 110.7 | 88.4 | 115.9 | 100.8 | 87.7 | 143.4 | 169.4 | 121.8 | 97.5 | 100.8 | 99.5 | 114.6 | 114.9 | 121.2 | 117.3 | 108.3 | 98.0 |
| DepthHints nopt [57] | 141.6 | 133.1 | 170.0 | 194.2 | 135.0 | 90.6 | 210.2 | 146.3 | 119.4 | 111.6 | 114.9 | 110.1 | 128.0 | 159.6 | 169.2 | 176.2 | 178.4 | 104.8 | 97.5 |
| DepthHints HR [57] | 112.0 | 93.9 | 100.7 | 91.0 | 114.4 | 93.9 | 96.3 | 188.1 | 150.0 | 148.2 | 130.4 | 91.5 | 94.8 | 103.4 | 108.2 | 111.8 | 109.3 | 97.2 | 93.4 |
| MonoDepth2 R18 [17] | 124.3 | 97.7 | 144.3 | 96.8 | 106.5 | 104.9 | 106.4 | 183.2 | 143.0 | 131.0 | 101.5 | 99.2 | 105.2 | 150.3 | 155.5 | 165.0 | 162.1 | 93.8 | 91.3 |
| MonoDepth2 nopt [17] | 136.3 | 148.5 | 164.3 | 211.6 | 152.0 | 83.8 | 235.3 | 93.4 | 91.3 | 100.0 | 114.4 | 118.6 | 118.7 | 148.4 | 153.2 | 161.5 | 156.2 | 111.0 | 90.3 |
| MonoDepth2 HR [17] | 106.1 | 99.2 | 134.3 | 100.0 | 97.8 | 113.9 | 114.4 | 161.1 | 106.2 | 106.5 | 115.4 | 95.4 | 111.4 | 84.9 | 90.5 | 93.2 | 88.7 | 95.2 | 101.0 |
| CADepth [60] | 118.3 | 94.6 | 127.5 | 88.4 | 112.3 | 108.8 | 90.4 | 138.5 | 170.3 | 120.4 | 96.0 | 97.7 | 96.4 | 142.2 | 143.7 | 154.1 | 150.0 | 100.0 | 98.0 |
| MonoViT [66] | 75.4 | **80.0** | 87.5 | 78.7 | 76.9 | **42.1** | 70.1 | 73.4 | 76.0 | 74.5 | **83.6** | 86.8 | 76.2 | **72.1** | **66.1** | **71.2** | **67.0** | 101.4 | **73.5** |

Table F: The **Resilience Rate (RR)** of 32 monocular depth estimation models in *KITTI-C*. All scores are given in percentage (%). Blocks from top to bottom: **[1st]** the baseline MonoDepth2 R18 [17]; **[2nd]** methods *w/* monocular inputs; **[3rd]** methods *w/* stereo inputs; **[4th]** methods *w/* monocular + stereo inputs. **Bold**: Best in col. Underline: Second best in col. Blue : Best in row. Red : Worst in row.

| Method | mRR | Bright | Dark | Fog | Frost | Snow | Contr | Defoc | Glass | Motio | Zoom | Elast | Quant | Gaus | Impul | Shot | ISO | Pixel | JPEG |
|---|---|---|---|---|---|---|---|---|---|---|---|---|---|---|---|---|---|---|---|
| MonoDepth2 R18 [17] | 84.5 | 98.8 | 81.7 | 95.9 | 82.1 | 55.5 | 92.3 | 85.8 | 86.0 | 89.0 | 90.7 | 98.9 | 91.6 | 69.9 | 69.4 | 74.9 | 69.5 | 97.1 | 91.3 |
| MonoDepth2 R18 [17] | 82.5 | 95.4 | 76.8 | 80.5 | 80.4 | 70.2 | 68.2 | 87.2 | 89.7 | 93.6 | 92.6 | 99.5 | 92.1 | 65.3 | 64.0 | 69.9 | 63.9 | 99.0 | 96.9 |
| MonoDepth2 nopt [17] | 82.4 | 98.3 | 70.4 | 95.4 | 82.3 | 47.2 | 88.7 | 68.5 | 83.9 | 86.9 | 86.7 | 99.0 | 88.6 | 76.1 | 73.1 | 77.1 | 74.0 | 97.3 | 90.5 |
| MonoDepth2 R50 [17] | 80.6 | 98.9 | 80.0 | 95.7 | 80.8 | 57.5 | 86.9 | 64.9 | 68.9 | 81.9 | 89.7 | 98.5 | 90.8 | 66.9 | 67.6 | 71.6 | 65.1 | 95.7 | 89.4 |
| MaskOcc R18 [48] | 83.0 | 98.5 | 81.0 | 95.9 | 81.2 | 57.5 | 90.6 | 77.2 | 79.8 | 87.4 | 90.5 | 98.6 | 92.4 | 67.6 | 66.8 | 72.0 | 66.0 | 97.1 | 93.2 |
| DNet R18 [59] | 83.3 | 98.9 | 83.5 | 95.7 | 77.4 | 56.2 | 89.7 | 73.9 | 77.1 | 85.9 | 89.0 | 98.5 | 92.0 | 72.3 | 71.9 | 76.4 | 72.9 | 96.9 | 91.8 |
| CADepth [60] | 80.1 | 98.5 | 78.5 | 96.2 | 75.8 | 52.8 | 90.5 | 72.2 | 73.2 | 80.2 | 88.8 | 98.5 | 90.6 | 64.7 | 63.6 | 69.2 | 61.9 | 96.0 | 90.3 |
| HR-Depth [39] | 82.9 | 99.0 | 80.1 | 95.6 | 81.2 | 58.5 | 88.6 | 72.5 | 78.8 | 83.0 | 87.4 | 98.7 | 91.6 | 71.7 | 70.6 | 74.8 | 70.5 | 97.4 | 92.8 |
| DIFFNet [67] | 85.4 | 99.0 | 86.6 | 96.8 | 89.2 | 72.2 | 93.4 | 54.2 | 74.6 | 80.2 | 89.4 | 98.7 | 93.0 | 78.8 | 81.7 | 83.0 | 81.3 | 96.3 | 88.9 |
| ManyDepth [56] | 83.1 | 98.6 | 82.8 | 94.8 | 81.2 | 59.4 | 88.1 | 85.1 | 82.2 | 90.0 | 91.9 | 98.8 | 92.5 | 65.0 | 62.7 | 69.9 | 62.5 | 97.3 | 93.3 |
| FSREDepth [22] | 83.9 | 97.9 | 82.9 | 96.6 | 85.6 | 68.1 | 93.2 | 79.6 | 81.6 | 88.2 | 91.5 | 98.9 | 92.1 | 67.3 | 65.8 | 70.7 | 66.6 | 95.7 | 87.1 |
| MonoViT [66] | 89.2 | 99.2 | 84.0 | 98.1 | 87.4 | 80.5 | 97.8 | 91.0 | 88.4 | 92.9 | 91.1 | 97.9 | 94.8 | 76.6 | 78.5 | 80.9 | 78.8 | 93.0 | 93.9 |
| MonoViT HR [66] | 89.7 | 99.1 | 84.1 | 97.8 | 85.5 | 80.7 | 97.5 | 94.4 | 91.5 | 94.4 | 90.1 | 98.5 | 94.4 | 77.0 | 79.8 | 81.8 | 78.9 | 95.5 | 94.0 |
| DynaDepth R18 [64] | 81.5 | 98.8 | 80.5 | 95.6 | 80.5 | 55.6 | 89.7 | 56.5 | 74.0 | 78.7 | 89.8 | 98.9 | 92.2 | 70.3 | 70.3 | 75.2 | 70.3 | 97.3 | 92.9 |
| DynaDepth R50 [64] | 78.0 | 98.3 | 79.1 | 95.6 | 76.2 | 50.9 | 90.1 | 52.8 | 61.6 | 76.9 | 88.2 | 98.7 | 90.5 | 65.6 | 63.5 | 69.7 | 62.2 | 95.5 | 88.4 |
| RA-Depth [42] | 78.8 | 98.1 | 75.9 | 96.6 | 84.2 | 64.9 | 92.4 | 55.4 | 69.9 | 68.8 | 87.0 | 97.1 | 90.9 | 63.8 | 66.0 | 66.2 | 60.3 | 91.3 | 89.4 |
| TriDepth R18 [6] | 81.6 | 98.4 | 79.3 | 92.0 | 75.0 | 56.9 | 83.2 | 82.9 | 79.2 | 89.2 | 91.7 | 99.0 | 90.7 | 65.9 | 63.7 | 70.2 | 63.7 | 97.2 | 90.0 |
| Lite-Mono Tiny [63] | 86.7 | 98.6 | 84.0 | 95.3 | 87.6 | 73.0 | 91.4 | 84.0 | 85.0 | 90.6 | 92.0 | 98.6 | 95.0 | 72.5 | 74.4 | 75.0 | 72.3 | 96.4 | 94.8 |
| Lite-Mono Small [63] | 84.7 | 98.6 | 84.6 | 94.7 | 84.6 | 64.4 | 86.1 | 73.1 | 81.1 | 85.2 | 89.9 | 98.5 | 94.8 | 73.5 | 75.0 | 77.0 | 72.8 | 95.6 | 94.5 |
| Lite-Mono Base [63] | 86.0 | 99.0 | 83.3 | 96.2 | 84.8 | 69.2 | 92.5 | 87.2 | 85.7 | 87.5 | 90.3 | 97.9 | 92.7 | 71.8 | 74.2 | 74.8 | 71.6 | 95.4 | 93.8 |
| Lite-Mono Large [63] | 85.5 | 99.1 | 86.1 | 97.3 | 83.0 | 63.1 | 94.8 | 86.6 | 86.3 | 86.9 | 90.0 | 97.9 | 94.9 | 70.9 | 72.6 | 74.7 | 71.1 | 93.5 | 90.9 |
| MonoDepth2 R18 [17] | 79.1 | 98.9 | 74.3 | 95.7 | 79.3 | 55.3 | 87.3 | 69.6 | 76.2 | 83.9 | 90.2 | 98.6 | 91.2 | 57.9 | 56.0 | 61.2 | 57.8 | 97.6 | 91.9 |
| MonoDepth2 nopt [17] | 79.2 | 96.4 | 68.0 | 83.3 | 76.2 | 66.5 | 77.4 | 86.4 | 90.4 | 92.1 | 90.7 | 99.3 | 89.9 | 54.2 | 51.3 | 56.4 | 52.2 | 98.4 | 96.7 |
| MonoDepth2 HR [17] | 81.6 | 98.3 | 81.0 | 94.3 | 72.9 | 53.3 | 86.3 | 64.3 | 77.9 | 81.7 | 85.4 | 98.5 | 89.9 | 74.4 | 74.4 | 76.8 | 74.3 | 96.3 | 89.6 |
| DepthHints [57] | 80.1 | 98.8 | 77.8 | 97.3 | 76.6 | 54.7 | 94.3 | 73.3 | 66.5 | 83.1 | 90.6 | 98.1 | 91.1 | 63.1 | 62.3 | 66.3 | 61.4 | 95.0 | 91.1 |
| DepthHints nopt [57] | 79.5 | 98.0 | 80.1 | 95.9 | 76.2 | 58.0 | 91.5 | 60.4 | 71.1 | 75.9 | 82.4 | 98.4 | 91.2 | 67.3 | 64.6 | 69.2 | 64.3 | 95.9 | 91.2 |
| DepthHints HR [57] | 73.2 | 95.5 | 60.5 | 80.7 | 72.3 | 62.0 | 70.1 | 74.3 | 82.1 | 87.6 | 88.8 | 99.1 | 87.0 | 44.7 | 39.5 | 46.3 | 35.6 | 97.9 | 93.4 |
| MonoDepth2 R18 [17] | 75.4 | 98.8 | 67.4 | 96.2 | 79.8 | 52.5 | 90.6 | 62.6 | 74.0 | 81.1 | 90.1 | 98.6 | 90.2 | 47.9 | 44.7 | 49.7 | 42.0 | 97.7 | 92.9 |
| MonoDepth2 nopt [17] | 76.7 | 94.5 | 63.2 | 78.7 | 67.8 | 67.0 | 65.6 | 90.4 | 91.2 | 91.8 | 90.2 | 99.2 | 90.3 | 50.4 | 47.3 | 52.8 | 46.1 | 98.2 | 96.4 |
| MonoDepth2 HR [17] | 82.4 | 98.3 | 70.4 | 95.4 | 82.3 | 47.2 | 88.7 | 68.5 | 83.9 | 86.9 | 86.7 | 99.0 | 88.6 | 76.1 | 73.1 | 77.1 | 74.0 | 97.3 | 90.5 |
| CADepth [60] | 76.7 | 98.5 | 72.3 | 97.0 | 77.4 | 49.9 | 93.4 | 74.4 | 66.1 | 83.2 | 90.7 | 98.2 | 91.5 | 51.0 | 49.6 | 53.5 | 47.0 | 96.1 | 90.8 |
| MonoViT [66] | 90.4 | 99.2 | 83.6 | 97.2 | 87.2 | 86.9 | 96.2 | 90.9 | 90.4 | 92.9 | 92.1 | 98.3 | 94.5 | 80.1 | 82.3 | 83.9 | 82.0 | 94.5 | 94.8 |

Table G: The **Depth Estimation Error (DEE)** of 10 monocular depth estimation models in the *NYUDepth2-C* benchmark. **Bold**: Best in col. Underline: Second best in col. Blue : Best in row. Red : Worst in row.

| Method | mDEE | Bright | Dark | Contr | Defoc | Glass | Motio | Zoom | Elast | Quant | Gaus | Impul | Shot | ISO | Pixel | JPEG |
|---|---|---|---|---|---|---|---|---|---|---|---|---|---|---|---|---|
| AdaBins EB5 [1] | .122 | .149 | .269 | .265 | .337 | .262 | .231 | .372 | .182 | .180 | .442 | .512 | .392 | .474 | .139 | .175 |
| BTS R50 [31] | .122 | .149 | .269 | .265 | .337 | .262 | .231 | .372 | .182 | .180 | .442 | .512 | .392 | .474 | .139 | .175 |
| AdaBins R50 [1] | .158 | .179 | .293 | .289 | .339 | .280 | .245 | .390 | .204 | .216 | .458 | .519 | .401 | .481 | .186 | .211 |
| DPT ViT-B [43] | **.136** | **.136** | **.182** | **.180** | **.154** | .166 | **.155** | .232 | **.139** | .165 | **.200** | .213 | **.191** | **.199** | .171 | .174 |
| SimIPU nopt [34] | .372 | .388 | .427 | .448 | .416 | .401 | .400 | **.433** | .381 | **.391** | .465 | .471 | .450 | .461 | .375 | **.378** |
| SimIPU ImageNet [34] | .244 | .269 | .370 | .376 | .377 | .337 | .324 | .422 | .306 | .289 | .445 | .463 | .414 | .449 | .247 | .272 |
| SimIPU KITTI [34] | .312 | .326 | .373 | .406 | .360 | **.333** | .335 | .386 | .316 | **.333** | .432 | .442 | .422 | .443 | **.314** | .322 |
| SimIPU WaymoOpen [34] | .243 | .269 | .348 | .398 | .380 | .327 | .313 | .405 | .256 | .287 | .439 | .461 | .416 | .455 | .246 | .265 |
| DepthFormer SwinT-1k [35] | .125 | .147 | .279 | .235 | .220 | .260 | .191 | .300 | .175 | .192 | .294 | .321 | .289 | .305 | .161 | .179 |
| DepthFormer SwinT-22k [35] | .086 | .099 | .150 | **.123** | .127 | .172 | .119 | .237 | .112 | .119 | .159 | **.156** | .148 | .157 | .101 | .108 |

Table H: The **Corruption Error (CE)** of 10 monocular depth estimation models in the *NYUDepth2-C* benchmark. All scores are given in percentage (%). **Bold**: Best in col. Underline: Second best in col. Blue : Best in row. Red : Worst in row.

| Method | mCE | Bright | Dark | Contr | Defoc | Glass | Motio | Zoom | Elast | Quant | Gaus | Impul | Shot | ISO | Pixel | JPEG |
|---|---|---|---|---|---|---|---|---|---|---|---|---|---|---|---|---|
| AdaBins EB5 [1] | 100.0 | 100.0 | 100.0 | 100.0 | 100.0 | 100.0 | 100.0 | 100.0 | 100.0 | 100.0 | 100.0 | 100.0 | 100.0 | 100.0 | 100.0 | 100.0 |
| BTS R50 [31] | 122.8 | 112.9 | 138.7 | 125.0 | 143.4 | 127.2 | 125.5 | 96.9 | 119.0 | 119.2 | 113.3 | 136.9 | 133.3 | 124.7 | 112.1 | 113.6 |
| AdaBins R50 [1] | 134.7 | 135.6 | 151.0 | 136.3 | 144.3 | 135.9 | 133.2 | 101.6 | 133.3 | 143.1 | 117.4 | 138.8 | 136.4 | 126.6 | 150.0 | 137.0 |
| DPT ViT-B [43] | 83.2 | 102.3 | 93.8 | 84.9 | 65.5 | 80.6 | 84.2 | 60.4 | 90.9 | 109.3 | 51.3 | 57.0 | 65.0 | 52.4 | 137.9 | 113.0 |
| SimIPU nopt [34] | 200.2 | 293.9 | 220.1 | 211.3 | 177.0 | 194.7 | 217.4 | 112.8 | 249.0 | 258.9 | 119.2 | 125.9 | 153.1 | 121.3 | 302.4 | 245.5 |
| SimIPU ImageNet [34] | 163.1 | 203.8 | 190.7 | 177.4 | 160.4 | 163.6 | 176.1 | 109.9 | 200.0 | 191.4 | 114.1 | 123.8 | 140.8 | 118.2 | 199.2 | 176.6 |
| SimIPU KITTI [34] | 173.8 | 247.0 | 192.3 | 191.5 | 153.2 | 161.7 | 182.1 | 100.5 | 206.5 | 220.5 | 110.8 | 118.2 | 143.5 | 116.6 | 253.2 | 209.1 |
| SimIPU WaymoOpen [34] | 159.5 | 203.8 | 179.4 | 187.7 | 161.7 | 158.7 | 170.1 | 105.5 | 167.3 | 190.1 | 112.6 | 123.3 | 141.5 | 119.7 | 198.4 | 172.1 |
| DepthFormer SwinT-1k [35] | 106.3 | 111.4 | 143.8 | 110.9 | 93.6 | 126.2 | 103.8 | 78.1 | 114.4 | 127.2 | 75.4 | 85.8 | 98.3 | 80.3 | 129.8 | 116.2 |
| DepthFormer SwinT-22k [35] | 63.5 | 75.0 | 77.3 | 58.0 | 54.0 | 83.5 | 64.7 | 61.7 | 73.2 | 78.8 | 40.8 | 41.7 | 50.3 | 41.3 | 81.5 | 70.1 |

Table I: The **Resilience Rate (RR)** of 10 monocular depth estimation models in the *NYUDepth2-C* benchmark. All scores are given in percentage (%). **Bold**: Best in col. Underline: Second best in col. Blue : Best in row. Red : Worst in row.

| Method | mRR | Bright | Dark | Contr | Defoc | Glass | Motio | Zoom | Elast | Quant | Gaus | Impul | Shot | ISO | Pixel | JPEG |
|---|---|---|---|---|---|---|---|---|---|---|---|---|---|---|---|---|
| AdaBins EB5 [1] | 85.8 | 97.8 | 90.8 | 88.7 | 86.2 | 89.4 | 91.9 | 69.4 | 95.4 | 95.6 | 68.7 | 70.5 | 79.5 | 69.8 | 98.7 | 95.3 |
| BTS R50 [31] | 80.6 | 96.9 | 83.3 | 83.7 | 75.5 | 84.1 | 87.6 | 71.5 | 93.2 | 93.4 | 63.6 | 55.6 | 69.3 | 59.9 | 98.1 | 94.0 |
| AdaBins R50 [1] | 81.6 | 97.5 | 84.0 | 84.4 | 78.5 | 85.5 | 89.7 | 72.5 | 94.5 | 93.1 | 64.4 | 57.1 | 71.1 | 61.6 | 96.7 | 93.7 |
| DPT ViT-B [43] | 95.3 | 100.0 | 94.7 | 94.9 | 97.9 | 96.5 | 97.8 | 88.9 | 99.7 | 96.6 | 92.6 | 91.1 | 93.6 | 92.7 | 96.0 | 95.6 |
| SimIPU nopt [34] | 92.5 | 97.5 | 91.2 | 87.9 | 93.0 | 95.4 | 95.5 | 90.3 | 98.6 | 97.0 | 85.2 | 84.2 | 87.6 | 85.8 | 99.5 | 99.0 |
| SimIPU ImageNet [34] | 85.0 | 96.7 | 83.3 | 82.5 | 82.4 | 87.7 | 89.4 | 76.5 | 91.8 | 94.1 | 73.4 | 71.0 | 77.5 | 72.9 | 99.6 | 96.3 |
| SimIPU KITTI [34] | 91.6 | 98.0 | 91.1 | 86.3 | 93.0 | 97.0 | 96.7 | 89.2 | 99.4 | 97.0 | 82.6 | 81.1 | 84.0 | 81.0 | 99.7 | 98.6 |
| SimIPU WaymoOpen [34] | 85.7 | 96.6 | 86.1 | 79.5 | 81.9 | 88.9 | 90.8 | 78.6 | 98.3 | 94.2 | 74.1 | 71.2 | 77.2 | 72.0 | 99.6 | 97.1 |
| DepthFormer SwinT-1k [35] | 87.3 | 97.5 | 82.4 | 87.4 | 89.1 | 84.6 | 92.5 | 80.0 | 94.3 | 92.3 | 80.7 | 77.6 | 81.3 | 79.4 | 95.9 | 93.8 |
| DepthFormer SwinT-22k [35] | 94.2 | 98.6 | 93.0 | 96.0 | 95.5 | 90.6 | 96.4 | 83.5 | 97.2 | 96.4 | 92.0 | 92.3 | 93.2 | 92.2 | 98.4 | 97.6 |

Table J: The **Depth Estimation Error (DEE)** of 32 monocular depth estimation models in the *KITTI-S* benchmark. Blocks from top to bottom: **[1st]** the baseline MonoDepth2_R18 [17]; **[2nd]** methods *w/* monocular inputs; **[3rd]** methods *w/* stereo inputs; **[4th]** methods *w/* monocular + stereo inputs. **Bold**: Best in col. Underline: Second best in col.

| Method | mDEE | Cartoon | Digital Art | Ink Paint | Kids' Draw | Murals | Oil Paint | Penciling | Shadow Play | Sketch | Stained Glass | Relief | Water Color |
|---|---|---|---|---|---|---|---|---|---|---|---|---|---|
| MonoDepth2_R18 [17] | .365 | .324 | .434 | .351 | .259 | .326 | .328 | .418 | .388 | .416 | .566 | .255 | .317 |
| MonoDepth2_R18 nopt [17] | .378 | .279 | .289 | .589 | .235 | .412 | .249 | .290 | .589 | **.288** | .596 | .259 | .464 |
| MaskOcc_R18 [48] | .368 | .358 | .356 | .260 | .265 | .358 | .375 | .333 | .314 | .576 | .554 | .288 | .384 |
| DNet_R18 [59] | .400 | .444 | .381 | .293 | .280 | .389 | .336 | .422 | .290 | .608 | .553 | .290 | .516 |
| CADepth [60] | .406 | .446 | .380 | .379 | .250 | .421 | .413 | .470 | .317 | .585 | .543 | .242 | .401 |
| HR-Depth [39] | .324 | .318 | .332 | .238 | .228 | .299 | .306 | .337 | .316 | .388 | .531 | .265 | .331 |
| DIFFNet [67] | .310 | .227 | .351 | **.206** | .206 | .386 | .244 | .276 | .289 | .385 | .502 | .294 | .360 |
| ManyDepth [56] | .323 | .251 | .373 | .212 | .205 | .344 | .300 | .360 | .343 | .331 | .553 | .310 | .300 |
| FSREDepth [22] | .293 | .275 | .277 | .294 | .221 | .310 | .220 | .270 | .301 | .318 | .417 | .261 | .352 |
| MonoViT [66] | .238 | .179 | **.229** | .252 | .196 | .240 | .203 | .237 | .208 | .325 | .356 | **.205** | **.221** |
| DynaDepth_R18 [64] | .371 | .349 | .358 | .291 | .264 | .288 | .293 | .338 | .363 | .492 | .534 | .298 | .585 |
| DynaDepth_R50 [64] | .447 | .502 | .408 | .259 | .287 | .398 | .437 | .492 | .372 | .595 | .565 | .460 | .589 |
| RA-Depth [42] | .365 | .304 | .335 | .354 | .262 | .340 | .310 | .342 | .425 | .372 | .476 | .379 | .475 |
| TriDepth [6] | .379 | .304 | .403 | .256 | .217 | .440 | .352 | .480 | .318 | .517 | .573 | .249 | .436 |
| Lite-Mono_Tiny [63] | .280 | .236 | .310 | .251 | .221 | .254 | .283 | .285 | .232 | .356 | .446 | .237 | .252 |
| Lite-Mono_Small [63] | .303 | .210 | .346 | .218 | .210 | .224 | .238 | .374 | .307 | .445 | .435 | .295 | .336 |
| Lite-Mono_Base [63] | .288 | .234 | .313 | .262 | .218 | .231 | .241 | .291 | .240 | .459 | .476 | .229 | .264 |
| Lite-Mono_Large [63] | .323 | .208 | .435 | .249 | .306 | .348 | .258 | .363 | .210 | .458 | .507 | .263 | .266 |
| MonoDepth2_R18 [17] | .456 | .382 | .483 | .329 | .299 | .499 | .442 | .449 | .388 | .584 | .770 | .363 | .486 |
| MonoDepth2_R18 nopt [17] | .427 | .351 | .334 | .587 | .234 | .440 | .309 | .357 | .475 | .316 | .774 | .305 | .643 |
| MonoDepth2_R18 [17] | .414 | .335 | .444 | .275 | .267 | .431 | .402 | .438 | .367 | .459 | .809 | .299 | .438 |
| MonoDepth2_R18 nopt [17] | .403 | .336 | .350 | .510 | .236 | .445 | .338 | .394 | .457 | .321 | .743 | .264 | .433 |
| CADepth [60] | .401 | .398 | .446 | .265 | .260 | .476 | .423 | .479 | .250 | .615 | .552 | .314 | .336 |
| MonoViT [66] | **.232** | **.173** | .236 | .224 | **.192** | **.208** | **.200** | **.226** | **.205** | .315 | **.338** | .217 | .250 |

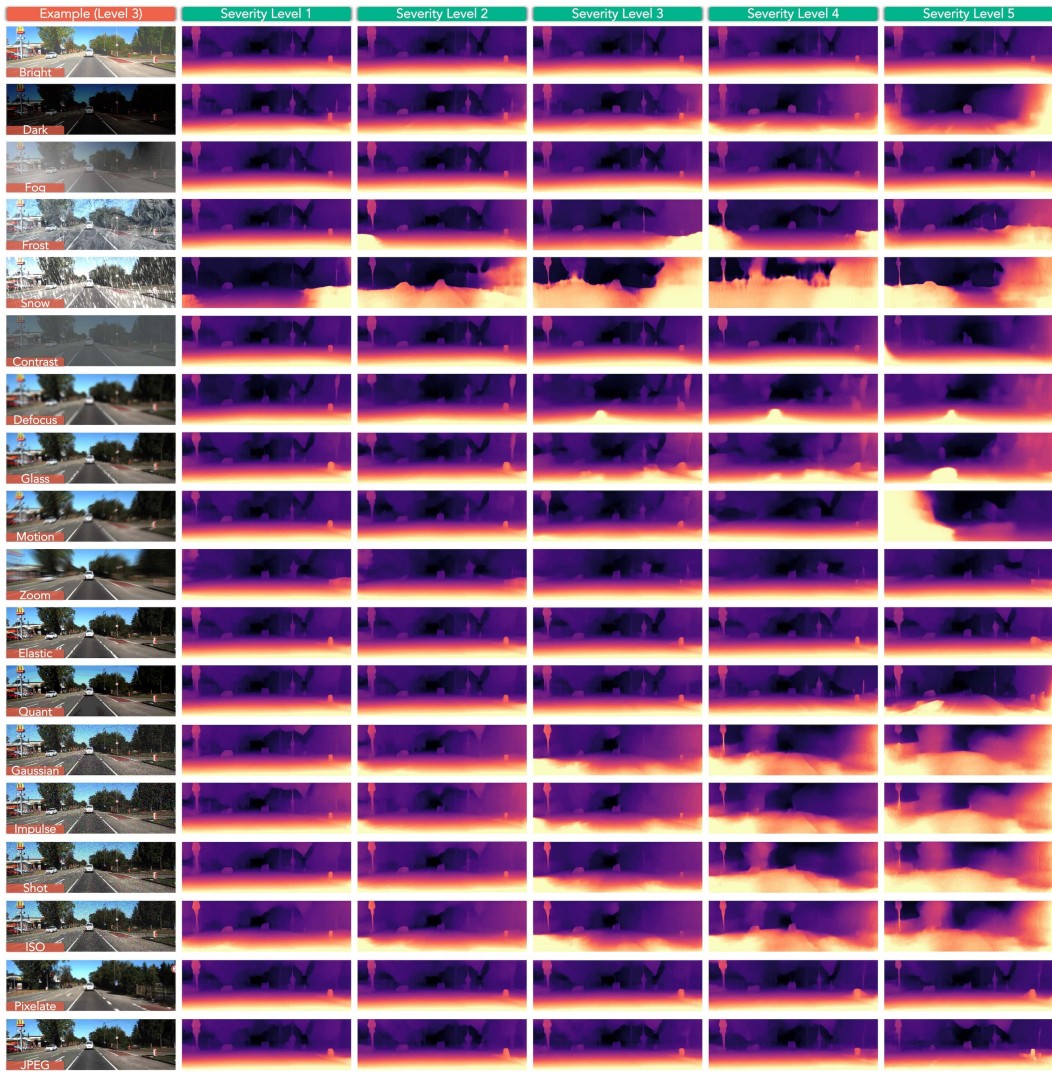

Figure J: Qualitative results of MonoDepth2 [17] under each corruption type across five severity levels. We show examples from the third level in the first column. The lighter regions correspond to near distances and vice versa. Best viewed in color. Zoomed-in for more details.

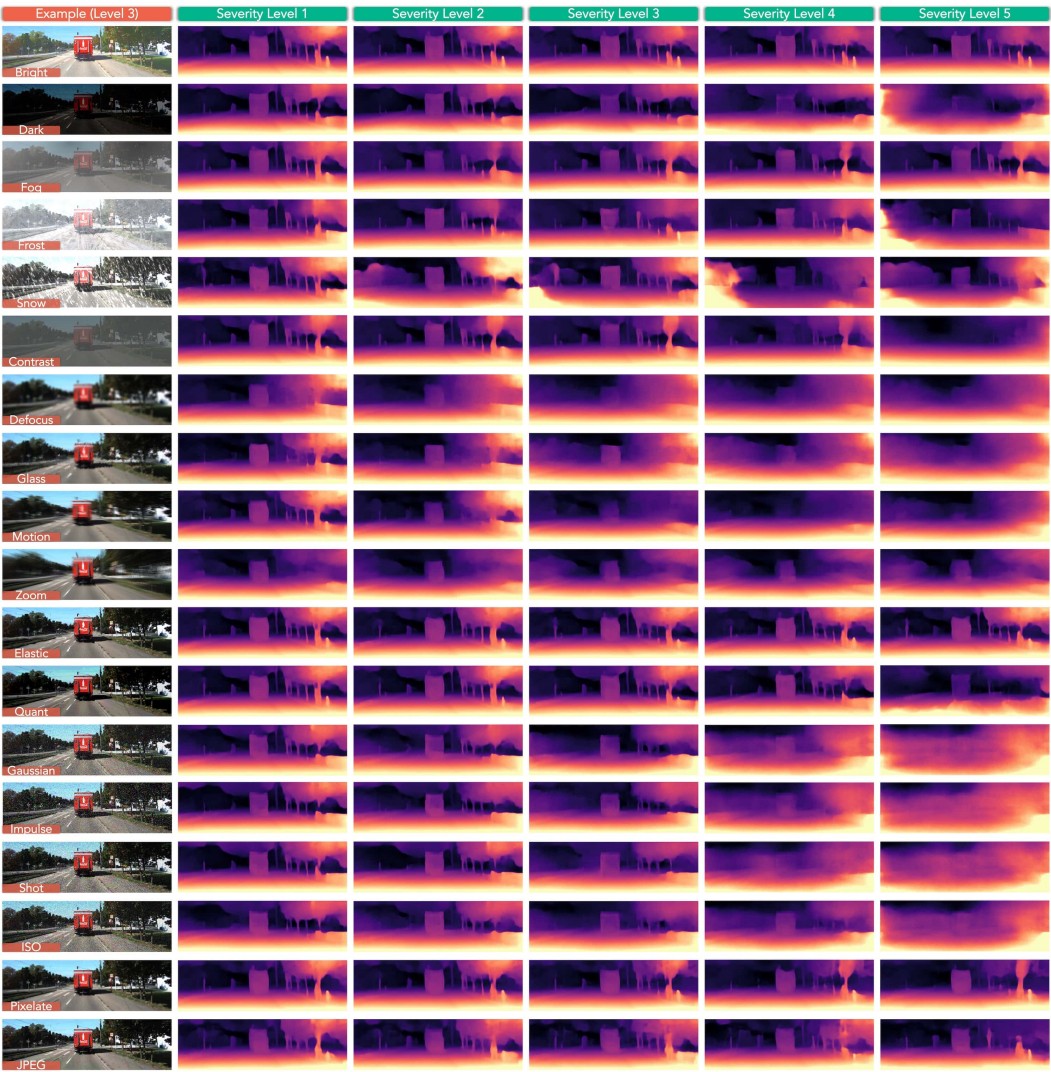

Figure K: Qualitative results of Lite-Mono [63] under each corruption type across five severity levels. We show examples from the third level in the first column. The lighter regions correspond to near distances and vice versa. Best viewed in color. Zoomed-in for more details.

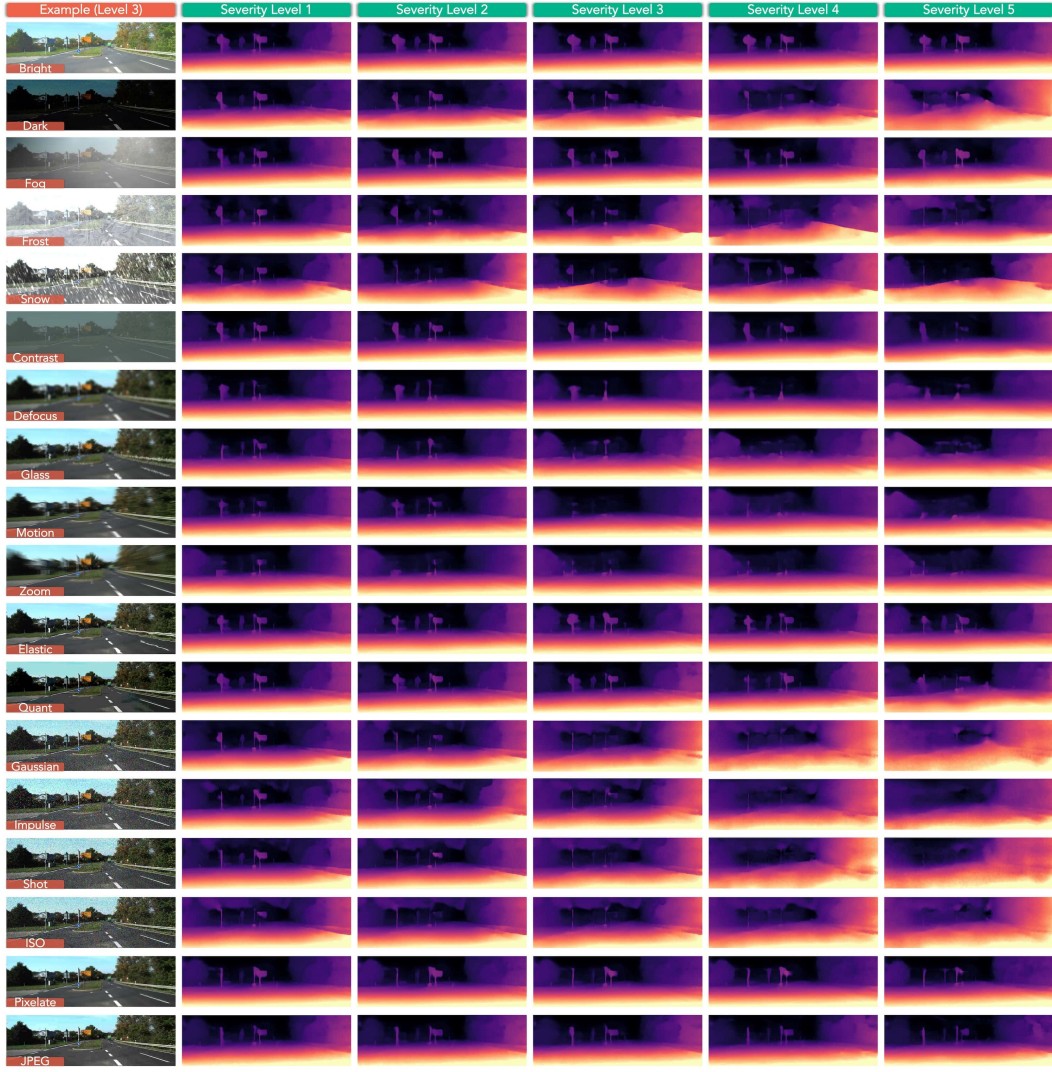

Figure L: Qualitative results of MonoViT [66] under each corruption type across five severity levels. We show examples from the third level in the first column. The lighter regions correspond to near distances and vice versa. Best viewed in color. Zoomed-in for more details.

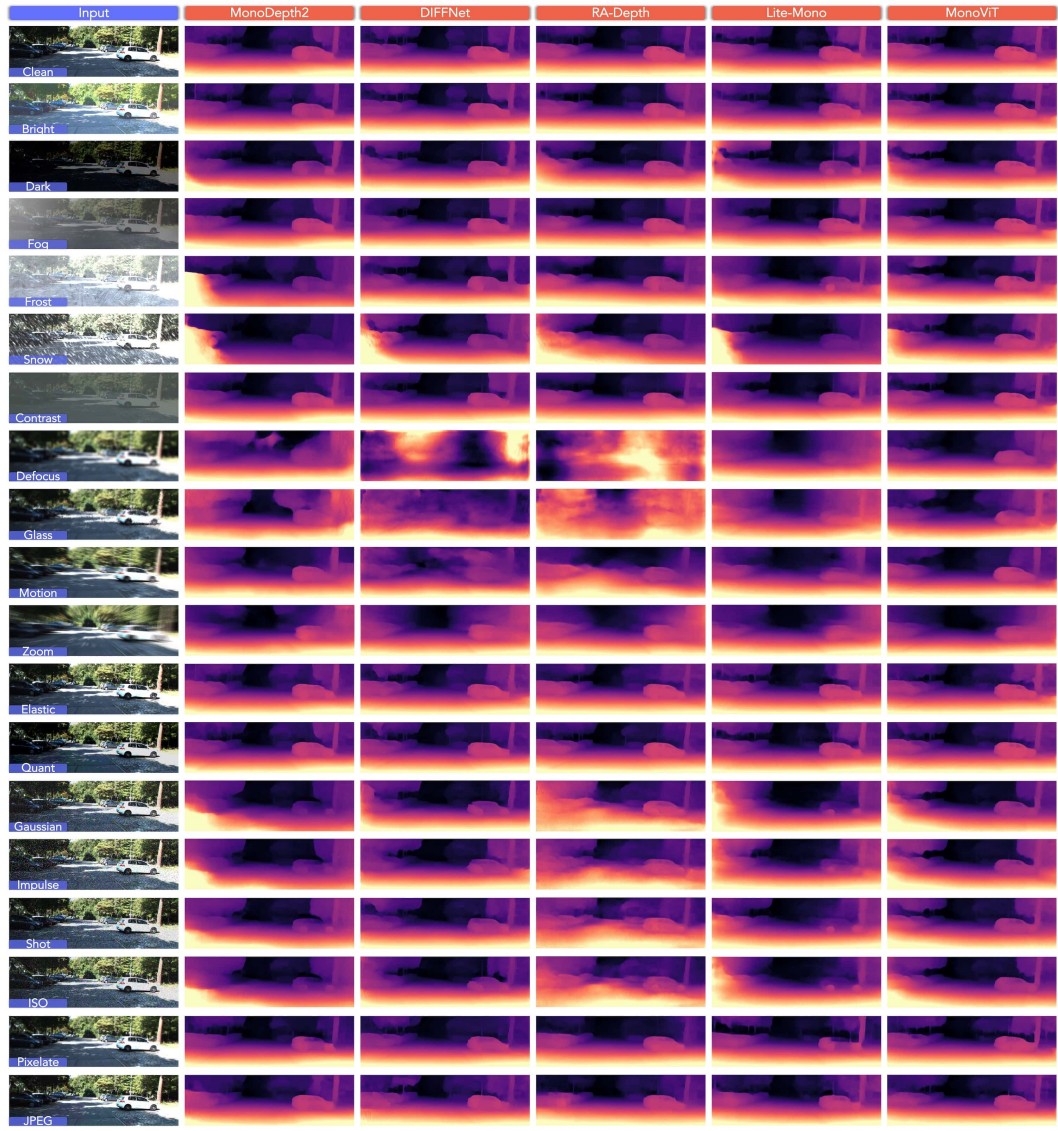

Figure M: Qualitative results of five monocular depth estimation models in the *KITTI-C* benchmark, including MonoDepth2 [17], DIFFNet [67], RA-Depth [42], Lite-Mono [63], and MonoViT [66]. The lighter regions correspond to near distances and vice versa. Best viewed in color. Zoomed-in for more details.

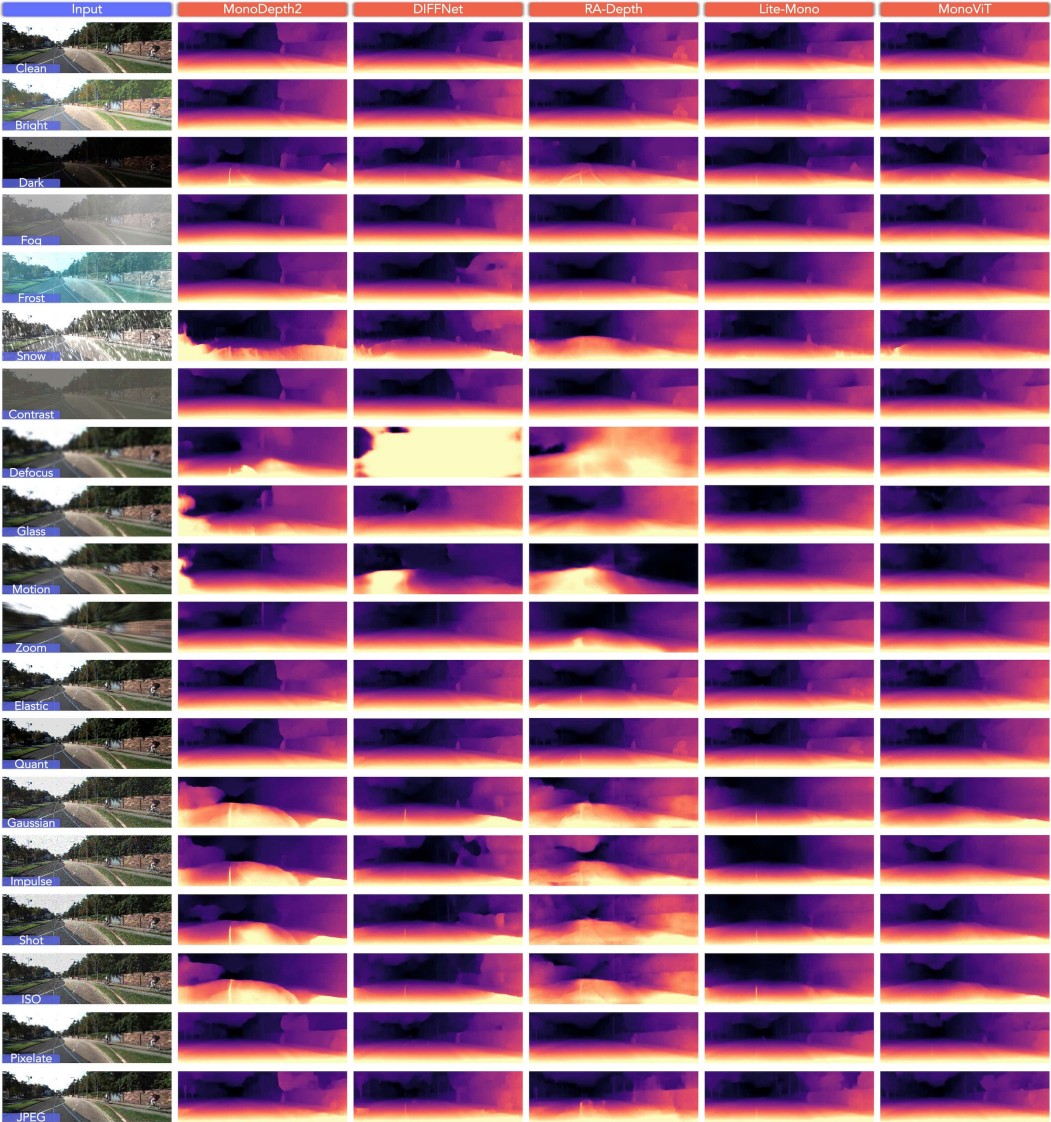

Figure N: Qualitative results of five monocular depth estimation models in the *KITTI-C* benchmark, including MonoDepth2 [17], DIFFNet [67], RA-Depth [42], Lite-Mono [63], and MonoViT [66]. The lighter regions correspond to near distances and vice versa. Best viewed in color. Zoomed-in for more details.

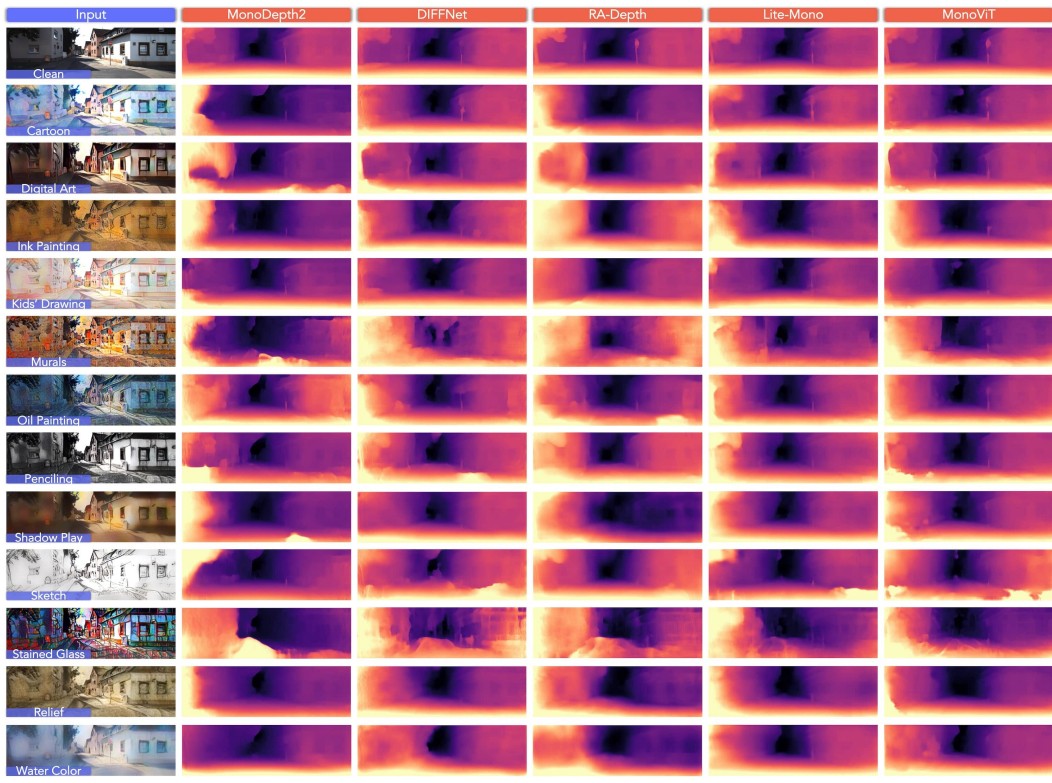

Figure O: Qualitative results of five monocular depth estimation models in the *KITTI-S* benchmark, including MonoDepth2 [17], DIFFNet [67], RA-Depth [42], Lite-Mono [63], and MonoViT [66]. The lighter regions correspond to near distances and vice versa. Best viewed in color. Zoomed-in for more details.

# F  Public Resources Used

In this section, we acknowledge the use of public resources, during the course of this work:

- KITTI Vision Benchmark[4] . . . . . . . . . . . . . . . . . . . . . . . . . . . . . . . . . . . . . . . CC BY-NC-SA 3.0
- NYU Depth Dataset V2[5] . . . . . . . . . . . . . . . . . . . . . . . . . . . . . . . . . . . . . . . . . . . . . . . . . Unknown
- nuScenes[6] . . . . . . . . . . . . . . . . . . . . . . . . . . . . . . . . . . . . . . . . . . . . . . . . . . . . CC BY-NC-SA 4.0
- nuScenes-devkit[7] . . . . . . . . . . . . . . . . . . . . . . . . . . . . . . . . . . . . . . . . . . . . Apache License 2.0
- Cityscapes[8] . . . . . . . . . . . . . . . . . . . . . . . . . . . . . . . . . . . . . . . . . Custom Cityscapes License
- Foggy-Cityscapes[9] . . . . . . . . . . . . . . . . . . . . . . . . . . . . . . Custom Foggy-Cityscapes License
- ACDC[10] . . . . . . . . . . . . . . . . . . . . . . . . . . . . . . . . . . . . . . . . . . . . . . . Custom ACDC License
- SeasonDepth Benchmark Toolkit[11] . . . . . . . . . . . . . . . . . . . . . . . . . . . . . . CC BY-NC-SA 4.0
- Make3D[12] . . . . . . . . . . . . . . . . . . . . . . . . . . . . . . . . . . . . . . . . . . . . Custom Make3D License
- MonoDepth2[13] . . . . . . . . . . . . . . . . . . . . . . . . . . . . . . . . . . . Custom MonoDepth2 License
- DepthHints[14] . . . . . . . . . . . . . . . . . . . . . . . . . . . . . . . . . . . . . . . Custom DepthHints License
- MaskOcc[15] . . . . . . . . . . . . . . . . . . . . . . . . . . . . . . . . . . . . . . . Custom MonoDepth2 License
- DNet[16] . . . . . . . . . . . . . . . . . . . . . . . . . . . . . . . . . . . . . . . . . . Custom MonoDepth2 License
- CADepth[17] . . . . . . . . . . . . . . . . . . . . . . . . . . . . . . . . . . . . . . . . . . . . . . . . . . . . . MIT License
- HR-Depth[18] . . . . . . . . . . . . . . . . . . . . . . . . . . . . . . . . . . . . . . . . . . . . . . . . . . . . MIT License
- DIFFNet[19] . . . . . . . . . . . . . . . . . . . . . . . . . . . . . . . . . . . . . . . . . . . . . . . . . . . . . . Unknown
- ManyDepth[20] . . . . . . . . . . . . . . . . . . . . . . . . . . . . . . . . . . . . . Custom ManyDepth License
- FSRE-Depth[21] . . . . . . . . . . . . . . . . . . . . . . . . . . . . . . . . . . . . . . . . . . . . . . . . . . MIT License
- MonoViT[22] . . . . . . . . . . . . . . . . . . . . . . . . . . . . . . . . . . . . . . . . . . . . . . . . . . . . . MIT License
- DynaDepth[23] . . . . . . . . . . . . . . . . . . . . . . . . . . . . . . . . . . . . . . . . . . . . . . . . . . . . Unknown
- RA-Depth[24] . . . . . . . . . . . . . . . . . . . . . . . . . . . . . . . . . . . . . . . . . . . . . . . . . . . . . Unknown
- TriDepth[25] . . . . . . . . . . . . . . . . . . . . . . . . . . . . . . . . . . . . . GNU General Public License v3.0
- Lite-Mono[26] . . . . . . . . . . . . . . . . . . . . . . . . . . . . . . . . . . . . . . . . . . . . . . . . . . . . . Unknown

---

[4] https://www.cvlibs.net/datasets/kitti.
[5] https://cs.nyu.edu/~silberman/datasets/nyu_depth_v2.html.
[6] https://www.nuscenes.org/nuscenes.
[7] https://github.com/nutonomy/nuscenes-devkit.
[8] https://github.com/mcordts/cityscapesScripts.
[9] https://github.com/sakaridis/fog_simulation-SFSU_synthetic.
[10] https://acdc.vision.ee.ethz.ch.
[11] https://github.com/SeasonDepth/SeasonDepth.
[12] http://make3d.cs.cornell.edu/data.html.
[13] https://github.com/nianticlabs/monodepth2.
[14] https://github.com/nianticlabs/depth-hints.
[15] https://github.com/schelv/monodepth2.
[16] https://github.com/TJ-IPLab/DNet.
[17] https://github.com/kamiLight/CADepth-master.
[18] https://github.com/shawLyu/HR-Depth.
[19] https://github.com/brandleyzhou/DIFFNet.
[20] https://github.com/nianticlabs/manydepth.
[21] https://github.com/hyBlue/FSRE-Depth.
[22] https://github.com/zxcqlf/MonoViT.
[23] https://github.com/SenZHANG-GitHub/ekf-imu-depth.
[24] https://github.com/hmhemu/RA-Depth.
[25] https://github.com/xingyuuchen/tri-depth.
[26] https://github.com/noahzn/Lite-Mono.

- Monocular-Depth-Estimation-Toolbox[27] .......................... Apache License 2.0
- BTS[28] .............................................. GNU General Public License v3.0
- AdaBins[29] ......................................... GNU General Public License v3.0
- DPT[30] ........................................................................ MIT License
- SimIPU[31] .................................................... Apache License 2.0
- DepthFormer[32] ............................................... Apache License 2.0
- ImageCorruptions[33] ........................................... Apache License 2.0
- 3DCC[34] ......................................... Attribution-NC 4.0 International
- ImageNet-C[35] ................................................. Apache License 2.0
- StylizeDatasets[36] ................................................... MIT License
- PyTorch-AdaIN[37] .................................................. MIT License

---

[27] https://github.com/zhyever/Monocular-Depth-Estimation-Toolbox.

[28] https://github.com/cleinc/bts.

[29] https://github.com/shariqfarooq123/AdaBins.

[30] https://github.com/isl-org/DPT.

[31] https://github.com/zhyever/SimIPU.

[32] https://github.com/zhyever/Monocular-Depth-Estimation-Toolbox/tree/main/configs/depthformer.

[33] https://github.com/bethgelab/imagecorruptions.

[34] https://github.com/EPFL-VILAB/3DCommonCorruptions.

[35] https://github.com/hendrycks/robustness.

[36] https://github.com/bethgelab/stylize-datasets.

[37] https://github.com/naoto0804/pytorch-AdaIN.

