# OpenReview forum: "RoboDepth: Robust Out-of-Distribution Depth Estimation under Corruptions"
_NeurIPS.cc/2023/Track/Datasets_and_Benchmarks — NeurIPS 2023 Datasets and Benchmarks Poster_

### Official Review · Reviewer_pgHi · 2023-07-20
**Overall a good study, could be improved in significant ways but so far can be accepted.**

**Rating:** 6
**Confidence:** 4
**Correctness:** Yes.
**Clarity:** Yes, the paper was pleasant to read.

**Strengths:**

Robustness is an important topic in critical applications like robotic navigation. The authors propose a systematic approach to test different methods over a large set of possible corruptions.

**Additional Feedback:**

No additional feedback

**Documentation:**

The code to reproduce the kind of corruptions refereed to in the document is publicly available. Documentation is provided with the code as a README.

The license for the code is missing in the git repository.

**Limitations:**

The corruptions are simulated and thus easy to reproduce during training for future methods that will want to use this benchmark. In my opinion this is not a deal-breaker, but still a significant limitation of the proposed method.

Real data might have bee desirable here, even if it implies creating a dedicated dataset, which might not be feasible.

**Opportunities For Improvement:**

I am not sure the weaknesses can easily be addressed, see limitations for comments.

**Relation To Prior Work:**

Mostly yes, albeit I would be happy if the authors were to clarify the links between the proposed dataset and the public RoboDepth Challence at ICRA this year: https://robodepth.github.io/

Some missing references that might be good to add (but not mandatory):

https://www.sciencedirect.com/science/article/pii/S1077314219301456
https://ieeexplore.ieee.org/abstract/document/8807270

**Summary And Contributions:**

The authors propose a set of corruption filters aiming at simulating real life problems that can hinder monocular depth estimation algorithms.
A large number of monocular depth estimation methods is evaluated on benchmarks created with the proposed simulation procedure.

---

> ### Author Response · Authors · 2023-08-20
> **Authors' Response to Reviewer pgHi**
>
> We thank Reviewer pgHi for devoting time to this review and providing valuable comments. Our responses are as follows.
>
> ---
> > ***Q1:** "The corruptions are simulated and thus easy to reproduce during training for future methods that will want to use this benchmark. In my opinion, this is not a deal-breaker, but still a significant limitation of the proposed method."*
>
> **A:** We thank the reviewer for the comment.
> - The theme of this work is to evaluate depth estimation models in out-of-training-distribution scenarios, where there is a discrepancy (caused by common corruptions) between the training and testing distributions. To ensure fairness in comparisons, all the benchmarked models adopt the same training and testing configurations so that the results are reflecting robustness w.r.t. the model architectures.
> - For future methods that want to use this benchmark: As this task aims to probe the corruption robustness of depth estimation models, future works are also required to adopt the above setting. Note that similar types of benchmarks have already been widely followed, such as ImageNet-C [R1] and many others [R2,R3,R4,R5], and this protocol has been proven effective.
> - We would like to highlight that we adopted this protocol in the RoboDepth Challenge; all the participants followed the exact same training and testing configurations in the competition. We found this protocol simple to apply and it creates a fair comparison environment for different approaches.
>
> ---
> > ***Q2:** "Real data might have been desirable here, even if it implies creating a dedicated dataset, which might not be feasible."*
>
> **A:** We thank the reviewer for the comment.
> - We agree with the reviewer that real data is desirable for a more realistic robustness evaluation. However,  it is often of great difficulty to collect and annotate data in adverse conditions. Besides, there might exist other factors that could impede the robustness comparison, such as sensor (e.g. different FOVs and resolutions) and location (e.g. different cities) discrepancies.
> - Following [R1] and [R3], we generate corruption simulations to build the robustness evaluation sets. These datasets are of large scale, contain diverse corruption types, and are not affected by external discrepancies.
> - We verified the validity of our corruption simulations through two types of experiments, please kindly refer to the response to Reviewer j7EB's Q1 for more details.
> - What is more, we have added some real-world datasets (i.e. [R5] and [R6]) into our benchmark for a better and more comprehensive evaluation. We will include more real corruption datasets in RoboDepth.
>
> ---
> > ***Q3:** "I would be happy if the authors were to clarify the links between the proposed dataset and the public RoboDepth Challenge at ICRA this year: https://robodepth.github.io."*
>
> **A:** Thanks for mentioning.
> - The RoboDepth Challenge is an academic competition designed to facilitate and advance robust depth estimation under corruptions. it was developed based on the benchmarks and datasets in RoboDepth. The baseline models of this competition were set according to our observations and analyses.
> - Specifically, we established two tracks in this competition, one for robust self-supervised depth estimation and another for robust fully-supervised depth estimation. This competition attracted more than two hundred teams on the challenge servers. Among them, 66 teams made a total of 1137 valid submissions.
> - We summarize the details of this competition and the winning solutions in this [technical report](https://arxiv.org/pdf/2307.15061.pdf). Diverse designs and techniques that are proven useful in improving the robustness of depth estimation models under common corruptions have been proposed. We hope that this kind of pursuit can shed light on the development of depth estimation systems that are reliable, scalable, and generalizable.
>
> ---
> > ***Q4:** "Some missing references that might be good to add (but not mandatory): [R7] and [R8]."*
>
> **A:** Thanks for your suggestion. We found the mentioned references relevant and have added them to the revised manuscript.
>
> ---
> **References:**
> - [R1] “Benchmarking neural network robustness to common corruptions and perturbations,” ICLR, 2019.
> - [R2] “Benchmarking the robustness of semantic segmentation models,” CVPR, 2020.
> - [R3] “Benchmarking robustness in object detection: Autonomous driving when winter is coming,” NeurIPS Workshop, 2020.
> - [R4] “Benchmarking the robustness of spatial-temporal models against corruptions,” NeurIPS, 2021.
> - [R5] “nuScenes: A multimodal dataset for autonomous driving,” CVPR, 2020.
> - [R6] “Semantic foggy scene understanding with synthetic data,” IJCV, 2018.
> - [R7] “On the benefit of adversarial training for monocular depth estimation,” CVIU, 2020.
> - [R8] “Semi-supervised adversarial monocular depth estimation,” IEEE TPAMI, 2020.
>
> ---
> Last but not least, we thank Reviewer pgHi again for the time and effort devoted to this review.

---

> > ### Comment · Reviewer_pgHi · 2023-08-29
> > **Following the authors' response**
> >
> > I would like to thanks the authors for addressing the reviewers comments.
> >
> > Overall, the major concerns I has have been addressed. I think that the paper can be considered for acceptance in the conference. I will keep my rating for the moment, but depending on the reviewers/AC discussion phase I might consider increasing it.

---

> > > ### Author Response · Authors · 2023-08-29
> > > **Authors' Response to Reviewer pgHi**
> > >
> > > We sincerely thank Reviewer pgHi for participating in the Author-Reviewer discussion session and providing positive feedback.
> > >
> > > ---
> > > Once again, we thank you for the time and effort devoted and the valuable comments drawn during this review.
> > >
> > > Best regards,
> > >
> > > The Authors

---

### Official Review · Reviewer_76NB · 2023-07-20
**A Comprehensive Benchmark for Robust Out-of-Distribution Depth Estimation**

**Rating:** 9
**Confidence:** 4
**Correctness:** Correct.
**Clarity:** Clearly written

**Strengths:**

The paper introduces the RoboDepth OoD cases, which comprise 18 different corruptions covering various aspects of real-world scenarios. This comprehensive test suite allows for a thorough evaluation of depth estimation models. The dataset collections are comprehensive and meaningful

Evaluating depth estimation models under out-of-distribution OoD scenarios, such as weather and lighting conditions, sensor failure, and data processing issues, aligns well with real-world applications, especially for autonomous driving.

The benchmarking is comprehensive: it  provides a holistic view of the current state-of-the-art 42 existing depth estimation models.

The paper's detailed description of the RoboDepth test suite and the evaluation process ensures that the experiments can be easily reproduced by other researchers.

The project is well-documented and maintained.

**Additional Feedback:**

Good paper. I suggest high-score acceptance

**Documentation:**

We project is well-documented.

**Ethics:**

No ethical issues.

**Limitations:**

The authors adequately addressed the limitations.

**Opportunities For Improvement:**

 The paper mainly focuses on benchmarking existing models and discussing design considerations for improving robustness. How does this benchmark throw light on how to improve Monocular depth estimation for future work?

While the focus on autonomous driving and safety-critical applications is valuable, how is this work's potential applicability of depth estimation in other domains or industries?



**Relation To Prior Work:**

This work is original and unique.

**Summary And Contributions:**

The authors of this study aim to improve the accuracy and reliability of depth estimation models by focusing on out-of-distribution scenarios, which can occur commonly in practical settings such as autonomous driving. To achieve this goal, they have created a new dataset called RoboDepth, consisting of 18 types of disturbances categorized into four groups: weather/lighting conditions, sensor failures or movements, and data processing issues. Using RoboDepth, they evaluated 42 existing depth estimation models and found that many of them were not robust enough due to the lack of appropriate testing methods. In conclusion, the authors suggest several strategies for improving the robustness of these models and highlight the importance of evaluating their performance under diverse operating conditions.

---

> ### Author Response · Authors · 2023-08-20
> **Authors' Response to Reviewer 76NB**
>
> We thank Reviewer 76NB for devoting time to this review and providing valuable comments. Our responses are as follows.
>
> ---
> > ***Q1:** "How does this benchmark throw light on how to improve monocular depth estimation for future work?"*
>
> **A:** We thank the reviewer for the comment.
> - In this work, we focus on benchmarking and analyzing the robustness of depth estimation under corruptions. We observe that the modality (monocular, stereo, and a combination of both), pretraining strategies, and training resolutions are important factors that affect the models’ robustness. We also analyzed the model capacity, complexity, and corruption augmentation on existing models. Our study indicates that a mid-size monocular model with larger receptive fields and enhanced with proper data augmentations tends to have promising robustness against corruptions.
> - In this revision, we supplement more advanced techniques on improving the corruption robustness from the RoboDepth Challenge -- an academic competition developed based on the benchmarks and datasets in RoboDepth. Please kindly refer to this [technical report](https://arxiv.org/pdf/2307.15061.pdf) for more details.
> - Our winning solutions reveal the following aspects that can improve the corruption robustness of depth estimation systems:
>   - *Spatial- and frequency-domain augmentations:* Observing that the common data corruptions like blurs and noises contain distinct representations in both spatial and frequency domains, new data augmentation techniques can be proposed to enhance the feature learning.
>   - *Masked image modeling:* The masking-based image reconstruction approach exhibits potential for improving corruption robustness; this simple operation encourages the model to learn more robust representations by decoding masked signals from the remaining ones.
>   - *Image restoration and super-resolution:* The off-the-shelf restoration and super-resolution networks can be leveraged to handle degradation during the test time, such as noise contamination, illumination changes, and image compression.
>   - *Adversarial training:* The joint adversarial objectives between the depth estimation and a noise generator facilitate robust feature learning; such an approach also maintains the performance on in-distribution scenarios while tackling out-of-distribution cases.
>   - *Diffusion-based noise suppression:* The denoising capability of diffusion is naturally suitable for handling corruption situations; direct use of the denoising step in the pre-trained diffusion model could help suppress the noises introduced by different data corruptions.
>   - *Vision-language pre-training:* Leveraging the pre-trained text features and aligning them to the extracted image features via an adapter is popular among recent studies and is proven helpful to improve the performance of various visual perception tasks.
>   - *Learned model ensembling:* The fusion among multiple models is commonly used in improving the performance; an efficient, proper, and simple model ensembling strategy often combines the advantages of different models and largely improves the performance.
>
> ---
> > ***Q2:** "While the focus on autonomous driving and safety-critical applications is valuable, how is this work's potential applicability of depth estimation in other domains or industries?"*
>
> **A:** We thank the reviewer for the comment.
> - While this work is mainly focusing on robust depth estimation in autonomous driving and safety-critical applications, we believe a similar pursuit can be extended to other depth estimation tasks in related domains, for example, medical image analysis, such as stereoscopic, endoscopy, and microscopy depth estimation.
> - Given that different tasks frequently involve data of varying structures (e.g. collected by different sensors), the types of corruption applied should be thoughtfully tailored to align with the specific nature of each task.
> - We maintain the belief that fostering the development of robust depth estimation techniques in the face of corruptions will pave the way for the creation of more dependable depth estimation systems. These systems can then effectively meet real-world demands during their deployment phase.
>
> ---
> Last but not least, we thank Reviewer 76NB again for the time and effort devoted to this review.

---

### Official Review · Reviewer_ji4t · 2023-07-21
**Good Contribution to Robust Depth Estimation**

**Rating:** 7
**Confidence:** 4
**Correctness:** The claims made in the submission are…
**Clarity:** The paper is well-written and easy to…

**Strengths:**

1. The proposed RoboDepth benchmark on evaluating the robustness of depth estimation models under corruptions addresses an important gap in current testing procedures, which often do not account for these common situations in practical scenarios.

2. The benchmark consists of 18 corruptions across three categories for evaluating indoor and outdoor depth estimation models, covering a broad range of realistic conditions. And it presents a comprehensive evaluation on 42 existing depth estimation models.

3. The authors have made their well-documented work available on GitHub.

**Additional Feedback:**

None.

**Documentation:**

The code has been open source and well-documented.

**Ethics:**

There is no particular ethical concern.

**Limitations:**

The authors addressed the limitations of their work, but not the societal impact.

**Opportunities For Improvement:**

1. The paper could benefit from additional details about the datasets, including the quantity of samples for each corruption type and the quantitative measurement of various severity levels. The presentation of the results lacks clarity on the performance variation of these methods concerning different severity levels. A more detailed analysis could reveal whether the performance of different methods is consistent across varying levels of severity.

2. The paper lacks a precise explanation of what qualifies as an out-of-distribution (OOD) situation, and how it contrasts with in-distribution scenarios. In the context of general OOD detection, the distribution usually refers to label distribution, which does not seem to be the case in this paper.

**Relation To Prior Work:**

The prior work is discussed in the related work section.

**Summary And Contributions:**

This paper introduces RoboDepth, a benchmark designed for assessing the robustness of depth estimation models when faced with common distortions in real-world situations. The authors introduce three datasets and metrics for evaluating both indoor and outdoor depth estimation models. They conduct an evaluation of 42 cutting-edge models, investigating their robustness under varying types of corruption. Additionally, the paper provides insights into the design principles for creating more robust depth estimation models.

---

> ### Author Response · Authors · 2023-08-20
> **Authors' Response to Reviewer ji4t**
>
> We thank Reviewer ji4t for devoting time to this review and providing valuable comments. Our responses are as follows.
>
> ---
> > ***Q1**: "The paper could benefit from additional details about the datasets, including the quantity of samples for each corruption type and the quantitative measurement of various severity levels."*
>
> **A:** We thank the reviewer for the comment.
> - We generate for each corruption type five images (from different severity levels); there are a total of 18 corruptions for KITTI-C and 15 corruptions for NYUDepth2-C. We summarize the number of samples per corruption as follows:
> |Dataset|# Type|# Level|Resolution|# Images (Per Corrupt)|# Images (Total)|
> |-|:-:|:-:|:-:|:-:|:-:|
> |KITTI-C|18|5|640x192|3,485|62,730|
> |KITTI-C (High-Res)|18|5|1024x320|3,485|62,730|
> |NYUDepth2-C|15|4|640x480|2,616|39,240|
>
> - We have added the quantitative measurement of various severity levels on this GitHub page: https://github.com/ldkong1205/RoboDepth/blob/main/docs/RESULT.md#severity-level.
> - We also provide documents for models benchmarked in RoboDepth, which cover the detailed results of each corruption type across different severity levels. As an example, the document page of the baseline MonoDepth2 is at: https://github.com/ldkong1205/RoboDepth/blob/main/docs/results/kitti_c/MonoDepth2_mono_640x192.md.
> - We have the following observations from the results: Generally speaking, the depth estimation error (Abs Rel) increases as the severity level gets higher. The performance of different depth estimation methods is consistent across varying levels of severity. The degree of degradation for each model shows different trends in different corruption types. We thank the reviewer for the suggestion and will supplement these results and observations in the revised manuscript.
>
> ---
> > ***Q2:** "The paper lacks a precise explanation of what qualifies as an out-of-distribution (OOD) situation, and how it contrasts with in-distribution scenarios."*
>
> **A:** We thank the reviewer for the comment.
> - In this work, we concentrate more on OoD robustness and generalization; the depth estimation models are evaluated in an out-of-training-distribution situation, where there is a domain change between the training (often "clean") and the testing (corruptions that have a high likelihood to occur in the real world) distributions.
> - We benchmark depth estimation models on 18 out-of-training-distribution scenarios which are mainly caused by three sources: 1) adverse weather and lighting conditions; 2) sensor failure and movement disturbances; and 3) data processing issues.
> - We measure the corruption robustness of depth estimation models by comparing their performance on the original "clean" sets and the corruption sets. Our goal is to encourage designs that can improve depth estimation accuracy when models are tested in real-world corruption scenarios.
>
> ---
> Last but not least, we thank Reviewer ji4t again for the time and effort devoted to this review.

---

> > ### Comment · Reviewer_ji4t · 2023-08-31
> > **Thanks for the clarification**
> >
> > I would like to thank the authors for addressing the comments. I raised my rating to 7, good paper, accept.

---

> > > ### Author Response · Authors · 2023-08-31
> > > **Authors' Response to Reviewer ji4t**
> > >
> > > We sincerely thank Reviewer ji4t for participating in the Author-Reviewer discussion session and providing positive feedback.
> > >
> > > ---
> > > Once again, we thank you for the time and effort devoted and the valuable comments drawn during this review.
> > >
> > > Best regards,
> > >
> > > The Authors

---

### Official Review · Reviewer_YXSL · 2023-07-22
**good benchmarking but the writing of paper itself can be improved**

**Rating:** 5
**Confidence:** 3
**Correctness:** Mostly
**Clarity:** Mostly

**Strengths:**

Comprehensive experimental results and discussions.

**Additional Feedback:**

None

**Documentation:**

Yes

**Limitations:**

Yes

**Opportunities For Improvement:**

- The 5 question-answer pairs in Sec. 4 can be replaced with the answer as the paragraph starter directly (as was done in Sec. 4.3) whenever possible.

- The tone of the paper should be more concentrated on the benchmarking apparatus, and less on the "datasets" which are merely corrupted versions of other researchers' work. Open-sourcing the code for the corruption procedures and providing researchers with the corrupted versions are great, but the resulting datasets themselves should not be considered or implied as a contribution. The term "our datasets" is improper in this sense.

- Figure 7 [right] and Q4 in Line 232 are confusing. "Abs Rel" in the figure is absent in the main text. How the argument for Q4 needs more discussion.

**Relation To Prior Work:**

Yes

**Summary And Contributions:**

The authors presented a set of 18 corruptions for evaluating a monocular depth estimation models (MDEs) and used it to create dataset for analyzing 42 MDE models for indoor and outdoor scenes.

---

> ### Author Response · Authors · 2023-08-20
> **Authors' Response to Reviewer YXSL**
>
> We thank Reviewer YXSL for devoting time to this review and providing valuable comments. Our responses are as follows.
>
> ---
> > ***Q1:** "The 5 question-answer pairs in Sec. 4 can be replaced with the answer as the paragraph starter directly (as was done in Sec. 4.3)."*
>
> **A:** Thanks for your advice. We have replaced the Q-A pairs in Sec. 4.2 with the style of Sec. 4.3 as suggested.
>
> ---
> > ***Q2:** "The tone of the paper should be more concentrated on the benchmarking apparatus, and less on the 'datasets'."*
>
> **A:** We thank the reviewer for the suggestion.
> - We have modified the tone to not claim a contribution for the generated datasets and focused more on benchmarking and analyzing.
> - As suggested, the terms "our datasets" have been changed to the corresponding names of these datasets (e.g. KITTI-C) in the main text and supplementary files.
>
> ---
> > ***Q3:** "Figure 7 [right] and Q4 in Line 232 are confusing. "Abs Rel" in the figure is absent in the main text. How the argument for Q4 needs more discussion."*
>
> **A:** We thank the reviewer for the comment.
> - "Abs Rel" measures the depth estimation error rate, which was defined in Line 182. The specific calculation of "Abs Rel" is shown in Sec. B6 (Line 370) in the supplementary file.
> - The argument for Q4 is to verify the validity of corruption generations. Here we include more details as follows.
>   - We fine-tune with corruptions as augmentations on the KITTI dataset and test the models on real-world corruption datasets.
>   - **Goal:** Assuming that a corruption simulation is realistic enough to reflect real-world situations, a corruption-augmented model should achieve better generalizability than the "clean" model when tested on real-world corruption datasets.
>   - **Approach:** We validate this using SeasonDepth [R1] and Make3D [R2] (Figure 7 right in the manuscript), as well as nuScenes [R3], nuScenes-Night [R3], and Foggy-Cityscapes [R4] (new in the revision, which are datasets of larger scales). We adopt MonoDepth2 [R5] as the baseline, which is trained on KITTI for 21 epochs and fine-tuned with corruptions for 6 epochs with a small learning rate. We also test training with corruptions from scratch for 21 epochs and find the performance is similar to fine-tuning.
>   - **Results:** We provide the results and analysis on this GitHub page (also copied in the response to Reviewer j7EB's Q1): https://github.com/ldkong1205/RoboDepth/blob/main/docs/VALIDITY.md#study-2-robust-fine-tuning.
>   - We observe from these results that models trained with corruptions as augmentations often achieve better performance when tested on real-world corruption sets. The generated corruptions are helpful for closing the distribution gap between the clean training set and real-world scenarios, which validates the fidelity of these corruption simulations.
>
> - In the revision, we provide another study to further consolidate the argument for Q4.
>   - We calculate and measure the pixel distributions of synthetic corruption image sets and real-world corruption image sets.
>   - **Goal:** Assuming that a corruption simulation is realistic enough to reflect real-world situations, the distribution of a corrupted "clean" set should be similar to that of the real-world corruption set.
>   - **Approach:** We validate this using ACDC [R6], nuScenes [R3], and Cityscapes [R7], since these datasets contain: (1) real-world corruption data, and (2) "clean" data collected by the same sensor types from the same physical locations. We simulate corruptions using "clean" images and compare the distribution patterns with their corresponding real-world corrupted data. We do this to ensure that there is no extra distribution shift from aspects like sensor difference (e.g. FOVs and resolutions) and location discrepancy (e.g. environmental and semantic changes).
>   - **Results:** We provide the results and analysis on this GitHub page: https://github.com/ldkong1205/RoboDepth/blob/main/docs/VALIDITY.md#study-1-pixel-distribution.
>   - We observe that these synthetic corruptions can mimic real-world scenarios with good validity. We believe this, to a certain extent, ensures that the benchmarking results and analyses can be extrapolated to real data.
>
> ---
> **References:**
> - [R1] “SeasonDepth: Cross-season monocular depth prediction dataset and benchmark under multiple environments.” ICML Workshop, 2022.
> - [R2] “Make3D: Learning 3D scene structure from a single still image.” TPAMI, 2008.
> - [R3] “nuScenes: A multimodal dataset for autonomous driving.” CVPR, 2020.
> - [R4] “Semantic foggy scene understanding with synthetic data.” IJCV, 2018.
> - [R5] “Digging into self-supervised monocular depth prediction.” ICCV, 2019.
> - [R6] “ACDC: The adverse conditions dataset with correspondences for semantic driving scene understanding.” ICCV, 2021.
> - [R7] “The CityScapes dataset for semantic urban scene understanding.” CVPR, 2016.
>
> ---
> Last but not least, we thank Reviewer YXSL again for the time and effort devoted to this review.

---

### Official Review · Reviewer_j7EB · 2023-07-25
**Good work, but the relevance of simulated augmentations could be questionable?**

**Rating:** 7
**Confidence:** 5

**Strengths:**

+ The topic addressed is relevant.
+ The paper is mostly well structured and written. The evaluation metrics are very clear, also the dataset specifications, and the experimental validation of the data is thorough and solid.
+ The validation is extense, including results for 42 depth models and interesting conclusions.
+ The dataset has been used in a challenge at ICRA 2023, with a reasonable success, so the adoption of the data has already started.

**Additional Feedback:**

Nothing additional to comment, I am eager to read the authors responses to my comments.

**Clarity:**

The paper is mostly clear and nicely written, with the previously mentioned exception of missing details on the synthetic corruptions applied.

**Correctness:**

The dataset is created in a sound manner, and the data validation experiments suggest that all is correct. The methodology is solid and seemingly correct.

**Documentation:**

The documentation is complete and credible regarding the use of the data and maintenance plans. There are sufficient details to work with the data and report results in an appropriate manner.

**Ethics:**

In my opinion, there are no ethical concerns in the dataset presented in the submission.

**Limitations:**

The authors have included a section discussing the limitations of their work. In my opinion, the limitations associated to the use of synthetic data, currently missing, should be discussed. I cannot see a potential negative societal impact of this work, and it is not discussed in the authors' work.

**Opportunities For Improvement:**

- My most clear criticism is the validity of the synthetic augmentations. Are they realistic enough, so that the results and conclusions can be extrapolated to real data? The authors should extend the analysis in Figure 7 right with further details, in order to accurately assess if this experiment is a solid indication of the validity of the synthetic augmentations. In second place, the authors should include more details on how the synthetic corruptions were generated. In the paper the details are missing, and in the supplementary only references to the libraries used are included, without algorithmic details or performance metrics that inform on their realism.
- I think it would be important to report performance metrics comparing training and testing on clean data, and training and testing on corrupted data. The models tested in corrupted data show a degradation in the performance. However, it is not clear if the degradation is due to the domain change or to a different signal-to-noise ratio, and even training in corrupted data would lead to a degradation.

**Relation To Prior Work:**

Prior work is appropriately cited and discussed, up to my knowledge.

**Summary And Contributions:**

The dataset proposed is composed of 18 synthetic augmentations of the KITTI and NYUDepth datasets that reflect common situations in visual data (variations in weather and lighting, motion blur, defocus or processing artifacts). The goal is to assess the performance of depth prediction networks in OoD data, and to progress towards models that generalize to such situations.

---

> ### Author Response · Authors · 2023-08-20
> **Authors' Response to Reviewer j7EB**
>
> We thank Reviewer j7EB for devoting time to this review and providing insightful comments. Our responses are as follows.
>
> ---
> > ***Q1:** "Validity of synthetic augmentations: are they realistic enough, so that the results and conclusions can be extrapolated to real data? The authors should extend the analysis in Figure 7 right with further details."*
>
> **A:** Thanks for the comment. We agree with the reviewer that the validity of corruption simulations lays the foundation of our benchmarks. We conduct the following two types of experiments to further validate the quality of our simulated corruptions.
> - **Study 1:** We calculate and measure the pixel distributions of synthetic corruption image sets and real-world corruption image sets.
>   - **Goal:** Assuming that a corruption simulation is realistic enough to reflect real-world situations, the distribution of a corrupted "clean" set should be similar to that of the real-world corruption set.
>   - **Approach:**
>     - We validate this using ACDC [R1], nuScenes [R2], and Cityscapes [R3], since these datasets contain: (1) real-world corruption data, and (2) "clean" data collected by the same sensor types from the same physical locations.
>     - We simulate corruptions using "clean" images and compare the distribution patterns with their corresponding real-world corrupted data. We do this to ensure that there is no extra distribution shift from aspects like sensor difference (e.g. FOVs and resolutions) and location discrepancy (e.g. environmental and semantic changes).
>   - **Results:**
>     - We provide the results and analysis on this GitHub page: https://github.com/ldkong1205/RoboDepth/blob/main/docs/VALIDITY.md#study-1-pixel-distribution.
>     - We observe that these synthetic corruptions can mimic real-world scenarios with good validity. We believe this, to a certain extent, ensures that the benchmarking results and analyses can be extrapolated to real data.
> - **Study 2:** We fine-tune with corruptions as augmentations on the KITTI dataset and test the models on real-world corruption datasets.
>   - **Goal:** Assuming that a corruption simulation is realistic enough to reflect real-world situations, a corruption-augmented model should achieve better generalizability than the "clean" model when tested on real-world corruption datasets.
>   - **Approach:**
>     - We validate this using SeasonDepth [R4] and Make3D [R5] (Figure 7 right in the manuscript), as well as nuScenes [R2], nuScenes-Night [R2], and Foggy-Cityscapes [R6] (new in the revision, which are datasets of larger scales).
>     - We adopt MonoDepth2 [R7] as the baseline, which is trained on KITTI for 21 epochs and fine-tuned with corruptions for 6 epochs with a small learning rate. We also test training with corruptions from scratch for 21 epochs and find the performance is similar to fine-tuning.
>   - **Results:**
>     - We provide the results and analysis on this GitHub page (copied below): https://github.com/ldkong1205/RoboDepth/blob/main/docs/VALIDITY.md#study-2-robust-fine-tuning.
>     - [Table 1] nuScenes
> |Train|Backbone|Resolution|CorruptAug|Abs Rel|Sq Rel|RMSE|RMSE log|a1|a2|a3|
> |:-:|:-:|:-:|:-:|:-:|:-:|:-:|:-:|:-:|:-:|:-:|
> | KITTI | ResNet-18 | 640x192 | No  | 0.304 | 3.472 | 9.068 | 0.409 | 0.563 | 0.794 | 0.890 |
> | KITTI | ResNet-18 | 640x192 | Yes | 0.297 | 2.991 | 8.790 | 0.405 | 0.558 | 0.794 | 0.893 |
> | KITTI | ResNet-50 | 640x192 | No  | 0.302 | 3.219 | 9.054 | 0.416 | 0.555 | 0.786 | 0.886 |
> | KITTI | ResNet-50 | 640x192 | Yes | 0.294 | 2.947 | 8.754 | 0.404 | 0.565 | 0.795 | 0.892 |
>     - [Table 2] nuScenes-Night
> |Train|Backbone|Resolution|CorruptAug|Abs Rel|Sq Rel|RMSE|RMSE log|a1|a2|a3|
> |:-:|:-:|:-:|:-:|:-:|:-:|:-:|:-:|:-:|:-:|:-:|
> | KITTI | ResNet-18 | 640x192 | No  | 0.397 | 3.408 | 8.700 | 0.513 | 0.387 | 0.659 | 0.822 |
> | KITTI | ResNet-18 | 640x192 | Yes | 0.362 | 3.149 | 8.391 | 0.477 | 0.434 | 0.714 | 0.852 |
> | KITTI | ResNet-50 | 640x192 | No  | 0.418 | 3.599 | 8.928 | 0.539 | 0.363 | 0.626 | 0.802 |
> | KITTI | ResNet-50 | 640x192 | Yes | 0.357 | 3.128 | 8.168 | 0.462 | 0.444 | 0.723 | 0.861 |
>     - [Table 3] Foggy-Cityscapes
> |Train|Backbone|Resolution|CorruptAug|Abs Rel|Sq Rel|RMSE|RMSE log|a1|a2|a3|
> |:-:|:-:|:-:|:-:|:-:|:-:|:-:|:-:|:-:|:-:|:-:|
> | KITTI | ResNet-18 | 416x128 | No  | 0.421 | 7.057 | 15.207 | 0.527 | 0.360 | 0.636 | 0.806 |
> | KITTI | ResNet-18 | 416x128 | Yes | 0.385 | 6.310 | 14.654 | 0.489 | 0.399 | 0.682 | 0.836 |
> | KITTI | ResNet-18 | 512x256 | No  | 0.364 | 6.371 | 14.690 | 0.483 | 0.440 | 0.703 | 0.838 |
> | KITTI | ResNet-18 | 512x256 | Yes | 0.349 | 5.645 | 14.723 | 0.488 | 0.434 | 0.698 | 0.834 |
>     - We observe from these results that models trained with corruptions as augmentations often achieve better performance when tested on real-world corruption sets. The generated corruptions are helpful for closing the distribution gap between the clean training set and real-world scenarios, which validates the fidelity of these corruption simulations.

---

> > ### Author Response · Authors · 2023-08-20
> > **Authors' Response to Reviewer j7EB**
> >
> > > ***Q2:** "Include more details on how the synthetic corruptions were generated."*
> >
> > **A:** Thanks for the comment.
> > - We include simulation details on this GitHub page: https://github.com/ldkong1205/RoboDepth/blob/main/docs/CREATE.md#simulation-algorithm.
> > - We summarize the rationale behind each synthetic corruption as follows:
> >   - `brightness` alters the HSV color space of an image, adjusting the brightness component.
> >   - `dark` reduces the overall brightness of the image and introduces noise to mimic the challenges of capturing images in low-light environments.
> >   - `fog` adds a foggy texture generated through plasma fractals. The fog effect is controlled by parameters that determine the thickness and smoothness, creating a visual distortion resembling the appearance of fog.
> >   - `frost` applies a simulated frost effect to an image by overlaying frost textures obtained from pre-defined frost images.
> >   - `snow` generates a layer of snow-like particles and adds motion blur to create the appearance of falling snowflakes.
> >   - `contrast` modifies the contrast of an image by adjusting pixel values around their mean. The degree of adjustment is controlled by a parameter, resulting in enhanced or reduced contrast.
> >   - `defocus_blur` simulates the appearance of an out-of-focus photograph. This is achieved by convolving the image with a circular disk-shaped kernel.
> >   - `glass_blur` simulates the distortion caused by viewing an image through a frosted glass surface. This effect is achieved by locally shuffling pixels and applying Gaussian blurring.
> >   - `motion_blur` applies a blurring kernel that represents the motion of a camera or object during exposure. This creates the illusion of movement or motion streaks.
> >   - `zoom_blur` applies a series of zoomed and cropped layers with varying zoom factors. These layers are then combined to create the illusion of zooming or rushing effect toward the center of the image.
> >   - `elastic_transform` mimics the deformation of objects under stress or tension. It uses a combination of random displacement fields and interpolation to create the distortion effect.
> >   - `color_quant` applies color quantization to an image, reducing the number of distinct colors present in the image.
> >   - `gaussian_noise` adds Gaussian noise, introducing random fluctuations in pixel values. This simulates the effect of noise in a photographic or digital image.
> >   - `impulse_noise` adds salt-and-pepper noise to an image, introducing randomly occurring white and black pixels that resemble salt and pepper grains.
> >   - `shot_noise` simulates shot noise that occurs due to the discrete nature of light particles (photons) hitting a sensor during image capture. This noise appears as random variations in pixel intensity.
> >   - `iso_noise` simulates the noise that can be introduced in images due to higher ISO settings in photography. It combines Poisson noise and Gaussian noise to replicate the characteristics of ISO noise.
> >   - `pixelate` reduces the image resolution by creating larger blocks of pixels. This results in a mosaic-like appearance where image details are simplified.
> >   - `jpeg_compression` applies JPEG compression to an image, which involves encoding the image in a lossy format.
> >
> > - The specific configurations of our simulation algorithms are summarized as follows:
> > |Corruption|Parameter|Level 1|Level 2|Level 3|Level 4|Level 5|
> > |:-:|:-:|:-:|:-:|:-:|:-:|:-:|
> > | `brightness`|adjustment in HSV space|0.1|0.2|0.3|0.4|0.5|
> > | `dark` |scale factor|0.6|0.5|0.4|0.3|0.2|
> > | `fog` |(thickness, smoothness)|(1.5, 2.0)|(2.0, 2.0)|(2.5, 1.7)|(2.5, 1.5)|(3.0, 1.4)|
> > | `frost` |(frost intensity, texture influence) |(1.00, 0.40)|(0.80, 0.60)|(0.70, 0.70)|(0.65, 0.70)|(0.60, 0.75) |
> > | `snow` |(mean, std, scale, threshold, blur radius, blur std, blending ratio)|(0.1, 0.3, 3.0, 0.5, 10.0, 4.0, 0.8)|(0.2, 0.3, 2, 0.5, 12, 4, 0.7)|(0.55, 0.3, 4, 0.9, 12, 8, 0.7)|(0.55, 0.3, 4.5, 0.85, 12, 8, 0.65)|(0.55, 0.3, 2.5, 0.85, 12, 12, 0.55)|
> > | `contrast` | adjustment of pixel mean |0.40|0.30|0.20|0.10|0.05|
> > | `defocus_blur` | (kernel radius, alias blur) | (3.0, 0.1) | (4.0, 0.5) | (6.0, 0.5) | (8.0, 0.5) | (10.0, 0.5) |
> > | `glass_blur` | (sigma, max delta, iterations) | (0.7, 1.0, 2.0) | (0.9, 2.0, 1.0) | (1.0, 2.0, 3.0)|(1.1, 3.0, 2.0) | (1.5, 4.0, 2.0) |
> > | `motion_blur` | (radius, sigma) | (10, 3) | (15, 5) | (15, 8) | (15, 12) | (20, 15) |
> > | `zoom_blur` | (low, high, step size) | (1.00, 1.11, 0.01) | (1.00, 1.16, 0.01) | (1.00, 1.21, 0.02) | (1.00, 1.26, 0.02) | (1.00, 1.31, 0.03) |
> > | `elastic_transform` | deformation |0.050|0.065|0.085|0.100|0.120|
> > | `color_quant` | bit number |5|4|3|2|1|
> > | `gaussian_noise` | noise scale |0.08 | 0.12 | 0.18 | 0.26 | 0.38 |
> > | `impulse_noise` | noise amount |0.03|0.06|0.09|0.17|0.27|
> > | `shot_noise` | photon number |60|25|12|5|3|
> > | `iso_noise` | noise scale |0.08|0.12|0.18|0.26|0.38|
> > | `pixelate` | resize factor |0.60|0.50|0.40|0.30|0.25|
> > | `jpeg_compression` | compression quality | 25 | 18 | 15 | 10 | 7 |

---

> > > ### Author Response · Authors · 2023-08-20
> > > **Authors' Response to Reviewer j7EB**
> > >
> > > > ***Q3:** "Report performance metrics comparing training and testing on clean data, and training and testing on corrupted data."*
> > >
> > > **A:** Thanks for your suggestion.
> > > - We report the results of training and testing on clean data on this GitHub page: https://github.com/ldkong1205/RoboDepth/blob/main/docs/ENHANCE.md#train--test-on-clean-data.
> > > - We report the results of training and testing on corrupted data on this GitHub page: https://github.com/ldkong1205/RoboDepth/blob/main/docs/ENHANCE.md#train--test-on-corrupted-data.
> > > - We observe from these results that:
> > >   - *"Train on Clean, Test on Clean"*: Corresponds to the baseline result from the conventional depth estimation paradigm.
> > >   - *"Train on Corruptions, Test on Clean"*: Corresponds to a training distribution shift towards corrupted data but the testing distribution remains clean, which is not likely to occur in the real world. We observe that:
> > >     - Training with each corruption at a time or a combination of multiple corruptions would lead to performance degradation on the clean testing set.
> > >     - The probabilities of accessing corruptions during training play an important role; compared to training on all corrupted data (CorruptAug = 1.0), a hybrid of clean and corruptions results in lower depth estimation error.
> > >   - *"Train on Clean, Test on Corruptions"*: Corresponds to a testing distribution shift towards corrupted data while the model is trained on clean data, which is reflecting the real-world scenarios. We observe that the models tested show a degradation in the performance and the degree of degradation is different across various corruption types. This shows that the corruption sets are training-distribution-shifted domains.
> > >   - *"Train on Corruptions, Test on Corruptions"*: Corresponds to both training and testing distributions shifting towards corrupted data. We can see that:
> > >     - The performance on corruption sets is improved when the models are trained and tested with the same or similar corruption types.
> > >     - The performance on corruption sets might become worse when the models are trained on one type of corruptions (e.g. sensor & movement) and tested on another type (e.g. weather & lighting).
> > >   - To summarize, the corruption simulations are basically distribution-shifted domains that aim to mimic real-world scenarios; a robust model is expected to maintain good performance when tested on these corruption sets.
> > >
> > > ---
> > > **References:**
> > > - [R1] C. Sakaridis, D. Dai, and L. V. Gool. "ACDC: The adverse conditions dataset with correspondences for semantic driving scene understanding." ICCV, 2021.
> > > - [R2] C., Holger, V. Bankiti, A. H. Lang, S. Vora, V. E. Liong, Q. Xu, A. Krishnan, Y. Pan, G. Baldan, and O. Beijbom. "nuScenes: A multimodal dataset for autonomous driving." CVPR, 2020.
> > > - [R3] M. Cordts, M. Omran, S. Ramos, T. Rehfeld, M. Enzweiler, R. Benenson, U. Franke, S. Roth, and B. Schiele. "The CityScapes dataset for semantic urban scene understanding." CVPR, 2016.
> > > - [R4] H. Hu, B. Yang, Z. Qiao, S. Liu, D. Zhao, and H. Wang. “SeasonDepth: Cross-season monocular depth prediction dataset and benchmark under multiple environments.” ICML Workshop, 2022.
> > > - [R5] A. Saxena, M. Sun, and A. Y. Ng. “Make3D: Learning 3D scene structure from a single still image.” IEEE TPAMI, 2008.
> > > - [R6] C. Sakaridis, D. Dai, and L. V. Gool. “Semantic foggy scene understanding with synthetic data.” IJCV, 2018.
> > > - [R7] C. Godard, O. M. Aodha, M. Firman, and G. J. Brostow. “Digging into self-supervised monocular depth prediction.” ICCV, 2019.
> > >
> > > ---
> > > Last but not least, we thank Reviewer j7EB again for the time and effort devoted to this review.

---

> > > > ### Comment · Reviewer_j7EB · 2023-08-25
> > > > **Great work, very nice discussion and experimental additions to the paper**
> > > >
> > > > I would like to thank the authors for the hard and excellent experimental work that they did to address my concerns. The results show quite clearly that the simulated corruptions are sufficiently realistic, and the additional experiments show and quantify the performance degradation  of the models due to the corruption modes analyzed, and how the authors' dataset can be useful to address research challenges in that direction.
> > > >
> > > > Thank you also for the additional details on how the corruption modes are generated, I think the paper greatly benefits from having these details more straightforwardly available.
> > > >
> > > > I raised my rating to 7, good paper, accept.

---

> > > > > ### Author Response · Authors · 2023-08-25
> > > > > **Authors' Response to Reviewer j7EB**
> > > > >
> > > > > We sincerely thank Reviewer j7EB for participating in the Author-Reviewer discussion session and providing positive feedback. Your raised comments have helped us improve the quality and comprehensiveness of this work.
> > > > >
> > > > > ---
> > > > > Once again, we thank Reviewer j7EB for the time and effort devoted and the valuable comments drawn during this review.
> > > > >
> > > > > Best regards,
> > > > >
> > > > > The Authors

---

### Author Response · Authors · 2023-08-20
**General Response**

We sincerely thank our reviewers for devoting time to this review and offering valuable comments.

---
We are glad to see that the reviewers are acknowledging:
- *"The dataset specifications and the experimental validation of the data is thorough and solid"* (Reviewer j7EB)
- *"Comprehensive experimental results and discussions"* (Reviewer YXSL)
- *"The proposed RoboDepth benchmark addresses an important gap in current testing procedures, presents a comprehensive evaluation, and has well-documented work available on GitHub"* (Reviewer ji4t)
- *"The dataset collections are comprehensive and meaningful; the project is well-documented and maintained"* (Reviewer 76NB)
- *"Proposed a systematic approach to test different methods over a large set of possible corruptions"* (Reviewer pgHi)

---
We have polished the content, added more evidence and results, and clarified potential issues in the revision. Specifically, we included the following changes according to reviewers’ insightful comments:
- **Description & Assessment**
  - As suggested by Reviewer j7EB and Reviewer YXSL, we conducted a [validity assessment](https://github.com/ldkong1205/RoboDepth/blob/main/docs/VALIDITY.md#study-1-pixel-distribution) for the corruption simulations.
  - As suggested by Reviewer j7EB, we supplemented [more details](https://github.com/ldkong1205/RoboDepth/blob/main/docs/CREATE.md#simulation-algorithm) on how corruptions are generated.
- **Experiments & Observation**
  - As suggested by Reviewer j7EB, we supplemented [more results](https://github.com/ldkong1205/RoboDepth/blob/main/docs/ENHANCE.md#train--test-on-clean-data) on training and testing with corrupted data.
  - As suggested by Reviewer j7EB and Reviewer pgHi, we supplemented [robust fine-tuning results](https://github.com/ldkong1205/RoboDepth/blob/main/docs/VALIDITY.md#study-2-robust-fine-tuning) on real-world datasets.
  - As suggested by Reviewer ji4t, we added [results](https://github.com/ldkong1205/RoboDepth/blob/main/docs/RESULT.md#severity-level) for models under each severity level.
- **Writing & Elaboration**
  - As suggested by Reviewer YXSL, we modified the tone of this paper to concentrate more on the benchmarking apparatus.
  - As suggested by Reviewer YXSL, we modified the Q-A pairs in Sec. 4.2 to the paragraph starter style.
- **Reference & Literature Review**
  - As suggested by Reviewer pgHi, we supplemented the details of the [RoboDepth Challenge](https://robodepth.github.io) and attached our [technical report](https://arxiv.org/pdf/2307.15061.pdf) for sharing methods and advancements in improving depth estimation robustness.
  - As suggested by Reviewer pgHi, we added relevant references on adversarial training for monocular depth estimation.

---
We would like to re-emphasize the technical contributions of this work:
- We introduce RoboDepth, a systematically-designed robustness evaluation suite for depth estimation under data corruptions, sensor failure, and style shifts.
- We benchmarked intensively on state-of-the-art depth estimation models from indoor and outdoor scenes and analyzed their robustness against corruptions.
- Based on our observations, we draw in-depth discussion and analysis on the design considerations of building more robust depth estimation models for reliable, scalable, and practical applications.

---
We will actively participate in the Author-Reviewer discussion session. Please don’t hesitate to let us know of any additional comments on the manuscript, GitHub repository, and changes.

---
Last but not least, we thank the reviewers again for the time and effort devoted to this review.

---

### Author Response · Authors · 2023-08-25
**Looking forward to discussion**

Dear Reviewers,

Thanks for devoting time to review this paper and providing valuable comments.

We hope we have addressed your concerns and supplemented the necessary information about our benchmarks. It will be great to discuss with you if there are any other areas of improvement.

---
Just a recap:
We have polished the content, added more evidence and results, and clarified potential issues in the revision. Specifically, we included the following changes according to reviewers’ insightful comments:
- Description & Assessment
  - As suggested by Reviewer j7EB and Reviewer YXSL, we conducted a [validity assessment](https://github.com/ldkong1205/RoboDepth/blob/main/docs/VALIDITY.md#study-1-pixel-distribution) for the corruption simulations.
  - As suggested by Reviewer j7EB, we supplemented [more details](https://github.com/ldkong1205/RoboDepth/blob/main/docs/CREATE.md#simulation-algorithm) on how corruptions are generated.
- Experiments & Observation
  - As suggested by Reviewer j7EB, we supplemented [more results](https://github.com/ldkong1205/RoboDepth/blob/main/docs/ENHANCE.md#train--test-on-clean-data) on training and testing with corrupted data.
  - As suggested by Reviewer j7EB and Reviewer pgHi, we supplemented [robust fine-tuning results](https://github.com/ldkong1205/RoboDepth/blob/main/docs/VALIDITY.md#study-2-robust-fine-tuning) on real-world datasets.
  - As suggested by Reviewer ji4t, we added the [quantitative measurement](https://github.com/ldkong1205/RoboDepth/blob/main/docs/RESULT.md#severity-level) for models under each severity level.
- Writing & Elaboration
  - As suggested by Reviewer YXSL, we modified the tone of this paper to concentrate more on the benchmarking apparatus.
  - As suggested by Reviewer YXSL, we modified the Q-A pairs in Sec. 4.2 to the paragraph starter style.
- Reference & Literature Review
  - As suggested by Reviewer pgHi, we supplemented the details of the [RoboDepth Challenge](https://robodepth.github.io/) and attached our [technical report](https://arxiv.org/pdf/2307.15061.pdf) for sharing methods and advancements in improving depth estimation robustness.
  - As suggested by Reviewer pgHi, we added relevant references on adversarial training for monocular depth estimation.

---
We have also provided detailed point-to-point responses for each reviewer below. Please kindly refer to the response windows.

---
Once again, we thank the reviewers for the time and effort devoted to this review and look forward to hearing your thoughts on our responses.

Yours sincerely,

The Authors

---

### Decision · Program_Chairs · 2023-09-22

**Decision:**

Accept (Poster)

**Comment:**

Four out of five reviewers liked this paper, which offers valuable data for testing depth estimation models. The only reviewer with a rating below the acceptance threshold (5) raises minor concerns about the writing and positioning of the work. The authors addressed them well in the revision (but the reviewer did not seem to check back).